# Synthetic ZFTA fusions pinpoint disordered protein domain acquisition as a mechanism of brain tumorigenesis

Over 95% of ependymomas that arise in the cortex are driven by a gene fusion involving the zinc finger translocation-associated (ZFTA) protein. Here, using super-resolution and lattice light-sheet microscopy, we demonstrate that the most frequent fusion variant, ZFTA–RELA (ZR), forms dynamic nuclear condensates that are required for oncogene expression and tumorigenesis. Mutagenesis studies of ZR reveal a key intrinsically disordered region (IDR) in RELA that governs condensate formation. Condensate-modulating IDR mutations introduced into ZR impaired its genomic occupancy at oncogenic loci and inhibited the recruitment of transcriptional effector proteins, such as MED1, BRD4 and RNA polymerase II. Using nuclear magnetic resonance spectroscopy, we examined the DNA-binding residues of the critical zinc finger (ZF1) found in ZR and characterized their significance for condensate formation, genomic binding and oncogene activation. We generated synthetic ZFTA fusion proteins where IDRs from known condensate-forming proteins were grafted into ZR. Synthetic ZFTA fusion oncoproteins utilizing IDRs from EWS and FUS restored condensate formation, oncogene transcription and tumour initiation in mice. These findings provide key insights into the oncogenic mechanism of ZR and the importance of IDR acquisition in fusion oncoproteins in brain cancer.

Ependymomas (EPN) are aggressive brain tumours without targeted therapies[1–3]. Standard treatment for EPN has not changed in the past 30 years, consisting of maximal-safe surgery followed by radiotherapy[3]. EPN are a heterogeneous group of malignancies with over 95% of cortical brain EPN driven by gene fusions involving the zinc finger translocation-associated (*ZFTA*) gene, and less than 5% involving the gene encoding *YAP1*. Tumours driven by *ZFTA* gene fusions have poorer survival compared with *YAP1*-fused tumours that are effectively treated with surgery and radiotherapy[1]. These fusion proteins are essential drivers of the disease as expression in neural stem cell (NSC) models is sufficient to drive tumour initiation[4–8]. Translocation of *ZFTA* to its most frequent partner *RELA* (denoted ZR) leads to constitutive nuclear localization, aberrant DNA binding and activation of oncogenic expression programs[4,5,8,9]. Despite identification of key biological processes directed by the ZR fusion oncoprotein (FO) during neoplastic

transformation, our understanding of the molecular mechanisms of FO-driven oncogene activation in EPN remains incompletely understood[2]. Only recently have studies revealed potential regulatory mechanisms that govern *ZFTA* expression, which is elevated in ependymal cells during embryonic brain development[10]. However, so far, there have been few studies investigating the cellular localization or molecular function of the ZFTA protein.

Our previous work showed that one of the first stages of tumour initiation is facilitated by DNA binding of ZR, driven by $C_2H_2$ zinc finger (ZF) domains within ZFTA[4]. ZR preferentially recruits a host of transcriptional effector proteins such as BRD4, EP300, MED1 and RNA polymerase II (RNAPII), to highly active EPN target genes marked by super- and stretch-enhancers[4]. The underlying regulatory mechanisms and coordination between ZR and transcriptional co-factors to direct oncogenic transcription are unclear. Accumulating evidence in

✉e-mail: richard.kriwacki@stjude.org; stephen.mack@stjude.org

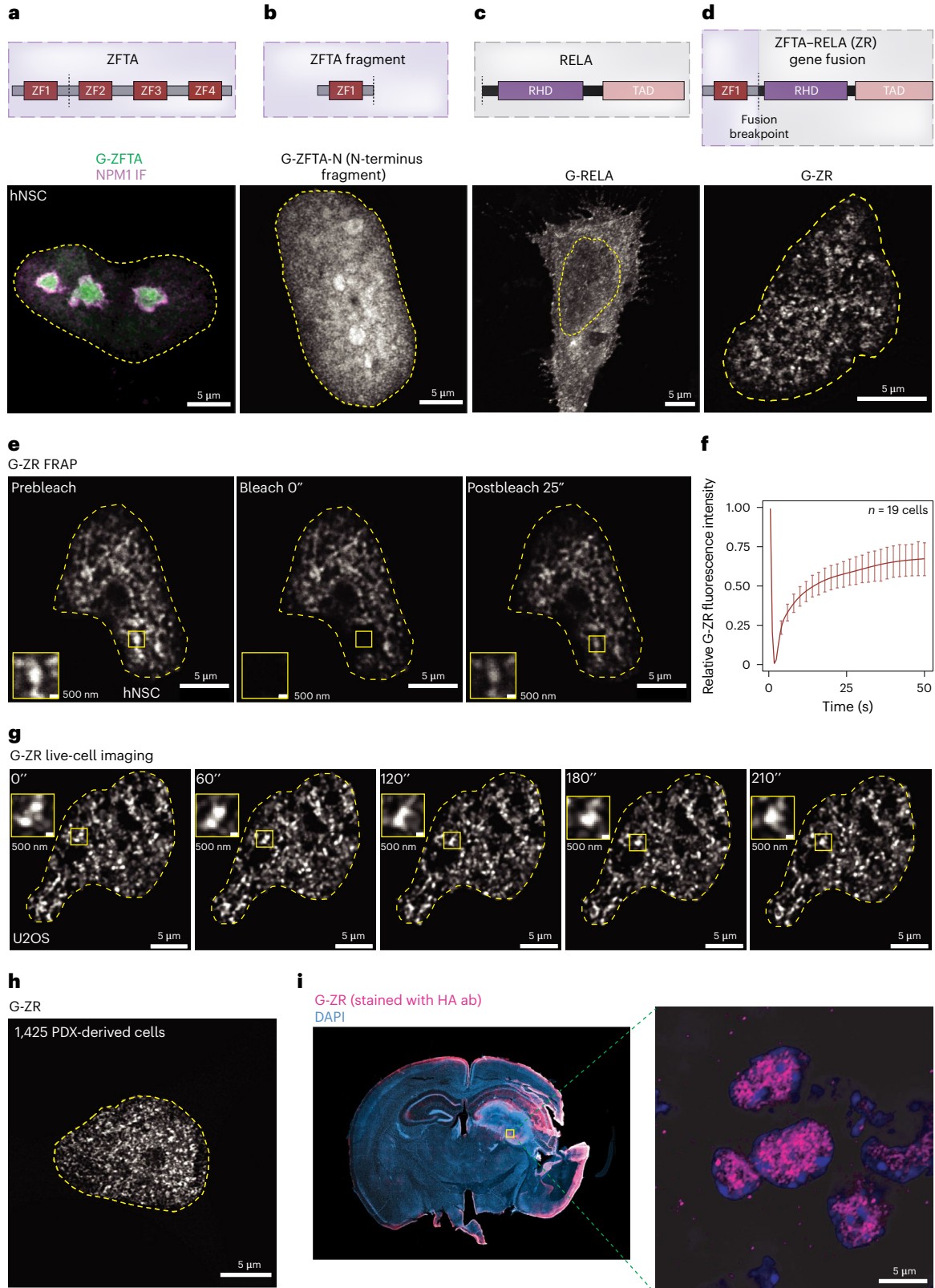

**Fig. 1 | ZFTA–RELA forms dynamic nuclear condensates in EPN models.**
**a**–**d**, Representative super-resolution images of hNSCs expressing G-ZFTA
(**a**), G-ZFTA N terminus (**b**), G-RELA (**c**) and G-ZR (**d**). A schematic cartoon of
the domains within each protein is shown above. The nucleolus is indicated
by NPM1 staining in **a**. **e**, Representative images of hNSCs expressing G-ZR
before bleaching, at the time of bleaching and after bleaching. A zoom-in of the
bleached area is shown in the bottom left corner of the images. **f**, Normalized
G-ZR fluorescence signal at the bleached area ($n$ = 19 independent cells). Data are
represented as a fitted curve through the mean values, using the LOESS method,
±s.d. **g**, Representative live-cell images of U2OS cells expressing G-ZR showing
a fusion event between two G-ZR condensates. **h**, A representative image of a
patient-derived xenograft (PDX) model electroporated with G-ZR. **i**, Staining of
a mouse brain slice harbouring a tumour driven by the G-ZR construct. The ZR
protein was detected using an anti-HA tag antibody.

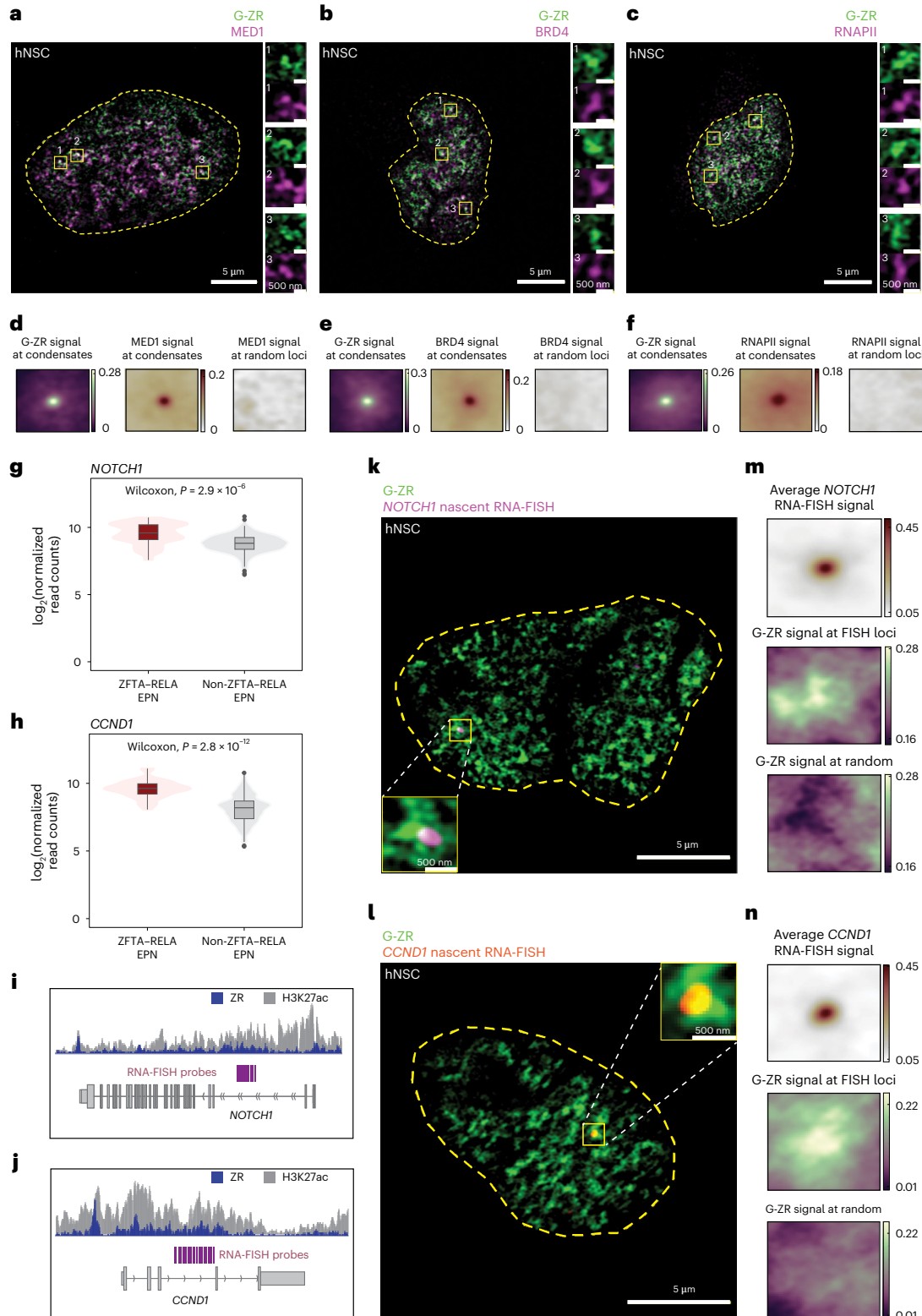

**Fig. 2 | ZFTA–RELA condensates are associated with active transcriptional machinery at EPN oncogenes. a–c**, Representative super-resolution images of hNSCs expressing G-ZR and stained for MED1 (**a**), BRD4 (**b**) and RNAPII (**c**). Zoomed-in views of three loci are provided on the right side of each image with channels separated. **d–f**, Averaged normalized G-ZR and staining target fluorescence intensity in a box at the centre of condensates as well as randomly selected loci for MED1 (**d**), BRD4 (**e**) and RNAPII (**f**). **g,h**, Expression comparison of *NOTCH1* (**g**) and *CCND1* (**h**) in primary tumour samples between ZFTA–RELA and non-ZFTA–RELA EPN (*n* = 35 for ZFTA–RELA EPN and 140 for non-ZFTA–RELA EPN; two-sided Wilcoxon test. Box plots represent the median and interquartile range (IQR). The whiskers extend to the smallest and largest values within 1.5× IQR. Outliers are shown with circles. **i,j**, The genomic locus of *NOTCH1* (**i**) and *CCND1* (**j**). H3K27ac marks (in primary tumour samples) are shown in grey, ZFTA–RELA binding pattern (in ZFTA–RELA-expressing H293T cells) in blue and RNA-FISH probes in purple. **k,l**, Representative super-resolution images of nascent RNA-FISH in hNSCs for *NOTCH1* (**k**) and *CCND1* (**l**). **m,n**, Averaged normalized G-ZR fluorescence and nascent RNA-FISH intensity in a box at the centre of RNA-FISH loci as well as randomly selected loci within the nucleus for *NOTCH1* (**m**) and *CCND1* (**n**).

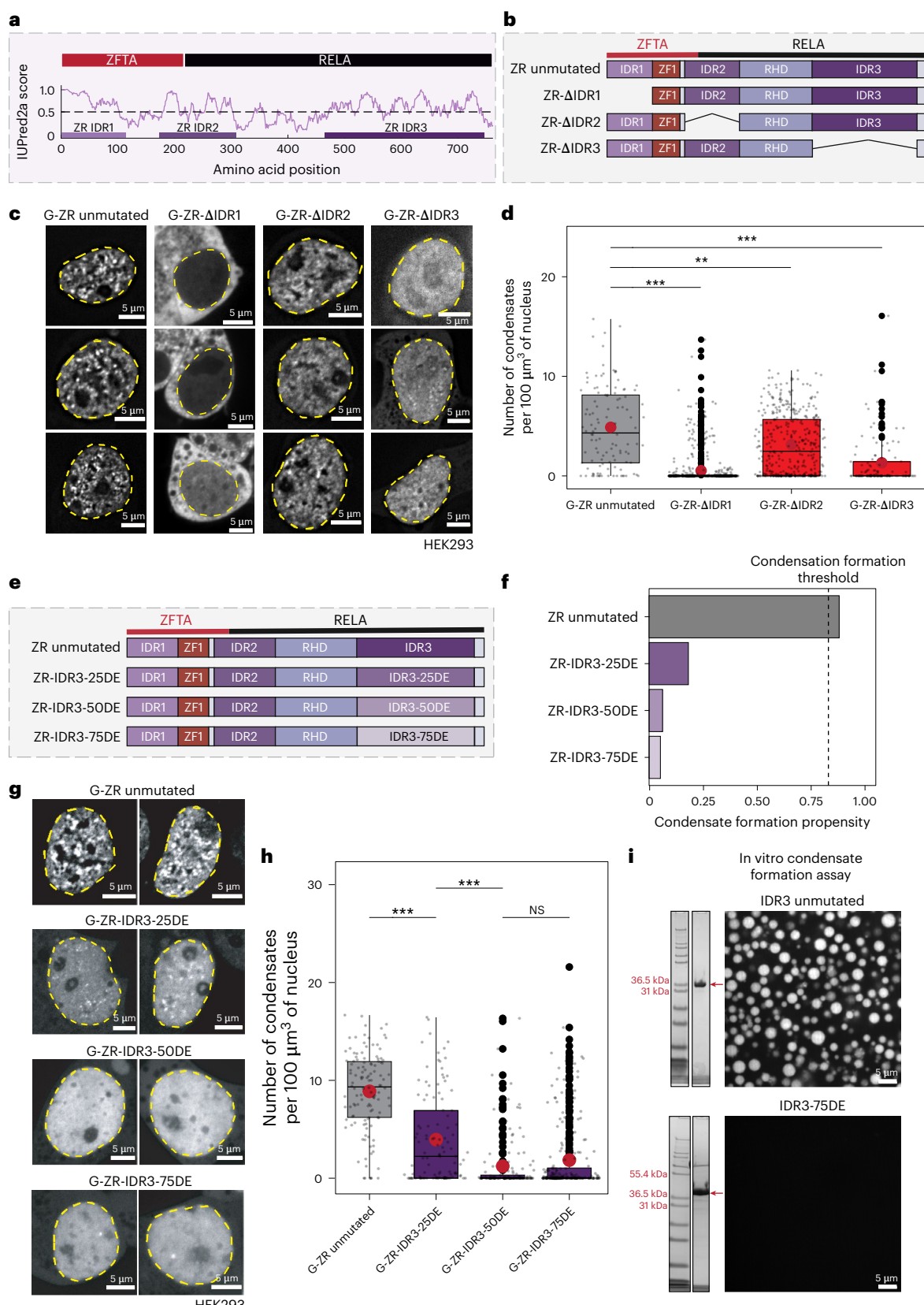

a broad range of cancers has shown that many FOs function through the formation of biomolecular condensates[11–17]. Condensates form through phase transitions driven by multivalent interactions between proteins and nucleic acids. Several FOs that form condensates contain intrinsically disordered regions (IDRs) that mediate both homotypic and heterotypic multivalent interactions[13,14,18–20]. We demonstrate

that one of three IDRs in ZR promotes aberrant condensate formation and governs oncogenic transcription programs. Furthermore, ZFTA variants involving fusions with other gene partners (that is, *MAML2* or *MAML3*) drive similar patterns of oncogene activation and have been shown to be sufficient for tumour formation[8]. We thus speculated that ZFTA FO-directed transformation involves ZFTA acquisition of a gene

**Fig. 3 | An IDR in RELA governs ZFTA–RELA condensate formation.**
**a**, Candidate IDRs within ZR and their location in the protein. **b**, A schematic of deletion mutants, each lacking one of the three predicted IDRs. **c**, Representative images of HEK293T cells expressing ZR unmutated or ZR-ΔIDR1–3. **d**, Quantification of the number of condensates in HEK293T cells expressing either unmutated ZR or ZR-ΔIDR1–3 mutants ($n$ = 73, 885, 238 and 68 independent cells for G-ZR unmutated, G-ZR-ΔIDR1, G-ZR-ΔIDR2 and G-ZR-ΔIDR3, respectively). Images from one imaging session were used for analysis, and each cell was considered a technical replicate. Two-sided analysis of variance (ANOVA) test. Box plots represent median and IQR. The whiskers extend to the smallest and largest values within 1.5× IQR. Outliers are shown with circles. Asterisks indicate the $P$ values (see below). **e**, A schematic of ZR unmutated and the IDR3 D/E substitution mutants (ZR-IDR3-25DE, ZR-IDR3-50DE and ZR-IDR3-75DE). **f**, Condensate formation probability scores of ZR unmutated, ZR-IDR3-25DE, ZR-IDR3-50DE and ZR-IDR3-75DE. **g**, Representative images of HEK293T cells expressing G-ZR unmutated or IDR3 D/E mutants (G-ZR-IDR3-25DE, G-ZR-IDR3-50DE and G-ZR-IDR3-75DE). **h**, Quantification of condensate numbers in HEK293T cells expressing either unmutated G-ZR or IDR3 D/E mutants (G-ZR-IDR3-25DE, G-ZR-IDR3-50DE and G-ZR-IDR3-75DE). $n$ = 135, 106, 155 and 245 independent cells for G-ZR unmutated, G-ZR-IDR3-25DE, G-ZR-IDR3-50DE and G-ZR-IDR3-75DE, respectively. Images from one imaging session were used for analysis, and each cell was considered a technical replicate. Two-sided ANOVA test. Box plots represent the median and IQR. The whiskers extend to the smallest and largest values within 1.5× IQR. Outliers are shown with circles. Asterisks indicate the $P$ values (see below). **i**, Representative images from in vitro condensate formation assays performed using purified IDR3 unmutated or IDR3-75DE. A Coomassie-stained protein gel image of each eluent is provided on the left side to indicate the purity of protein samples. Arrows point to the expected bands for each protein. NS, non-significant. *$P$ < 0.05, **$P$ < 0.01, ***$P$ < 0.001.

fusion partner capable of promoting condensate formation at oncogenic loci. This hypothesis is addressed by engineering synthetic *ZFTA* gene fusions, which leverage IDRs from known condensate-forming proteins, and testing their ability to support condensate formation, oncogenic transcription and tumorigenesis. Our study sheds light on the importance of condensate formation in ZR-driven oncogenesis and the relevance of IDR acquisition in specific gene fusions driving brain cancer.

## Results

### ZFTA–RELA forms dynamic nuclear condensates

To gain insights into the biophysical properties of ZR, we leveraged super-resolution microscopy to study its subcellular localization. The expression of monomeric enhanced green fluorescent protein (EGFP)-tagged constructs (termed 'G-' constructs) in human NSCs (hNSCs) revealed that full-length ZFTA (G-ZFTA) was largely restricted to the nucleolus, as indicated by colocalization with nucleophosmin (NPM1) (Fig. 1a). To eliminate the possibility of the EGFP tag leading to this localization pattern, it was replaced with a much smaller haemagglutinin (HA) tag. Immunostaining against the HA tag revealed a similar localization pattern for full-length ZFTA protein (Extended Data Fig. 1a). Expression of the ZFTA fragment (with one ZF domain, G-ZFTA-N), which is consistently retained in ZR gene fusions, resulted in diffuse nuclear localization (Fig. 1b). Full-length RELA (G-RELA), which is incorporated into ZR, was localized to the nucleus and cytosol (Fig. 1c). Conversely, expression of G-ZR led to the formation of nuclear puncta, reminiscent of biomolecular condensates (Fig. 1d).

To evaluate if G-ZR condensates exhibit internal dynamics, we performed fluorescence recovery after photobleaching (FRAP) experiments in hNSCs. G-ZR puncta showed liquid-like dynamics, as indicated by recovery of G-ZR fluorescence following photobleaching (Fig. 1e,f). Interestingly, ZFTA full-length protein also exhibited liquid-like dynamics in line with the liquid-like nature of the nucleolus[21] (Extended Data Fig. 1b,c). To further establish the liquid-like characteristics of ZR protein, we used lattice light-sheet microscopy to demonstrate that G-ZR condensates undergo fusion in three-dimensional (3D) time-lapse imaging (Fig. 1g and Supplementary Video 1). Finally, we validated our condensate findings in ZR-expressing human cell lines and in vivo using a genetic mouse model created by cortical implantation of NSCs stably expressing G-ZR (Fig. 1h,i and Extended Data Fig. 1d,e). Collectively, our data reveal that ZR forms dynamic nuclear assemblies that have properties similar to liquid-like condensates.

### ZFTA–RELA condensates colocalize with oncogenic targets

To determine if ZR condensates are associated with active gene transcription, we examined the colocalization of G-ZR with proteins known to be involved in active transcription (MED1, BRD4 and RNAPII) using immunofluorescence staining in hNSCs (Fig. 2a–f). We observed strong correlations between G-ZR in condensates and localization of MED1, BRD4 and RNAPII as compared with random nuclear loci (Fig. 2a–f and Extended Data Fig. 2a–f). To test whether G-ZR puncta were localized at highly transcribed ZR FO target genes, we performed nascent RNA fluorescence in situ hybridization (RNA-FISH) (Fig. 2g–n). Intronic RNA-FISH probes were designed against two highly transcribed ZR target genes, *NOTCH1* and *CCND1* (Fig. 2g,h and Extended Data Fig. 3a,b), marked by the presence of elevated H3K27ac chromatin immunoprecipitation (ChIP-seq) signal, seen across both mouse and human *ZFTA–RELA* EPN[22] (Fig. 2i,j). Analysis of multiple images (32 RNA-FISH loci for *NOTCH1* and 35 RNA-FISH loci for *CCND1*) revealed enrichment of ZR condensates near the centre of RNA-FISH loci (Fig. 2k–n and Extended Data Fig. 3c,d). Our findings demonstrate that ZR condensates are associated with active transcription and are enriched at key sites of ZR-driven oncogene expression.

**Fig. 4 | Disruption of ZFTA–RELA condensates abolishes chromatin occupancy, transcriptional activity and oncogenicity. a,b**, Heatmaps of CUT&RUN signal for ZR unmutated and ZR-ΔIDR3 centred around ZR peaks in HEK293T cells (**a**) and mNSCs (**b**). The heatmaps represent normalized read counts. **c**, Loss of IDR3 impairs the recruitment of co-activators and RNAPII in HEK293 cells. **d**, Global transcriptional changes following expression of unmutated ZR (left) or ZR-ΔIDR3 (right) (two-sided Wald test corrected for multiple testing using the Benjamini–Hochberg (BH) method). **e**, Transcriptional expression ($\log_2$-fold change (FC)) comparison between unmutated ZR and ZR-IDR3-25DE, ZR-IDR3-50DE or ZR-IDR3-75DE ($n$ = 8,910 genes; data are represented as a fitted regression line, with the shaded area denoting the 95% confidence interval). **f**, Read count comparison for condensate responsive genes between control, unmutated ZR and DE gradual mutants; $P$ < 2.22 × 10$^{-16}$ ($n$ = 6124 genes, two-sided Wilcoxon test). Box plots represent the median and IQR. The whiskers extend to the smallest and largest values within 1.5× IQR. **g**, Pathway enrichment analysis of condensate responsive genes (one-sided $P$ value for overrepresentation analysis is calculated by hypergeometric distribution with BH correction). **h**, Representative MRI images of mice intracranially injected with NSCs expressing ZR unmutated (top) or ZR-ΔIDR3. Tumour margins are indicated with red circles. **i**, Survival curve for mice intracranially injected with neuronal stem cells expressing ZR unmutated or ZR-ΔIDR3; $P$ = 0.0015 ($n$ = 15 mice per arm; Mantel–Cox test). **j**, Time to tumour formation for ZR unmutated and each gradual IDR3 D/E mutant. Tumour incidence is defined as the first detection of tumour on MRI scan (Mantel–Cox test). $N$ = 8, 13, 9 and 12 independent mice for ZR unmutated, ZR-IDR3-25DE, ZR-IDR3-50DE and ZR-IDR3-75DE, respectively. **k**, Representative MRI images of age-matched mice harbouring brain tumours generated with ZR unmutated or ZR-IDR3-25DE. Tumour margins are indicated with red circles. **l**, Tumour volume tracing using MRI images for tumours generated with ZR unmutated or ZR-IDR3-25DE. Data points show tumour volumes, starting at 5 weeks post birth ($n$ = 4 independent mice for ZR unmutated and $n$ = 7 independent mice for ZR-IDR3-25DE; data are represented as mean ± s.e.m).

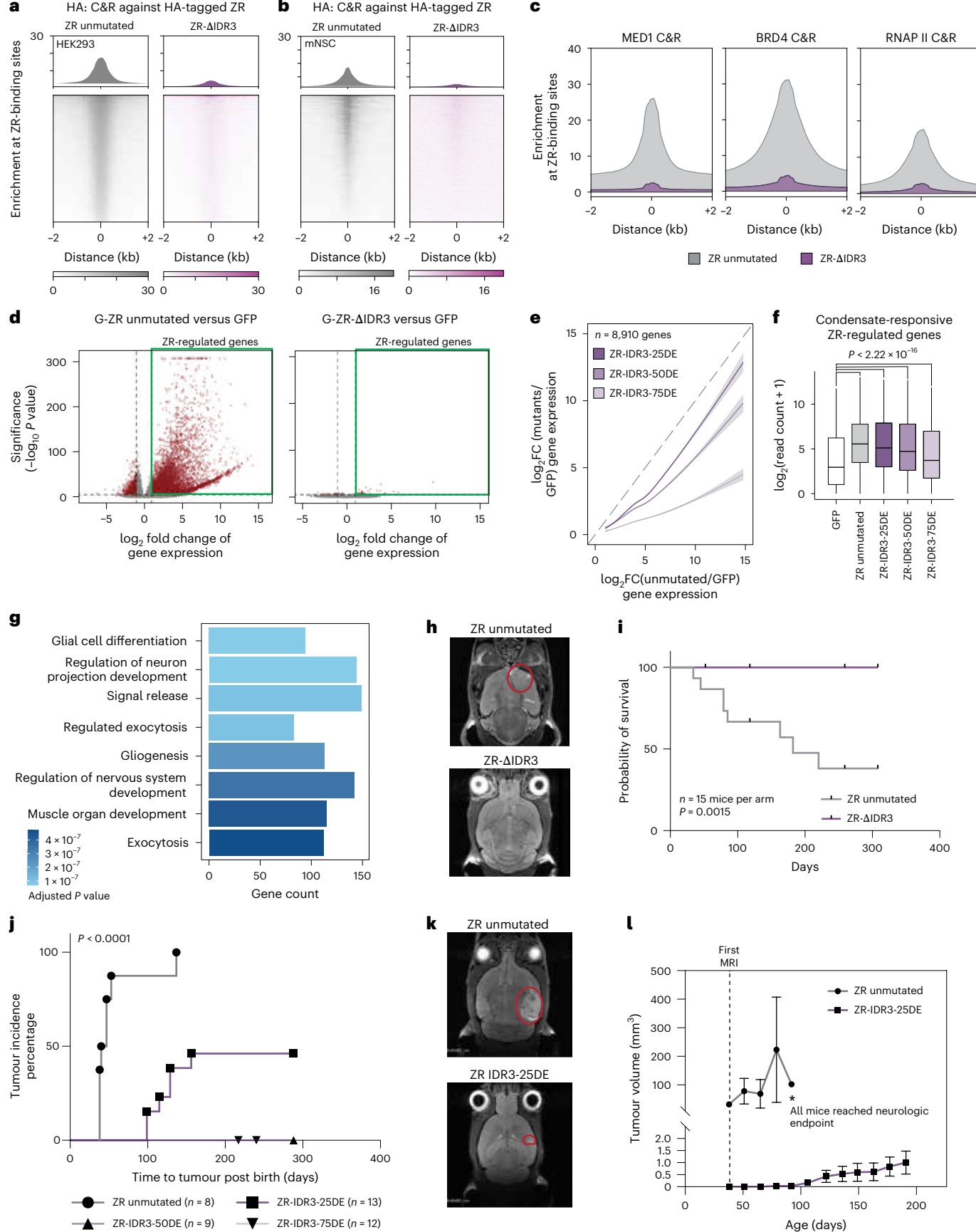

## ZFTA–RELA condensates are regulated by a disordered domain

Using our recently developed computational pipeline (https://sak.stjude.org/), we identified three IDRs within ZR. The first, IDR1, is found in the ZFTA portion of the FO. The second, IDR2, spans the breakpoint between ZFTA and RELA. The third, IDR3, is at the C terminus of the RELA protein, encompassing its transactivation domain (TAD) (Fig. 3a). To determine whether the IDRs in ZR were important for condensate formation, we generated a series of IDR deletion mutants (denoted ZR-ΔIDR1, ZR-ΔIDR2 or ZR-ΔIDR3) (Fig. 3b). All GFP-tagged IDR1–3 mutants, tested first in HEK293T cells, exhibited decreased puncta formation compared with the unmutated ZR. While G-ZR-ΔIDR1 showed disrupted nuclear localization, G-ZR-ΔIDR3 displayed diffuse nuclear localization, suggesting an important role for IDR3 in condensate formation (Fig. 3c,d). Although G-ZR-ΔIDR2 reduced the number of condensates, this effect was less pronounced compared with IDR3 deletion (Fig. 3d). G-ZR-ΔIDR3 was therefore prioritized, and its effects on condensates were validated in a hNSC model (Extended Data Fig. 4a).

We sought to define the amino acid enrichments within ZR-IDR3 necessary for condensate formation. To this end, we leveraged the SAK pipeline and an associated machine learning model (FO-Puncta machine learning (ML) model) that predicts condensation formation propensity for FOs based on physicochemical features derived from their amino acid sequences[23]. We found that IDR3 was enriched in the amino acids proline, alanine and glutamine, which are known to contribute to multivalent interactions underlying condensate formation by FOs[23] (Extended Data Fig. 4b). Because our previous work demonstrated that enrichment of negatively charged amino acids within FOs is anticorrelated with their ability to form condensates, we mutated 25%, 50% or 75% of the three enriched amino acids to negatively charged aspartic acid and glutamic acid (termed ZR-IDR3-25DE, ZR-IDR3-50DE and ZR-IDR3-75DE mutants; Fig. 3e and Supplementary Table 5). In silico analysis showed that ZR-IDR3-D/E mutants altered physicochemical feature values (Extended Data Fig. 4c) and reduced ML-generated condensation formation propensity values (Fig. 3f).

ZR-IDR3-D/E mutants were first evaluated in HEK293T cells, demonstrating that IDR3 mutants progressively altered condensate formation, with G-ZR-IDR3-50DE and G-ZR-IDR3-75DE mutants displaying diffuse nuclear localization (Fig. 3g,h and Extended Data Fig. 4d). To control for effects of introducing aspartic acid (D) and glutamic acid (E) residues on charge balance within ZR, we also created an orthogonal series of graded mutations using glycine (G) and serine (S) substitutions, denoted ZR-IDR3-25GS, ZR-IDR3-50GS and ZR-IDR3-75GS (Supplementary Table 5). Our rationale was that introducing glycine and serine mutations would preserve charge balance while still altering amino acids important for mediating multivalent interactions

underlying condensate formation. ZR-IDR3-GS mutants had a similar effect on reducing the number of ZR condensates formed when expressed in HEK293T cells (Extended Data Fig. 4e,f).

Finally, we sought to characterize the condensate formation propensity of IDR3 in condensate formation assays, in vitro. We found that purified unmutated IDR3 protein alone could form condensates through homotypic interactions (Fig. 3i, top). Conversely, at similar protein concentrations, IDR3-75DE lost phase separation capacity, while the IDR3-75GS mutant formed amorphous aggregates (Fig. 3i, bottom, and Extended Data Fig. 4g). Our findings demonstrate that IDRs within ZR contribute variably to condensate formation, with IDR3 playing the most pronounced role.

## ZFTA–RELA condensates govern oncogenic transformation

Given that deletion of IDR3 abrogated condensate formation by ZR, we next tested whether its loss would also result in loss of DNA binding, or alternatively, if the $C_2H_2$ ZF within ZFTA was solely responsible for DNA interaction. To our surprise, we found that the ZR-ΔIDR3 mutant bound the same DNA sites as unmutated ZR, albeit to a much reduced extent, based on CUT&RUN mapping in both HEK293T and mouse NSCs (Fig. 4a,b and Extended Data Fig. 5a). ZR-ΔIDR3 impaired the recruitment of key regulators of gene transcription, including MED1, BRD4 and RNAPII, to ZR target genes (Fig. 4c and Extended Data Fig. 5a). The ZR-ΔIDR3 deletion mutant markedly failed to activate aberrant gene transcription across the genome (Fig. 4d). Next, we evaluated the ZR-IDR3 condensation substitution mutants and observed a graded reduction in chromatin binding (Extended Data Fig. 5b) and gene activation upon increased extent of mutagenesis of amino acids involved in condensate formation (Fig. 4e–g and Extended Data Fig. 5c). Interestingly, among the genes with the strongest response to gradual condensate disruption (Extended Data Fig. 5c), we observed an enrichment of glial cell differentiation and neuro-transmission genes (Fig. 4g), consistent with prior studies on tumour–neuronal cell interactions in EPN[24].

Mouse NSCs stably expressing ZR-ΔIDR3 failed to form tumours when injected in mice as compared with cells stably expressing unmutated ZR that robustly formed tumours (Fig. 4h,i and Extended Data Fig. 5d,e). In vivo experiments were extended to the ZR-IDR3-DE graded mutants that were stably expressed during embryonic brain development using piggyBAC transposition and in utero electroporation (IUE)[4,8]. EPN developed with 100% penetrance in mice expressing unmutated ZR. By contrast, 50% of mice expressing the ZR-IDR3-25DE mutant developed tumours, while no tumours were observed in mice expressing ZR-IDR3-50DE or ZR-IDR3-75DE (Fig. 4j). Using longitudinal magnetic resonance imaging (MRI) tracing, we observed that ZR-IDR3-25DE tumours formed significantly later and exhibited slower-growing

---

**Fig. 5 | DNA binding is necessary to form ZFTA–RELA condensates that activate oncogenic transcription. a**, AlphaFold 2 model of ZF1 of ZFTA (obtained from AlphaFold Protein Structure Database for UniProtID C9JLR9). The backbone residues shown in orange or purple are shown to be either α-helical or in a β-strand, respectively, based on NMR chemical shift values and CSI 3.0. The position of the $Zn^{2+}$ ion (grey sphere) is inferred from homology to a canonical ZF structure (PDB code 1ZNF). The cysteine and histidine residues within the $C_2H_2$ core are shown as sticks. α-Helix 2 (α2) is indicated. **b**, AlphaFold 2 model of ZF1 superimposed onto the structure of a canonical ZF domain bound to DNA (PDB: 2DRP). Basic residues in α2 positioned to bind in the major groove of DNA are labelled and shown as sticks. Backbone amides that have been assigned using NMR in the absence of DNA are shown as spheres that are coloured on the basis of intensity changes in the presence of DNA ($1 - I/I_0$). The most significant intensity changes are clustered in α2. **c**, A schematic of DNA-binding mutants generated via amino acid substitution in the ZF1 domain (ZR-ZF2A) or RELA rel-homology domain (ZR-RL3A). **d**, Fluorescence polarization (FP) results showing that amino acid substitutions in ZF1 results in a concomitant decrease in binding DNA (data are represented as mean ± s.d; data are fit to a 1:1 binding

model). **e**, Representative images of HEK293T cells expressing G-ZR unmutated, G-ZR-RL3A or G-ZR-ZF2A. **f**, Quantification of condensate numbers in HEK293T cells expressing unmutated G-ZR, G-ZR-RL3A, or G-ZR-ZF2A ($n = 214$, 62 and 40 independent cells for G-ZR unmutated, G-ZR-RL3A and G-ZR-ZF2A, respectively). Images from one imaging session was used for analysis, and each cell was considered a technical replicate; two-sided ANOVA test. Box plots represent the median and IQR. The whiskers extend to the smallest and largest values within 1.5× IQR. Outliers are shown with circles; asterisks indicate the $P$ values (see below). **g**, Global transcriptional changes following expression of G-ZR unmutated or G-ZR-ZF2A (two-sided Wald test corrected for multiple testing using the BH method). **h**, Read count comparison between ZR unmutated and ZR-ZF2A for RELA-specific and ZFTA-specific subset of genes ($n = 1,089$ for RELA-specific genes and 8,434 for ZFTA-specific; two-sided Wilcoxon test). Box plots represent the median and IQR. The whiskers extend to the smallest and largest values within 1.5× IQR. Outliers are shown with circles. **i**, Survival curve for mice intracranially injected with NSCs expressing unmutated ZR or ZR-ZF2A ($n = 15$ for ZR unmutated and $n = 10$ for ZR-ZF2A; Mantel–Cox test). NS, non-significant. *$P < 0.05$, **$P < 0.01$, ***$P < 0.001$. DBD, DNA binding domain.

tumours compared with unmutated ZR tumours (Fig. 4k,l). We observed very similar patterns in the ZR-IDR3-GS mutants, with impaired oncogene activation, and loss of tumour initiation capacity in mice electroporated with ZR-IDR3-75GS as compared with unmutated ZR control (Extended Data Fig. 6a–d). Finally, to evaluate whether IDR3 was necessary and sufficient to initiate tumours when fused to ZFTA, we created a minimal construct that expressed ZFTA-N fragment joined to

IDR3, eliminating most of IDR2 and the RHD region of RELA (Extended Data Fig. 6e). This mutant, denoted ZR-IDR3-minimal, was capable of forming condensates, activating oncogenic ZR targets and initiating tumours when stably expressed using IUE (Extended Data Fig. 6f–h). Together, we demonstrate that IDR3 within ZR is crucial for neoplastic transformation and controls oncogenic expression programs through condensate formation.

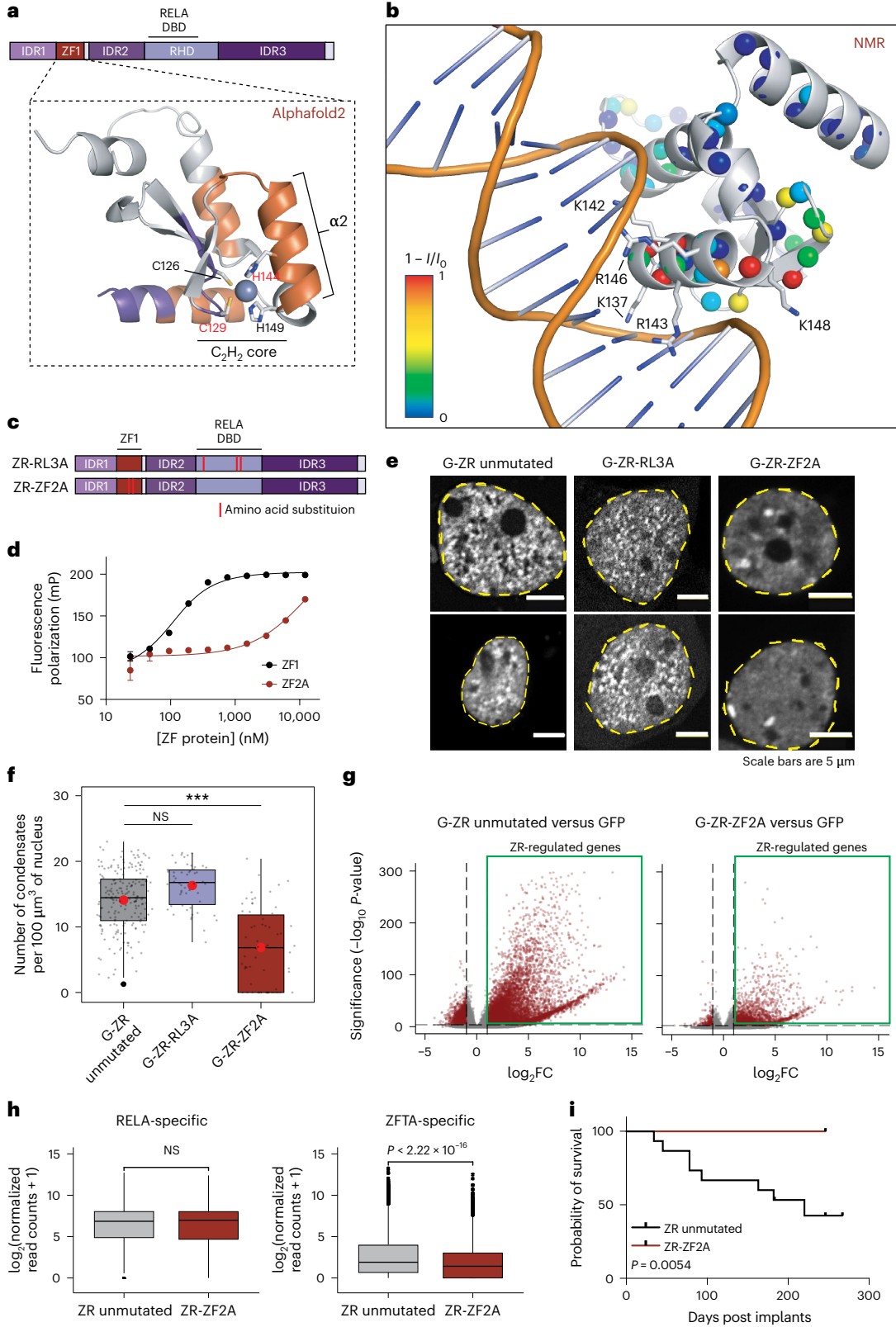

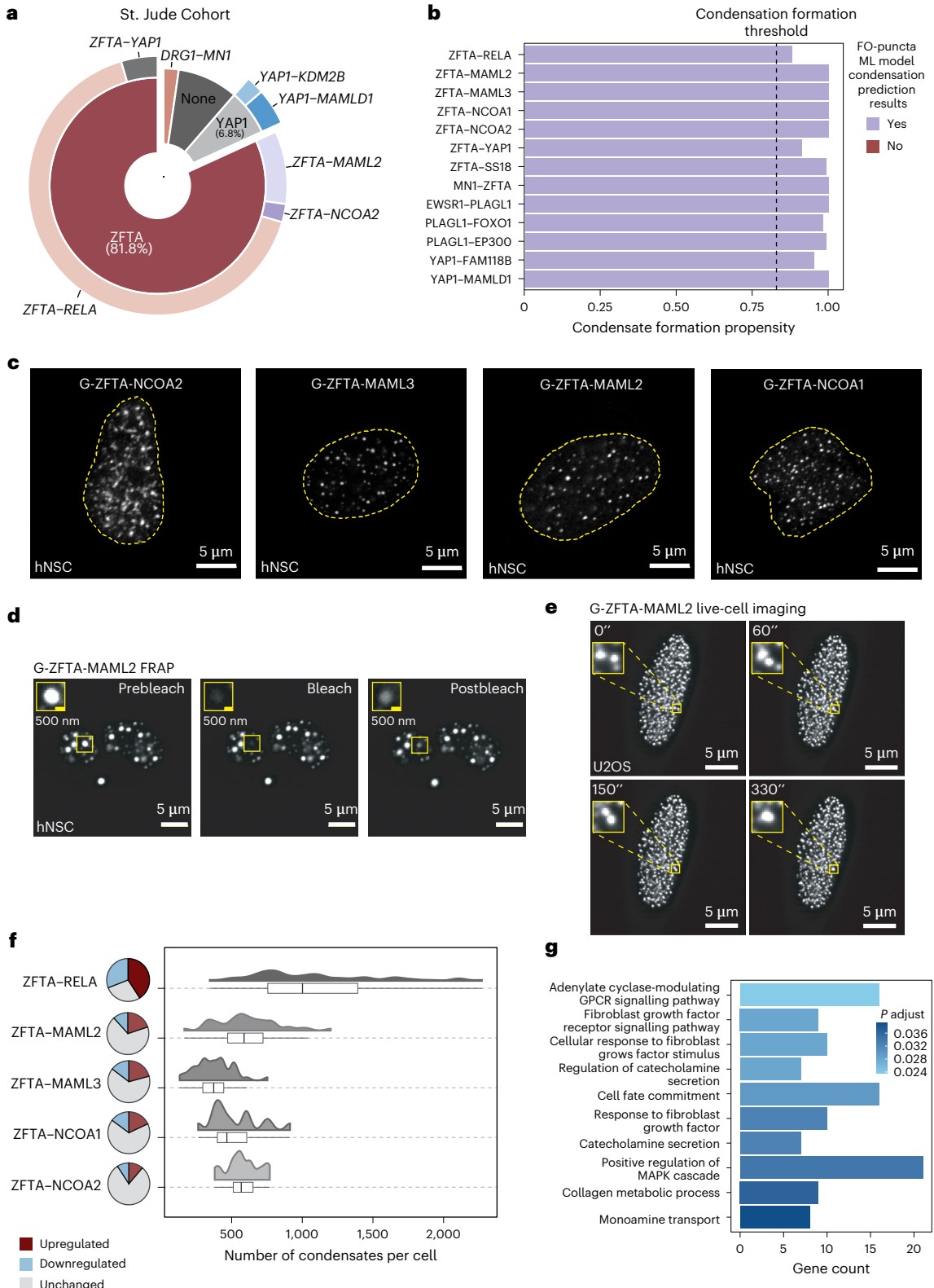

**Fig. 6 | ZFTA FOs acquire condensate propensity to facilitate neoplastic transformation. a**, The distribution of EPN fusions in St. Jude Children's Research hospital patient cohort from 2012 to 2024. **b**, Condensate formation propensity scores of EPN FOs. **c**, Representative images of hNSCs expressing various ZFTA FOs. **d**, Representative images of hNSCs expressing ZFTA–MAML2 before bleaching, at the time of bleaching and after bleaching. **e**, Live images capturing a merging event for two ZFTA–MAML2 condensates. **f**, Comparison between number of condensates per cell and number of upregulated genes between various ZFTA FOs (box plots represent the median and IQR; the whiskers extend to the smallest and largest values within 1.5× IQR). **g**, Pathway enrichment analysis of shared downstream target genes between ZFTA FOs (one-sided *P* value for overrepresentation analysis is calculated by hypergeometric distribution with BH correction).

## DNA binding is necessary to form ZFTA–RELA condensates

Unlike RELA[25], the mechanism of DNA binding by ZFTA protein is not fully defined. CUT&RUN mapping in HEK293T showed that ZR binds hundreds of genomic DNA sites, which is independent of the RELA portion of the fusion protein (Extended Data Fig. 7a). To understand how ZR binds DNA, we first used the cognate site identification (CSI) method[26] to identify the DNA sequences bound by ZR, which revealed the PlagI motif as the top binding sequence (Extended Data Fig. 7b). We next used fluorescence polarization to show that the one ZFTA ZF domain in ZR (denoted ZF1) bound specifically to a DNA oligonucleotide containing a single Plag transcription factor (TF) DNA motif (Extended Data Fig. 7c) and that a similar DNA oligonucleotide perturbed the two-dimensional $^1H$–$^{15}N$ heteronuclear single quantum (HSQC) spectrum of ZF1 (Extended Data Fig. 7d). To gain insight into the mechanism of DNA binding by ZF1, we performed structural analyses. We examined the model of the ZF1 structure deposited in the AlphaFoldDB (AF2) database (UniProt: C9JLR9)[27] (Fig. 5a), which revealed a classic $C_2H_2$ ZF domain[28] surrounded by additional α-helices, and is supported by nuclear magnetic resonance (NMR) spectroscopy data (Fig. 5b and Extended Data Fig. 7e). We docked the structure of ZF1 bound to DNA using the TTK/DNA structure (PDB: 2DRP), which positioned basic residues in ZF1 to interact with the major groove of B-form double-helical DNA (Fig. 5b). Based on these findings, we engineered point mutations within the $C_2H_2$ core of ZF1 in ZR to disrupt its structure and DNA binding activity (His144 and Cys129 to Ala, denoted ZR-ZF2A; Fig. 5c) and confirmed that the structure of ZF2A is disrupted by circular dichroism with a concomitant disruption in its ability to bind DNA (Fig. 5d and Extended Data Fig. 7f). To independently disrupt RELA-mediated DNA binding, we introduced previously established mutations within the RELA DNA-binding domain[25] (denoted ZR-RL3A) (Fig. 5c). When expressed in cells, the RELA DNA-binding mutations did not significantly affect the number of condensates. (Fig. 5e,f). By contrast, ZR-ZF2A strikingly decreased the number of condensates, concurrent with appearance of a small number of large foci, probably representing chromatin-dissociated condensates, as seen previously for chromatin-dissociated NUP98–HOXA9[14] (Fig. 5e,f and Extended Data Fig. 8a). Unlike ZF1 deletion mutants that abrogate nuclear localization[5], the ZR-ZF2A mutant was retained in the nucleus (Fig. 5e). Moreover, both ZR-RL3A and ZR-ZF2A retained their liquid-like dynamics, as indicated by FRAP (Extended Data Fig. 8b–e). As expected, the ZR-ZF2A mutant could not engage oncogenic target genes, such as *Notch1*, nor activate the ZR-specific gene expression program (Fig. 5g and Extended Data Fig. 7g). However, ZR-ZF2A retained its ability to activate the RELA-directed inflammatory program (Fig. 5h). ZR-RL3A activated a larger subset of genes, seen across HEK293 and mNSC cells (Extended Data Fig. 8f,g) that were enriched in ZR-specific genes but lacked RELA-directed inflammatory program activation. Finally, expression of ZR-ZF2A completely abolished tumour initiation capacity as compared with an unmutated ZR construct (Fig. 5i). These findings demonstrate that DNA binding mediated by the ZF1 ZF domain of ZR is necessary for condensate formation on chromatin, aberrant transcription of oncogenic target genes and tumorigenesis.

## Non-RELA ZFTA fusions share condensate propensity

The *ZFTA–RELA* translocation comprises 82% of cortex-derived EPN documented at St. Jude Children's Research Hospital from 2012 to 2024 (Fig. 6a). ZFTA fusions involving other partners or YAP1 fusions accounted for 18% of cases (termed 'other EPN FOs'). A meta-analysis of ZFTA fusion variants described in peer-reviewed literature reveals that ZFTA non-RELA fusion partners are often transcription factors or transcriptional co-activators (Supplemental Table 1). We hypothesized that oncogenic properties of ZFTA fusions require acquisition of a gene partner that displays one or more IDRs with physicochemical features associated with condensate formation. To address this hypothesis, we applied the FO-Puncta ML model[23] to predict the condensate formation probability of the other EPN FOs. Strikingly, all of these, based on their patterns of Shapley Additive Explanations (SHAP) and physicochemical feature values (Extended Data Fig. 9a,b; SHAP values indicate which features are most predictive[23]), were predicted to form condensates, including ZFTA–NCOA1, ZFTA–NCOA2, ZFTA–MAML2, ZFTA–MAML3 and ZFTA–YAP1 (Fig. 6b). This list was extended to include FOs characterized by *PLAGL1* fusions, histologically similar disease entities that target similar sites in the tumour epigenome, and *YAP1* fusions, which have a documented role in condensate formation[29] (Fig. 6b). To validate the machine learning-based modelling, we engineered GFP-tagged constructs for several ZFTA fusion variants and demonstrated that they all formed nuclear condensates with distinct punctate localization patterns (Fig. 6c and Supplementary Table 5). Like ZR, the ZFTA–MAML2 fusion variant exhibited dynamic behaviour in both FRAP and time-lapse lattice light-sheet microscopy experiments (Fig. 6d,e and Supplementary Video 2). Using RNA sequencing (RNA-seq), we demonstrated that all ZFTA fusion variants led to activation of a core oncogenic gene network (Fig. 6f). Interestingly, ZR, which displayed more condensates per cell, was associated with greater numbers of activated genes than the other variants (Fig. 6f). Shared activated genes from ZFTA FOs converged upon known ZR oncogenic programs but also included gene regulatory pathways such as catecholamine secretion and monoamine transport, which have been implicated in EPN progression[24] (Fig. 6g). These data suggest a model in which ZFTA gene fusions acquire an IDR partner prone to form condensates, which is necessary for effective chromatin binding, oncogene activation and tumour initiation.

## Synthetic ZFTA FOs drive oncogenesis

We next sought to test the hypothesis that condensate-prone IDR acquisition is a central component of ZFTA gene fusion-driven tumorigenesis. To this end, we identified IDR regions of known condensate-forming proteins, such as LAF1, FUS, DDX4, NUP98, EWS, NCOA2, SS18 and BRD4, and substituted these regions for IDR3 in ZR (Fig. 7a). Critically, most of these synthetic ZFTA FOs have never been observed before in biomedicine, such as those involving FUS and EWS (Fig. 7b), denoted ZFTA-sub-FUS:IDR and ZFTA-sub-EWS:IDR, respectively. While most synthetic ZFTA FOs led to aberrant cellular localization, protein aggregation or the absence of condensate formation, ZFTA-sub-FUS:IDR and

**Fig. 7 | Synthetic ZFTA FOs drive oncogenic transcription and tumorigenesis. a**, A schematic cartoon of the 'IDR swapping' experiment. Synthetic fusions were tested for condensate formation capacity, transcriptional activity and oncogenicity. **b**, A schematic representation of the synthetic fusions tested in this study. **c**, Representative images of HEK293T cells expressing indicated synthetic fusions. **d**, A read count comparison for ZR downstream target genes between ZFTA–RELA and synthetic fusions ($n = 7,716$ genes). Box plots represent the median and IQR. The whiskers extend to the smallest and largest values within 1.5× IQR. Outliers are shown with circles. $N = 3$ independent replicates. **e**, Representative MRI images of tumours generated using ZR-sub-FUS:IDR and ZR-sub-EWS:IDR synthetic fusions. **f**, Read count comparison for ZR EPN signature genes between healthy brain tissue and tumours generated using ZR-sub-EWS:IDR or ZR-sub-FUS:IDR. For each case, one brain tumour and matched normal brain tissue from the same mouse was used ($n = 93$ genes; two-sided Wilcoxon test). Box plots represent the median and IQR. The whiskers extend to the smallest and largest values within 1.5× IQR. Outliers are shown with circles. **g**, A schematic representation of ZR-IDR3 GS mutant and corresponding EWS rescue construct. **h**, Representative images of HEK293T cells expressing ZR-IDR3 mutant or the EWS rescue construct. **i**, Representative MRI image of brain tumor generated using ZR-IDR3-75GS EWS rescue construct. **j**, Read count comparison for ZR EPN signature genes between healthy brain tissue and tumours generated using EWS rescue construct ($n = 93$ genes, two-sided Wilcoxon test). Box plots represent the median and IQR. The whiskers extend to the smallest and largest values within 1.5× IQR. Outliers are shown with circles. One brain tumour and matched normal brain tissue from the same mouse was used. **k**, Proposed steps in ZFTA FO-driven EPN initiation. CSF, cerebral spinal fluid.

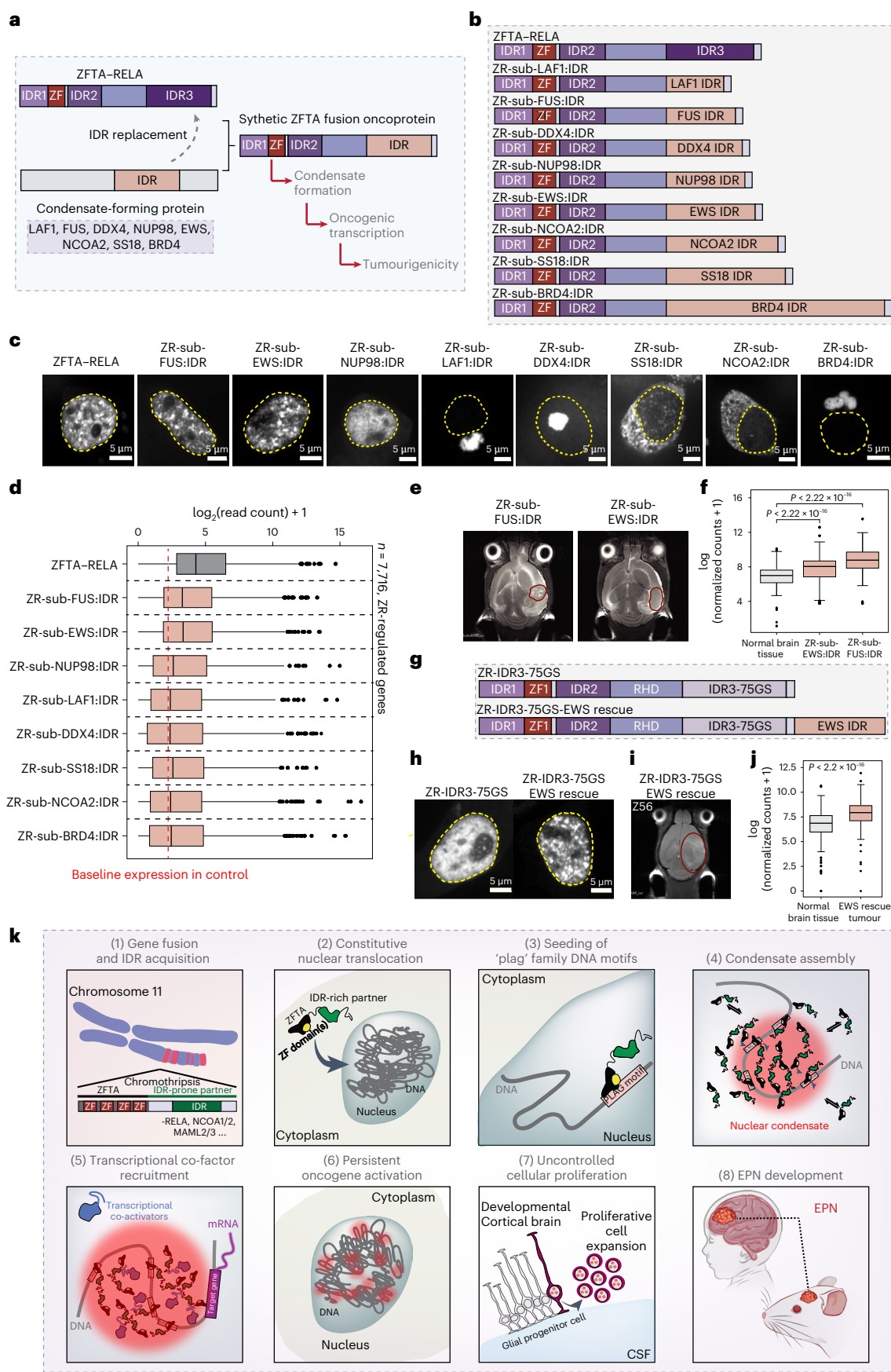

ZFTA-sub-EWS:IDR formed nuclear condensates like those seen in cells expressing unmutated ZR protein (Fig. 7c). Compared with defective ZR synthetic FOs, ZFTA-sub-EWS:IDR and ZFTA-sub-FUS:IDR also activated ZR transcriptional programs (Fig. 7d; n = 7,716, ZR regulated genes). We used an IUE model for brain tumour transformation to test the capacity of ZFTA-sub-FUS:IDR and ZFTA-sub-EWS:IDR to initiate tumours in mice[4,8]. Critically, both EWS and FUS synthetic ZFTA FOs were capable of driving tumorigenesis, albeit at a lower penetrance as compared with unmutated ZR (Fig. 7e and Extended Data Fig. 10a). RNA-seq profiling of ZR-sub-FUS:IDR and ZR-sub-EWS:IDR tumours revealed activation of ZR signature transcriptional programs (Fig. 7f and Extended Data Fig. 10b,c). To provide further support for a causal link between condensate formation and tumorigenicity, we sought to rescue the condensate formation capacity of ZR-IDR3 mutants. Addition of EWS:IDR to ZR-IDR3 mutants restored their ability to form nuclear condensates (Fig. 7g,h and Extended Data Fig. 10d,e). Importantly, these 'rescue' constructs were able to drive brain tumour formation with a transcriptional program similar to that of ZR EPN (Fig. 7i,j and Extended Data Fig. 10f,g). These gain-of-function experiments underscore the relevance of IDR acquisition to ZFTA-driven condensate formation, gene activation and oncogenesis, providing evidence of a causal link between aberrant condensate formation and cellular transformation.

## Discussion

Our findings support that condensate formation is the biophysical basis for activation of oncogene expression in ZFTA fusion-driven tumours. Condensate formation by ZR arises due to the juxtaposition of regions of the ZFTA and RELA proteins, which are normally localized in the nucleolus and throughout the cell, respectively, creating a FO with chimeric features that is constitutively localized in the nucleoplasm (Fig. 7k). Expression of ZR results in hundreds of nuclear puncta that exhibit liquid-like characteristics and colocalize with active transcriptional programs in EPN cells. Formation of transcriptional condensates by ZR is mediated predominantly by a disordered region we named IDR3. Of the three IDRs in ZR, deletion of IDR3 had the most pronounced effects on condensate formation. Multisite mutations of enriched amino acids in IDR3 altered condensate formation, impaired DNA binding, reduced recruitment of transcriptional activators, diminished activation of oncogenic programs and inhibited tumour development.

Disruption of ZR condensate formation through IDR3 deletion or mutation led to significant impairment of ZR DNA binding, despite an intact ZF DNA-binding domain. This leads us to hypothesize that cooperating regulatory proteins are important for stabilizing ZR condensates at oncogenic loci. Furthermore, our findings suggest that at least some FO molecules localize at oncogenic loci through IDR3-based multivalent interactions and not through DNA binding. This concept is supported by condensate assays of ZR-IDR3 revealing its propensity to form condensates through homotypic interactions. Our observations that transcriptional regulators (for example, BRD4, MED1 and RNA Pol II) colocalize with ZR within condensates suggest that heterotypic interactions are also involved in condensate formation. These findings have potentially important ramifications as understanding the molecular processes and proteins necessary for condensation of ZFTA FOs may yield important regulators of transcription as targets for therapy[30]. Indeed, we have shown that proteins such as BRD4 are recruited to ZR-bound loci, and EPN cells are sensitive to BET inhibitors[4]. Importantly, a rare subset of EPN (<6% of EPN cases) driven by YAP1 gene fusions, has been shown to also form condensates[31]. This suggests a convergent, yet molecularly distinct, mechanism of condensate formation by FOs in the pathobiology of EPN development[29].

If ZFTA FOs are oncogenic dependencies and represent direct therapeutic targets, we would reason that determining the 3D structural properties of the ZFTA FOs would help to inform their functional mechanisms as well as future drug discovery. Given that a key ZFTA $C_2H_2$ ZF (ZF1) is critical for DNA binding and ZR-driven oncogenic

transcription, we interrogated its structure using NMR. This structural analysis confirmed folding of ZF1 consistent with modelling using AlphaFold 2. Mutation of two key amino acids in the ZF1 domain of ZFTA (and not mutations in the RELA DNA-binding domain) displaced ZR from chromatin and converted many small, chromatin-associated condensates into few, large DNA-dissociated ones, supporting that chromatin-associated FO condensates are required for oncogenic function. These studies potentially support approaches to design DNA-binding inhibitors against $C_2H_2$ ZF domains found within the ZFTA FOs, as a possible direction for drug discovery against these oncogenic drivers.

EPN are highly resistant to a variety of chemotherapies, including carboplatin[2,3]. Transcriptional condensates have been shown to concentrate chemotherapy drugs such as platinum agents, resulting in heterogeneous distribution in the nucleus and reduced efficacy[32,33]. One extension of our findings is the hypothesis that ZFTA FO condensates in EPN cells may modify responses to conventional chemotherapy as a potential resistance mechanism to therapy. Understanding these processes may yield important insights on approaches to select, modify or combine chemotherapies with possible agents that disrupt condensate assembly or even concentrate therapeutic drugs within condensates[34,35].

The association between FOs, condensate formation and tumorigenesis is a recurring theme seen across several paediatric and adult cancers, such as those driven by NUP98–HOXA9[13,14] and EWS–FLI1[15,36,37] gene fusions and many others[23]. While IDR enrichment represents a recurring theme as a driver of condensate assembly, most studies have relied on loss-of-function approaches (that is, deletion mutagenesis) or purification of IDR regions and testing in in vitro systems. Our studies are among the first to employ gain-of-function strategies using synthetic FOs that probe the significance of IDR acquisition. Synthetic ZFTA FOs grafted with EWS or FUS IDRs, which have never been observed in biomedicine, lead to restoration of condensate formation, oncogenic transcription and brain tumour initiation. These findings support the general concept that many FOs acquire IDRs to promote condensation formation—a necessary process that governs oncogene activation and neoplastic transformation.

## Online content

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

A. Arabzade[1,2,3,4,17], H. K. Shirnekhi[5,6,17], S. Varadharajan [2,3,4,17], S. M. Ippagunta[2,3,4,17], A. H. Phillips[5,6], N. Laboe[2,3], D. W. Baggett[5,6], Wahiduzzaman[5,6], M. Jo[5,6], T. Zheng [2,3,4], R. Pathak[2,3,4], D. Gee[2,3,4], D. Bhimsaria[7], H. Wu[8], X. Gao[9], J. Liu[9], E. Emanus[2,3,4], A. Bland[2,3,4], A. Kardian[2,3,10], A. Hancock[2,3,4], B. Holcomb[2,3,4], T. Wright[2,3,4], T. Bugbee[2,3,4], H. Sun[2,3,4], M. Zhai [2,3,4], E. Caesar[2,3,4], M. Park[2,3,4], S. Tripathi[4,5], A. Shirinifard[2], K. Lowe[2], A. Khalighifar[2], R. A. Petersen [2], S. King[2], D. Stabley[2], A. Pitre[11], G. E. Campbell [11], C-G Park[5], W. T. Freyaldenhoven[5], B. Chandra[5], Y. Xia[5], E. Bonten[12], A. Achari[12], S. Kandikonda[12], A. Carisey [13], S. B. Pounds[8], J. Xu [9], D. W. Ellison[9], B. Deneen [10,14], K. C. Bertrand[3,15], R. W. Kriwacki [5,6,16] ✉ & S. C. Mack [2,3,4] ✉

[1]Department of Chemical and Biomolecular Engineering, Rice University, Houston, TX, USA. [2]Department of Developmental Neurobiology, St. Jude Children's Research Hospital, Memphis, TN, USA. [3]Neurobiology and Brain Tumor Program, St. Jude Children's Research Hospital, Memphis, TN, USA. [4]Center of Excellence in Neuro-Oncology Sciences, St. Jude Children's Research Hospital, Memphis, TN, USA. [5]Department of Structural Biology, St.

Jude Children's Research Hospital, Memphis, TN, USA. [6]Cancer Biology Program, St. Jude Children's Research Hospital, Memphis, TN, USA. [7]Indian Institute of Technology, Roorkee, India. [8]Department of Biostatistics, St. Jude Children's Research Hospital, Memphis, TN, USA. [9]Department of Pathology, St. Jude Children's Research Hospital, Memphis, TN, USA. [10]Cancer and Cell Biology Program, Baylor College of Medicine, Houston, TX, USA. [11]Cell and Tissue Imaging Center – Light Microscopy, St. Jude Children's Research Hospital, Memphis, TN, USA. [12]Department of Chemical Biology and Therapeutics, St. Jude Children's Research Hospital, Memphis, TN, USA. [13]Center for Pediatric Neurological Disease Research, St. Jude Children's Research Hospital, Memphis, TN, USA. [14]Center for Cancer Neuroscience, Baylor College of Medicine, Houston, TX, USA. [15]Department of Oncology, Division of Neuro-Oncology, St. Jude Children's Research Hospital, Memphis, TN, USA. [16]Department of Microbiology, Immunology and Biochemistry, University of Tennessee Health Sciences Center, Memphis, TN, USA. [17]These authors contributed equally: A. Arabzade, H. K. Shirnekhi, S. Varadharajan, S. M. Ippagunta. ✉e-mail: richard.kriwacki@stjude.org; stephen.mack@stjude.org

## Methods

### Mouse models

All cortical implants were done in NSG (NOD Scid gamma) mice (St. Jude Animal colony), and all IUEs were done in CD-1 mice (Charles River). Mice were housed in an American Association of Accreditation of Laboratory Animal Care-accredited facility. All animal studies were approved by the Animal Care and Use Committee and performed in accordance with best practices outlined by the NIH Office of Laboratory Animal Welfare. Animal protocol number 3124. All cortical implants were done in female NSG mice. All IUEs were performed in CD-1 mice, and both male and female offspring were analysed for survival curves.

### Cell culture

HEK293T (ATCC, CRL-3216) and U2OS (ATCC, HTB-96) cell lines were grown on tissue culture-treated plates using Dulbecco's modified Eagle medium (DMEM) supplemented with foetal bovine serum, sodium pyruvate, glutamine and primocin. hNSCs (ENSA, Milipore SCC003) and patient-derived xenograft cell lines were grown on Matrigel-coated plates using neural basal medium (Gibco 21103049) supplemented with sodium pyruvate, glutamine, B27, N2, bFGF (10 ng ml$^{-1}$) and rhEGF (20 ng ml$^{-1}$). All lines tested negative for *Mycoplasma* contamination, as assessed using a PCR-based approach. Cell lines were confirmed to be authentic and unique by short tandem repeat fingerprinting.

### DNA cloning

Full-length constructs and deletion mutants were generated using NEBuilder HiFi DNA Assembly (NEB E2621S). PCR reactions were performed using NEBNext High Fidelity 2x Mastermix (M0541S), and primers were designed in such a way that PCR fragments would have at least 20-bp overhangs. PCR reactions were gel purified using a Monarch DNA gel extraction kit (T1020S). DNA assembly reactions were performed for 1 h at 50 °C. Assembly reactions were transformed into chemically competent bacteria (NEB 5-alpha). DNA-binding-deficient mutants were generated using a QuickChange II site-directed mutagenesis kit (Agilent). Condensation-deficient mutant constructs were synthesized using gBlock Gene Fragments (Integrated DNA Technologies, IDT). Fragments and the destination vector (CL20) were cut with Not1 and Xba1 restriction enzymes and ligated together using a New England BioLabs' Quick Ligation Kit per the manufacturer instructions. All plasmid sequences were confirmed by whole-plasmid sequencing.

### Cellular transfection

Transfections were done using FuGENE HD transfection reagent (Promega, E2311) according to the manufacturer's instructions. In brief, for fixed cell experiments, cells were seeded to a density of 60–70% in a six-well plate on the day before transfection. For each transfection, 3 µg of endotoxin-free plasmid was mixed with 97 µl of Opti-MEM (Gibco 11058021), mixed and incubated at room temperature for 5 min. For live-cell confocal microscopy, cells were transfected in a 96-well plate with 100 ng of plasmid DNA in the CL20 vector backbone using FuGENE HD per the manufacturer's instructions. All FOs were transfected under identical conditions (for example, same amount of plasmid, same protocol and same incubation period).

### Cellular electroporation

Electroporations were performed using Mouse Neural Stem Cell Nucleofector kit (Lonza, VPG-1004). In brief, hNSCs were seeded 2 days before electroporation at roughly 40% confluency. On the day of electroporation, cells were treated with 10 µM Rock inhibitor (Y-27632 dihydrochloride, Tocris Bioscience) for 1 h. Five micrograms of endotoxin-free plasmid from a 2 µg µl$^{-1}$ solution were used in electroporation, and cells were supplemented with 1× CloneR2 (STEMCELL TECHNOLOGIES, 100-0691) after electroporation.

### Live-cell confocal imaging

All microscopy images were acquired on a 3i Marianas system configured with a Yokogawa CSU-W spinning disk confocal microscope utilizing a 100× Zeiss objective, 405-nm (Hoechst) and 488-nm (mEGFP) laser lines, and Slidebook (RRID:SCR_014300) 6.0 (3i). Three-dimensional images of cells were captured as z stacks with 0.2-mm spacing between planes, spanning 12.2 mm in total. Live cells were imaged at 37 °C in phenol red-free DMEM with high glucose (Gibco) supplemented with 1× penicillin–streptomycin, 10% foetal bovine serum, 4 mM L-glutamine and 25 mM HEPES.

### Lattice light-sheet live-cell imaging

To acquire super-resolution live-cell images a lattice light-sheet microscope (3i) with Slidebook version 2023.0 was used. The sample was illuminated using a multi-Bessel beam interference pattern (crop factor[38], $\varepsilon = -0.15$) filtered through an annular mask with inner and outer numerical apertures (NAs) of 0.472 and 0.55, respectively. This configuration produced a light sheet with an approximate propagation length of 20 µm and an axial thickness of 1.05 µm at 488 nm. Propagation length was determined through the excitation of fluorescein dye at 448 nm, imaging the single-Bessel illumination pattern emitted and measuring the full width at half maximum (FWHM) of the intensity profile along the direction of propagation at the maximum intensity. Axial thickness was determined through the measured $XZ$ point spread function (PSF) by scanning a fluorescent 100-nm microbead through the illumination pattern and measuring the FWHM of the profile along the axial direction at the pattern maximum. The theoretical values of propagation length and axial thickness for these settings are 22.5 µm and 1.0 µm at 488 nm (ref. [39]). Sample emissions were detected with a Zeiss 1.1 NA, 20× water immersion detection objective. Volume images were acquired in sample scanning mode with a 0.3-µm step size, 20-ms exposure time, 12% laser power and a 30-s interval between scans over a total duration of 30 min. Raw images were resampled to remove the skew orientation introduced by sample scanning and deconvolved using a standard Lucy–Richardson algorithm with acquired lattice light sheet (LLS) (PSFs). Videos were generated using ParaView[40]. To acquire super-resolution live-cell images, a lattice light-sheet microscope (3i) was used. The sample was illuminated by a multi-Bessel beam interference pattern (crop factor[38], $\varepsilon = -0.15$) filtered through an annular mask with inner and outer NAs of 0.472 and 0.55, respectively. This produced a light sheet with an approximate propagation length of 20 µm and an axial thickness of 1.05 µm at 488 nm. Propagation length was determined through the excitation of fluorescein dye at 448 nm, imaging the single-Bessel illumination pattern emitted and measuring the FWHM of the intensity profile along the direction of propagation at the maximum intensity. Axial thickness was determined through the measured $XZ$ PSF by scanning a fluorescent 100-nm microbead through the illumination pattern and measuring the FWHM of the profile along the axial direction at the pattern maximum. The theoretical values of propagation length and axial thickness for these settings are 22.5 µm and 1.0 µm at 488 nm[39]. Sample emissions were detected with a Zeiss 1.1 NA, 20× water immersion detection objective. Volume images were acquired in sample scanning mode with a 0.3-µm step interval, 20-ms exposure time, 12% laser power and a 30-s time interval between scans over a total duration of 30 min. Raw images were resampled to remove the skew orientation introduced by sample scanning and deconvolved using a standard Lucy–Richardson algorithm with acquired LLS PSFs. Videos were generated using ParaView[40].

### FRAP

hNSCs were electroporated as described previously and plated on two-well chambered coverslips (ibidi, 80286). FRAP time lapses were acquired on a Marianas system (Intelligent Imaging Innovations) configured with a CSU-W1 (Yokogawa) spinning disk on a Zeis Axio Observer (Zeiss) microscope. Images were collected with an alpha Plan

Apo 100×/1.46 NA objective (Zeiss) through a ZT405/488/561/640tpc (Chroma) dichroic and FF01-525/30 (Semrock) emission filter onto a Prime 95B camera (Teledyne Photometrics), resulting in 110-nm pixels. Illumination was provided by a 488-nm laser going through either the Vector (Intelligent Imaging Innovations) for photobleaching or CSU-W1 for image acquisition. Acquisition was controlled using Slidebook 6 (Intelligent Imaging Innovations). Bleaching was performed over a region with 1-µm radius using 20% laser power and the images were acquired every 1 s. FIJI was used to measure the average fluorescence intensity of the bleached area, whole nucleus and background, before and after bleaching. For each timepoint, background values were subtracted from bleached and whole-nucleus values. Intensity values for the bleached area were normalized twice, first to the whole-nucleus intensity and then to the prebleach values. Postbleach FRAP data were averaged over 19 cells.

### Immunofluorescence staining
hNSCs were electroporated with an eGFP-tagged ZFTA–RELA construct as described previously and plated on Matrigel-coated number 1.5 coverslips. After 24 h, cells were washed with phosphate-buffered saline (PBS) and fixed with 4% paraformaldehyde for 10 min at room temperature. Cells were then washed with PBS, three times for 5 min. Cells were then blocked with 3% goat serum in antibody dilution buffer for 1 h at room temperature. Coverslips were then incubated with primary antibodies (Supplementary Table 2) in a humidified chamber overnight at 4 °C. Cells were washed with PBST (PBS buffer containing 0.1% Tween20) three times for 5 min. Coverslips were then incubated with secondary antibodies with goat origin for 1 h at room temperature in a humidified chamber. After 1 h, cells were washed with PBST three times for 5 min. DAPI (1 µg ml⁻¹) was then used to stain the nucleus. After a quick rinse with PBST to remove excess DAPI, coverslips were gently dipped in distilled water to remove residual salts, air dried slightly and then mounted on a slide using diamond prolong mounting medium (Invitrogen P36965). Slides were kept at room temperature and protected from light for at least 48 h before imaging.

### Fluorescence slide scanning
Fluorescence slide scanning was performed using a Zeiss Axio Scan. Z1 with a Hamamatsu ORCA-Flash4.0 V3 camera using Zeiss ZEN 3.1 software. Images were created with a Zeiss Plan-Apochromat 20×/0.8 objective lens with illumination by Zeiss Colibri.2 light-emitting diodes (365 nm, 470 nm and 625 nm) and corresponding filters (Zeiss Filter Set 49, 38 HE and 50, respectively).

### RNA-FISH
hNSCs were electroporated with an EGFP-tagged ZRfus1 construct as described previously. Subsequent steps were performed in a PCR hood and all the equipment were treated with RNase away (Invitrogen 10328011). Cells were fixed in 4% paraformaldehyde made with RNase-free PBS. Cells were washed in RNase-free PBS, three times for 5 min. Cells were permeabilized for 30 min with 0.5% Triton-X100 in RNase-free PBS. Cells were washed twice with RNase-free PBS and then once with wash buffer A (20% Stellaris Wash Buffer A, SMF-WA1-60, 10% deionized formamide, in RNase-free water) for 5 min at room temperature. Stellaris RNA-FISH probes designed to hybridize with the intronic regions of genes (Supplementary Table 3) were diluted in the hybridization buffer (90% Stellaris RNA FISH Hybridization Buffer, SMF-HB1-10, and 10% deionized formamide) to a final concentration of 125 nM. Cells were hybridized with probes overnight at 37 °C in a humidified chamber protected from light. Cells were then washed in wash buffer A for 30 min at 37 °C. Nuclei were counterstained with 1 µg ml⁻¹ DAPI in wash buffer A for 30 min at 37 °C. Cells were then washed with Stellaris Wash Buffer B (SMF-WB1-20) for 5 min at room temperature. Cover glasses were then dipped in water to remove salt residues, slightly dried and mounted on a microscope slide using

Prolong Diamond mounting medium (Invitrogen P36965). Slides were kept at room temperature for at least 2 days before imaging on a 980 AiryScan super-resolution microscope.

### Condensate identification and colocalization analysis
Fiji[41] was used to split the images into different channels (ZR, staining target and DAPI). To identify the ZR condensates, ZR-channel images were normalized on a 16-bit scale. ilastik, a pixel classification software based on machine learning algorithms, was used to identify puncta and generate mask images for them. Using Python and mask images, the centre of each condensate and the size of the bounding box (to understand the size of each condensate on the $z$ axis) were calculated. Assuming a 61-pixel-by-61-pixel box around the centre of each condensate, the intensity values were gathered from the ZR channel and staining target channel. Pixel intensity values were normalized to fall between 0 and 1. These values were then averaged on the $z$ axis according to the bounding box size. These values were then averaged over all the identified condensates, and the final matrix was used to generate heatmaps.

To pick a random location, the DAPI channel was used to identify the boundaries of the nucleus. An equal number of unique random locations were selected for each cell as the number of puncta found, ensuring that a 61-by-61-pixel area around each location fell entirely within the nucleus. For the depth in the $z$ direction, a random value was selected between the smallest and largest condensate sizes identified for each cell. Pixel intensity values in staining target channel were gathered in each box, normalized to 0 to 1 and averaged over the $z$ direction. These values were then averaged over all identified puncta, and the final matrix was used to generate heatmaps. A similar approach was used to analyse the RNA-FISH images, with the difference that the 61-by-61-pixel box was centred on the RNA-FISH locus.

To compare the number of condensates across various mutants in the study, puncta density was calculated for each cell as the number of puncta per 100 µm³, using the number of puncta per cell and the cell volume determined by PunctaTools. Cells were excluded from analysis if the cell volume was less than 200 µm³, suggesting a mis-segmented or out-of-frame cell. Cells were also excluded from analysis if their average intensity was in the bottom 1/3 of expressing cells for that condition. Low-expressing cells were excluded because adjacently expressing cells resulted in diffuse fluorescent signals that caused false-positive segmentation in low-expressing cells. To access the codes and images, please refer to BioImages accession number S-BIAD1556.

### RNA extraction
HEK293T cells were transfected using FuGene HD as described previously. Two days after transfection, GFP-positive cells were sorted and used for RNA extraction. Subsequent steps were performed in a benchtop PCR hood, and all the equipment was treated with RNase Away. RNA was extracted using miRNeasy Cell/Tissue Advanced Kit (Qiagen, 217684) following the manufacturer's recommendations.

### RNA-seq analysis
Sequencing reads were adapter trimmed using Trimgalore and aligned to hg38 (for H293T data) or mm10 (mNSC data) using STAR (v). The read counts for Refseq genes were quantified using htseq-count. Differential expression analysis was performed using DEseq2. Genes with low expression, defined as having a total count across all samples below 10, were filtered out from the input matrix before DESeq2 analysis.

### Cut&Run
Cut&Run experiments were performed using Epicypher CATANA kit (14-1048). In brief, for each reaction, 500,000 cells were pelleted and washed twice with room temperature wash buffer. Cells were then incubated with activated ConA beads at room temperature for 10 min. Cell–bead complexes were incubated overnight at 4 °C. The next day, after three washes with cell permeabilization buffer, cell–bead complexes

were incubated with pAG-MNase enzyme at room temperature for 10 min. After another three rounds of wash with cell permeabilization buffer, ice-cold calcium chloride was added to the cell–bead–enzyme complex and incubated for 2 h at 4 °C to complete the digestion step. Digestion was then quenched using the stop buffer, and DNA fragments were purified using buffers included in the kit.

## Cut&Run data analysis

Paired-end reads were adapter and quality trimmed using Trimgalore (v0.6.5, default parameters, http://www.bioinformatics.babraham. ac.uk/projects/trim_galore/; RRID:SCR_011847) and aligned to mouse genome mm10 using Bowtie2 (v2.5.3, parameters: --local -D 20 -R 3 -N 0 -L 20 -i S,1,0.50 --no-unal --no-mixed --no-discordant --phred33 -I 10 -X 700; RRID:SCR_016368). Duplicated reads were then marked and removed using picard MarkDuplicates (v2.21.1; http://broadinstitute. github.io/picard/) and Samtools (v1.19.2; RRID:SCR_002105), respectively, with default parameters. Deeptools (v3.5.4; RRID:SCR_016366) was used to convert all the resulting BAM files to Bigwig format for visualization. MACS2 (v2.2.7.1) was used to call peaks on the resulting BAM files, applying a $P$ value threshold of $1 \times 10^{-3}$.

## Western blot

Western blot analysis was performed using standard techniques. In brief, cells were lysed using RIPA buffer (Thermo Fisher, 89900) supplemented with protease inhibitor cocktail (Cell Signaling Technologies, 5871S). Lysates were separated by 4–20% Novex Tris-glycine gel (Thermo Fisher, XP04200BOX). Proteins were transferred overnight. Blots were blocked using 1% casein solution (Bio-Rad, 1610783). Primary antibody incubation was performed overnight at 4 °C (anti-HA tag antibody: abcam ab9110 1:4,000 dilution, anti-Bactin antibody: Cell Signaling Technologies 4967 1:5,000 dilution and anti-p65 antibody: abcam ab16502 1:1,000 dilution). Blots were visualized using secondary antibodies, incubated at room temperature for 1 h (IRDye goat anti-mouse or rabbit IgG secondary, Licor)

## Stable cell line generation

Viral particles were generated using helper plasmids and following common transfection protocols. Mouse NSCs were infected, and the positive population was sorted based on their GFP signal. Expression was validated by performing western blotting following common protocols.

## Mouse orthotopic intracranial injections

All animal experiments were conducted with approval from St. Jude Children's Research IACUC committee. On the day of injection, cells were taken off the Matrigel plates using Acutase. Cells were counted, and 1 million cells per injection were spun down and resuspended in a solution of 1:1 PBS and Matrigel and kept on ice. The resuspension volume was set at 5 µl per injection. Mice were anaesthetized with a ketamine–xylazine cocktail (100 mg kg⁻¹ and 10 mg kg⁻¹, respectively). Following the loss of pedal reflex, scale hair was removed, and the surgery area was aseptically prepped. Anaesthetized animals were placed in a stereotaxic apparatus, after which the scalp and dura were removed to create a cortical window exposing the brain surface. Mice were then injected with cells, using 5 µl of cell suspension per injection. After implantation, the wound was sealed with wound clips.

## CSI DNA-binding assay

HEK293T cells were transiently transfected with HA-tagged ZFTA–RELA fusion plasmids using Lipofectamine 2000 reagent according to the protocol (Thermo Fisher 11668019). Following expression, cells were lysed with RIPA buffer (Thermo Fisher 89900) and spun down to collect supernatant. A DNA library (IDT) containing randomized central regions of 20 bp flanked by constant sequences complementary to primers was converted to dsDNA and brought to 74 ng µl⁻¹ before being combined 1% w/v BSA, 500 ng µl⁻¹ poly dI-DC (Thermo Fisher 20148E),

1% NP-40 (Thermo Fisher 85124) and 10× PBS. ZFTA–RELA-positive and ZFTA–RELA-negative cell lysates were incubated with this mixture for 1 h at room temperature. The mixture was then added to anti-HA beads (Thermo Fisher 88836) washed in binding buffer (10× PBS, 1% BSA and 1% NP-40,) and incubated for 30 min. Solutions were washed in binding buffer and aspirated on a magnetic plate three times before being resuspended in a PCR master mix (Lucigen Econo Taq 2X 30035-1 and custom primers). Library fragments attached to beads were amplified on a Bio-Rad thermal cycler and then purified using the New England Biolabs Monarch PCR & DNA clean-up kit (T1030L). Eluted DNA library fragments from each sample were diluted to a concentration of 74 ng µl⁻¹ and checked on a gel before being incubated with cell lysates again. After three rounds of this incubation and amplification, all purified library fragments from each sample were given a unique barcode and a sequencing adapter and then sequenced on a NovaSeq short-read sequencing amplicon kit, yielding ~500 million reads. Sequencing results were sorted by barcode, and the 20-bp library regions selected in each sample were ranked by enrichment and normalized to fusion-negative lysate samples.

## Expression and purification of ZF1 from *E. coli*

Synthetic DNA (IDT) encoding residues 81–172 of the wild-type ZFTA ZF domain 1 (ZF1) and the mutant ZFTA ZF domain 1 (C129A, H144A) (ZF2A) were subcloned into a modified pET28 vector that contains an N-terminal His6-MBP tag followed by a TEV protease cleavage site and pCoofy4 (Addgene #43986), respectively. Both proteins were expressed in *Escherichia coli* strain BL21(DE3) (New England Biolabs). Cells were cultured in lysogeny broth (LB) medium or M9 minimal medium in the presence of ¹⁵N ammonium chloride (1 g l⁻¹; Cambridge Isotope Laboratories) and unlabelled or ¹³C D-glucose (4 g l⁻¹; Cambridge Isotope Laboratories) and induced with 0.5 mM isopropyl β-D-1-thiogalactopyranoside (IPTG) and 50 mM zinc acetate at 18 °C for 16 h. *E. coli* pellets were resuspended in lysis buffer (50 mM Tris pH 7.5, 250 mM NaCl and 0.5 mM tris(2-carboxyethyl)phosphine TCEP) with EDTA-free protease inhibitors (Sigma Aldrich). The MBP–ZF1 fusions were purified via Ni-NTA agarose resin (Goldbio). MBP–ZF1 and MBP–ZF2A were dialysed against 25 mM citrate, pH 5.6, 1 M NaCl, 0.5 mM TCEP for 4 h before removing the MBP tag with uTEV3 or HRV3C protease, respectively. Both proteases were purified in house after expression in *E. coli* strain BL21(DE3) (New England Biolabs) using plasmids obtained from Addgene. ZF1 proteins were separated from the MBP tag using preparative gel filtration in 25 mM citrate, pH 5.6, 1 M NaCl, 0.5 mM TCEP (GE Healthcare, HiLoad 16/600 Superdex 75 pg). The ZF2A mutant exhibited poor stability after removal of the MBP tag, so experiments comparing MBP–ZF1 and MBP–ZF2A were conducted using material purified as described above via Ni-NTA agarose (Goldbio) and dialysed against 25 mM citrate, pH 5.6, 150 mM NaCl, 0.5 mM TCEP. All protein concentrations were determined by ultraviolet absorbance.

## Fluorescence polarization DNA-binding experiments

The Cy5 dye-labelled, double-stranded DNA (top sequence, 5′-Cy5-GATCCATGGCCCCATGGATG-3′) containing the CSI-identified ZFTA ZF1 DNA-binding sequence (5′-GGCCCC-3′) was commercially synthesized by IDT. A scrambled Cy5 dye-labelled, double-stranded DNA (top sequence, 5′-Cy5-ACACTCGGACTGGTTCGAGC-3′) was used as a negative control. Double-stranded DNA was prepared by annealing the Cy5-labelled strands with their reverse complements with slow cooling from 98 °C to room temperature in a 1 l water bath. Protein–DNA binding reactions were assembled by mixing Cy5 dye-labelled, double-stranded DNA with an equal volume of the ZFTA ZF1, MBP–ZF1 or MBP–ZF2A serial dilutions, yielding a final DNA concentration of 100 nM and final ZFTA ZF1 concentrations ranging from 7.5 µM to 15 nM in 25 mM citrate buffer (pH 5.6) containing 150 mM NaCl and 0.5 mM TCEP. Reactions were incubated for 20 min at room temperature to reach binding equilibrium. Fluorescence polarization values

were measured using a CLARIOstar microplate reader with excitation at 590 nm and emission measured at 675 nm. Experiments were performed in triplicate. Binding isotherms were fit to a 1:1 binding model using GraphPad Prism (Dotmatics).

## NMR spectroscopy and secondary structure analysis of AF2 model of ZFTA ZF1

Three-dimensional HNCA[42], HNCACB[43] and CBCA(CO)NH[44] NMR spectra were acquired for ZF1 (380 μM $^{13}$C/$^{15}$N-labelled ZF1 in 25 mM citrate, 150 mM NaCl, 0.5 mM TCEP, pH 5.6, and 5% $D_2O$ (NMR buffer) at 25 °C) using a Bruker 700 MHz spectrometer equipped with a TCI CryoProbe. The time domain sizes (in numbers of data points) of the three NMR dimensions ($^1$H, $^{13}$C and $^{15}$N) were 2,048, 48, 128; 2,048, 48, 96; and 2,048, 48, 128; respectively, and the total acquisition time was 3 days and 15 h. The data were processed using nmrPipe[45], and zero filling, linear prediction and cosine square window function were applied for all dimensions. Peak picking was performed with Sparky[46], and the peak lists were sent to the I-PINE web server[47] for backbone resonance assignments followed by manual inspection to verify and extend the assignments. The chemical shifts of $^1$H, $^{15}$N$_H$, $^{13}$C$_{alpha}$ and $^{13}$C$_{beta}$ resonances for 75 of 96 total residues in ZFTA were assigned. The chemical shift assignments were used to generate secondary structure predictions for the assigned resonances using CSI 3.0[48]. Secondary structure elements in the AF2 model of ZF1 were assigned using STRIDE[49]. Double-stranded DNA with the sequence 5'-CGGCCCCG-3' (IDT) was prepared with slow annealing as described above and added to a sample of $^{13}$C/$^{15}$N-ZF1 at 20% stoichiometric excess and concentrated with a 3 kDa molecular weight cut-off centrifugal filtration device (Millipore) to a final protein concentration of 140 μM. Resonance shifts and intensity changes in the HSQC spectrum of the DNA:$^{15}$N-ZF1 were followed by inspection and by measuring changes in resonance intensities at the chemical shifts observed in the apo spectrum. 3D HNCA[42], HNCACB[43] and CBCA(CO)NH[44] NMR spectra were acquired for ZF1 (380 mM $^{13}$C/$^{15}$N-labelled ZF1 in 25 mM citrate, 150 mM NaCl, 0.5 mM TCEP, pH 5.6, and 5% $D_2O$ (NMR buffer) at 25 °C) using a Bruker 700 MHz spectrometer equipped with a TCI CryoProbe. The time domain sizes (in numbers of data points) of the three NMR dimensions ($^1$H, $^{13}$C and $^{15}$N) were 2,048, 48, 128; 2,048, 48, 96; and 2,048, 48, 128; respectively, and the total acquisition time was 3 days and 15 h. The data were processed using nmrPipe[45], and zero filling, linear prediction and cosine square window function were applied for all dimensions. Peak picking was performed with Sparky[46], and the peak lists were sent to the I-PINE web server[47] for backbone resonance assignments followed by manual inspection to verify and extend the assignments. The chemical shifts of $^1$H, $^{15}$N$_H$, $^{13}$C$_{alpha}$ and $^{13}$C$_{beta}$ resonances for 75 of 96 total residues in ZFTA were assigned. The chemical shift assignments were used to generate secondary structure predictions for the assigned resonances using CSI 3.0[48]. Secondary structure elements in the AF2 model of ZF1 were assigned using STRIDE[49]. Double-stranded DNA with the sequence 5'-CGGCCCCG-3' (IDT) was prepared with slow annealing as described above and added to a sample of $^{13}$C/$^{15}$N-ZF1 at 20% stoichiometric excess and concentrated with a 3 kDa molecular weight cut-off centrifugal filtration device (Millipore) to a final protein concentration of 140 μM. Resonance shifts and intensity changes in the HSQC spectrum of the DNA:$^{15}$N-ZF1 were followed by inspection and by measuring changes in resonance intensities at the chemical shifts observed in the apo spectrum.

**qRT–PCR.** RNA was extracted as outline before. cDNA was prepared using Qiagen Reverse Transcription Kit (RT31-100) following the manufacturer's recommended protocol. cDNA product was directly used for qRT–PCR analysis using SsoAdvanced Universal SYBR Green Supermix (Bio-Rad 1725270) following the manufacturer's recommended protocol. Primer sequences are provided in Supplementary Table 6.

**IUE.** All animal procedures in this study were performed with Institutional Animal Care and Use Committee approval. After anaesthesia with 5% isoflurane, pregnant mice, at E16.5, were subjected to abdominal incision to expose the uterus. DNA plasmid cocktail (pBCAG-(construct of interest), pbCAG-eGFP, pX330-sg*Tp53* and GLAST-PBase) was injected into the lateral ventricles with a glass pipette. Electric pulses were then delivered to the embryos by gently clasping their heads with forceps-shaped electrodes. Six 33 V pulses of 55 ms were applied at 100-ms intervals. The uterus was then repositioned into the abdominal cavity, and the abdominal wall and the skin were then sutured. After birth, pups were traced on a weekly basis by bioluminescence imaging to monitor brain tumour formation. Mouse tumours were collected on the basis of isolation of GFP$^+$ tumours and dissociation into single cells using the Brain Tumor Dissociation Kit (MACS, #130−095−942) following the manufacturer's instructions.

**IDR expression and purification.** Synthetic DNA (IDT) encoding IDR3, IDR3_GS and IDR3_DE was incorporated into a pCoofy expression vector bearing a His$^6$-MBP tag[49] using HiFi Assembly (New England Biolabs) according to the manufacturer's instructions. IDR3 proteins were expressed in BL21(DE3) *E. coli* (Millipore), which was grown in shaker flasks at 37 °C to an OD$_{600}$ of ~0.6–0.8. At this point, the cultures were cooled to 16 °C, and protein expression was induced by adding 0.5 mM IPTG (GoldBio) to the medium. Cultures were collected via centrifugation ~16 h after induction. Cells were resuspended in lysis buffer (50 mM Tris, 500 mM NaCl and 0.5 mM TCEP, pH 7.5), lysed via sonication and cleared by centrifugation at 30,000$g$ for 30 min at 4 °C, and the supernatant was incubated with Ni$^{2+}$NTA agarose (GoldBio). The resin was extensively washed, and protein was eluted with an imidazole gradient. The His$^6$-MBP tags were removed by incubation with 3C protease (PMID: 33166527) during an overnight dialysis step at 4 °C to remove imidazole. This was followed by orthogonal Ni$^{2+}$-NTA chromatography and then reverse-phase HPLC using a C4 column (Vydac).

**In vitro phase separation microscopy.** Microscopy experiments were performed in 25 mM Tris, pH 7.5, 1 M NaCl, 0.5 mM TCEP and 5% (w/v) PEG-8K with 2 μM rhodamine B. The protein concentration was 50 μM. All microscopy images were acquired on a 3i Marianas system configured with a Yokogawa CSU-W spinning disk confocal microscope utilizing a 100× Zeiss objective.

**MRI.** MRI was performed on a Bruker Biospec 94/20 MRI system (Bruker Biospin MRI GmbH). Before scanning, mice were anaesthetized in a chamber (3% isoflurane in oxygen delivered at 0.5 l min$^{-1}$) and maintained using nose-cone delivery (1–2% isoflurane in oxygen delivered at 0.2 l min$^{-1}$). Animals were provided thermal support using a heated bed with warm water circulation and a physiological monitoring system to monitor breath rate. MRI was acquired with a mouse brain cryoprobe positioned over the mouse head and placed inside an 86-mm transmit/receive coil. After the localizer, T2-weighted rapid acquisition with refocused echoes (RARE) sequences were performed in the coronal (repetition time/echo time = 2,000/20.4 ms, matrix size 256 × 256, field of view 20 mm × 20 mm, slice thickness 0.5 mm, number of slices 16) and axial (repetition time/echo time = 2,500/23 ms, matrix size 256 × 256, field of view 20 mm × 20 mm, slice thickness 0.5 mm, number of slices 32) orientations. Tumour volumes were analysed using 3D Slicer software.

## Statistics and reproducibility

All experiments were independently repeated at least twice with similar results. In each set of live-cell imaging, RNA-seq or CUT&RUN, ZR unmutated was included as a positive control. Sample sizes and the statistical tests are included in the figure legends wherever appropriate. No statistical methods were used to predetermine sample sizes. No randomization was used to collect the data. Data distribution was assumed to be normal, but this was not formally tested. Data collection and analysis were not performed blind. No data were excluded from analyses.

**Reporting summary**

Further information on research design is available in the Nature Portfolio Reporting Summary linked to this article.

## Data availability

Genomic data are accessible through Gene Expression Omnibus under accession number GSE266375. Image analysis data are accessible through BioImage Archive 'ZFTA_RELA HEK293T Puncta Studies' with accession number S-BIAD1556. Previously published crystal structures were reanalysed, which are available under PDB accession codes 1ZNF and 2DRP. Source data are provided with this paper.

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

## Acknowledgements

This study was supported by a NCI Cancer Center Support Grant, P30 CA021765, St. Jude Children's Research Hospital Research Collaborative on Transcription Regulation in Pediatric Cancer Grant, Alex's Lemonade Stand Foundation 'A' Award. This work is supported by R01NS128184, R01CA280203, R01CA284455, U01CA281823, U01CA294103, DOD-IDEA (CA220510) and an DOD-IMPACT (CA220247) award (to S.C.M.) and R01CA246125 and U54CA243124 (to R.W.K.). We also acknowledge the NBTS, CERN Foundation and Robert Connor Dawes Foundation for their ongoing support of brain tumour research, including ependymoma, and this project. We thank and acknowledge A. Ansari for providing reagents, training and support regarding the CSI experiments, which was supported by GM120625 and NS108376. We also acknowledge that part of Fig. 7 was drawn using images adapted from BioRender.com.

## Author contributions

A. Arabzade, H.K.S., S.V., S.M.I., R.W.K. and S.C.M. designed the project and analysed the data. A. Arabzade, H.K.S., S.V., M.J., N.L. and S.M.I. performed cell, molecular and animal experiments. S.M.I., A. Kardian, E.E. and A.B. performed IUE animal experiments. A.P., W.T.F. and B.C. performed purification, biophysical assays and NMR. D.W.B. and A.S. performed imaged analysis. S.T. performed protein sequence and IDR analysis. K.L. helped with cell sorting. A.P., G.E.C., R.A.P., S. King, A.C. and D.S. helped with image acquisition. T.Z., R.P., D.G., T.W., A.H. and B.H. helped with plasmid cloning and preparation. A. Arabzade, H.K.S., S.V., S.M.I., R.W.K. and S.C.M. wrote the manuscript. All authors contributed to editing the manuscript and figures.

## Competing interests

The authors declare no competing interests.

## Additional information

**Extended data** is available for this paper at https://doi.org/10.1038/s41556-025-01745-3.

**Correspondence and requests for materials** should be addressed to R. W. Kriwacki or S. C. Mack.

**A**

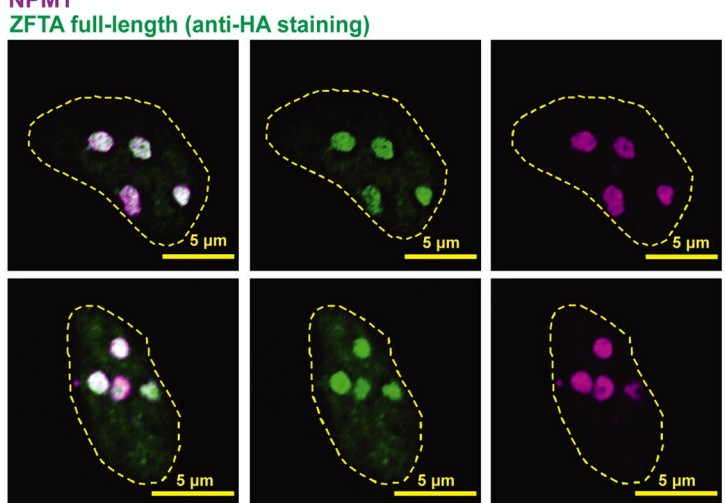

**B**

**G-ZFTA FRAP**

**C**

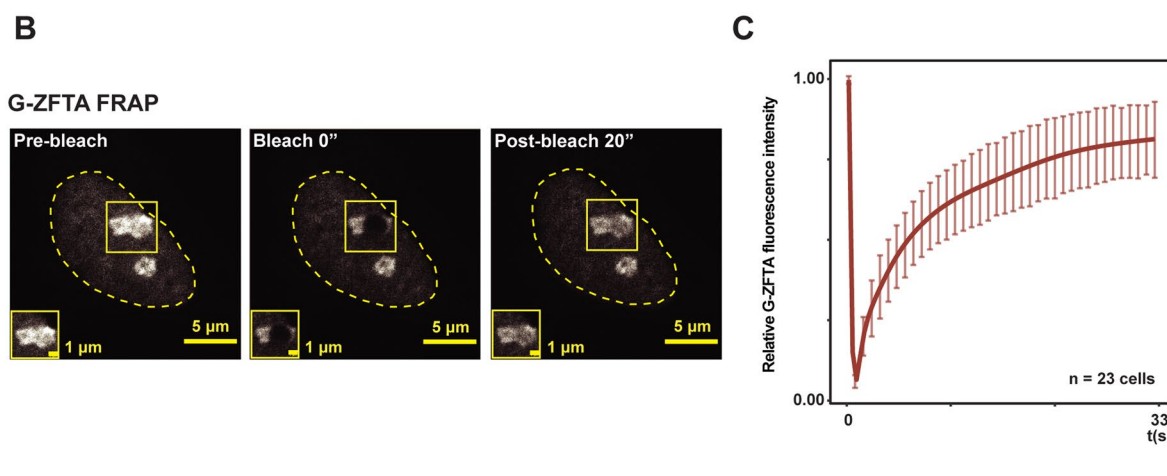

**D**

**E**

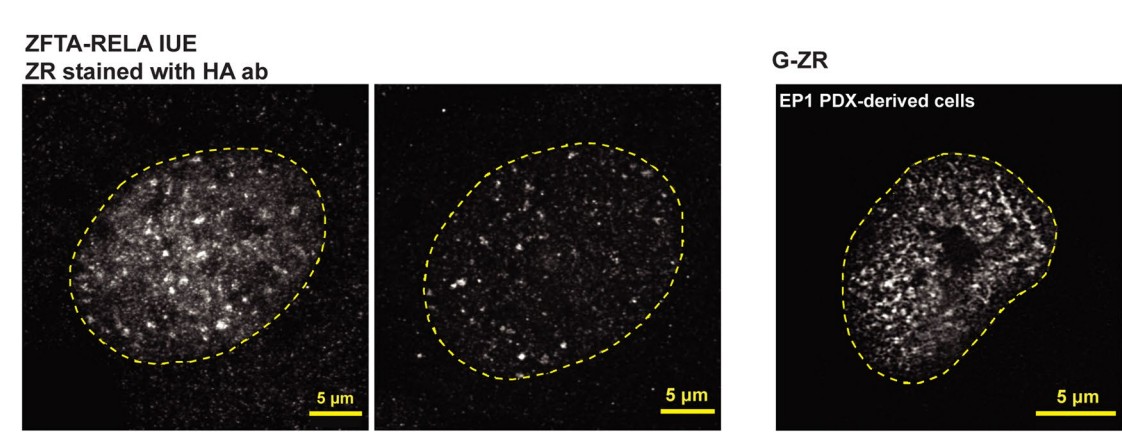

**Extended Data Fig. 1 | ZFTA protein is a dynamic nucleolar protein.**
(**a**) 2 Representative IF images of ZFTA full-length protein in hNSCs. ZFTA full-length protein is visualized via anti-HA staining and nucleolus is marked with NPM1 staining. Each channel individually and channels merged (on the left side) is provided. (**b**) Representative images of hNSCs expressing G-ZFTA before, at the time of bleaching and post-bleach. Zoom-in of the bleached area is provided at the bottom left corner of the images. (**c**) Normalized G-ZFTA-full length fluorescence signal at the bleached area (n = 23 independent cells; data are represented as the fitted curve through the mean values, using leoss method, ± s.d).(**d**) Representative IF images of mouse In Utero Electroporation (IUE) ZFTA-RELA ependymoma model. ZFTA-RELA is stained with anti-HA tag antibody. (**e**) Representative super resolution image of ZFTA-RELA fusion-driven human cell line expressing G-ZR.

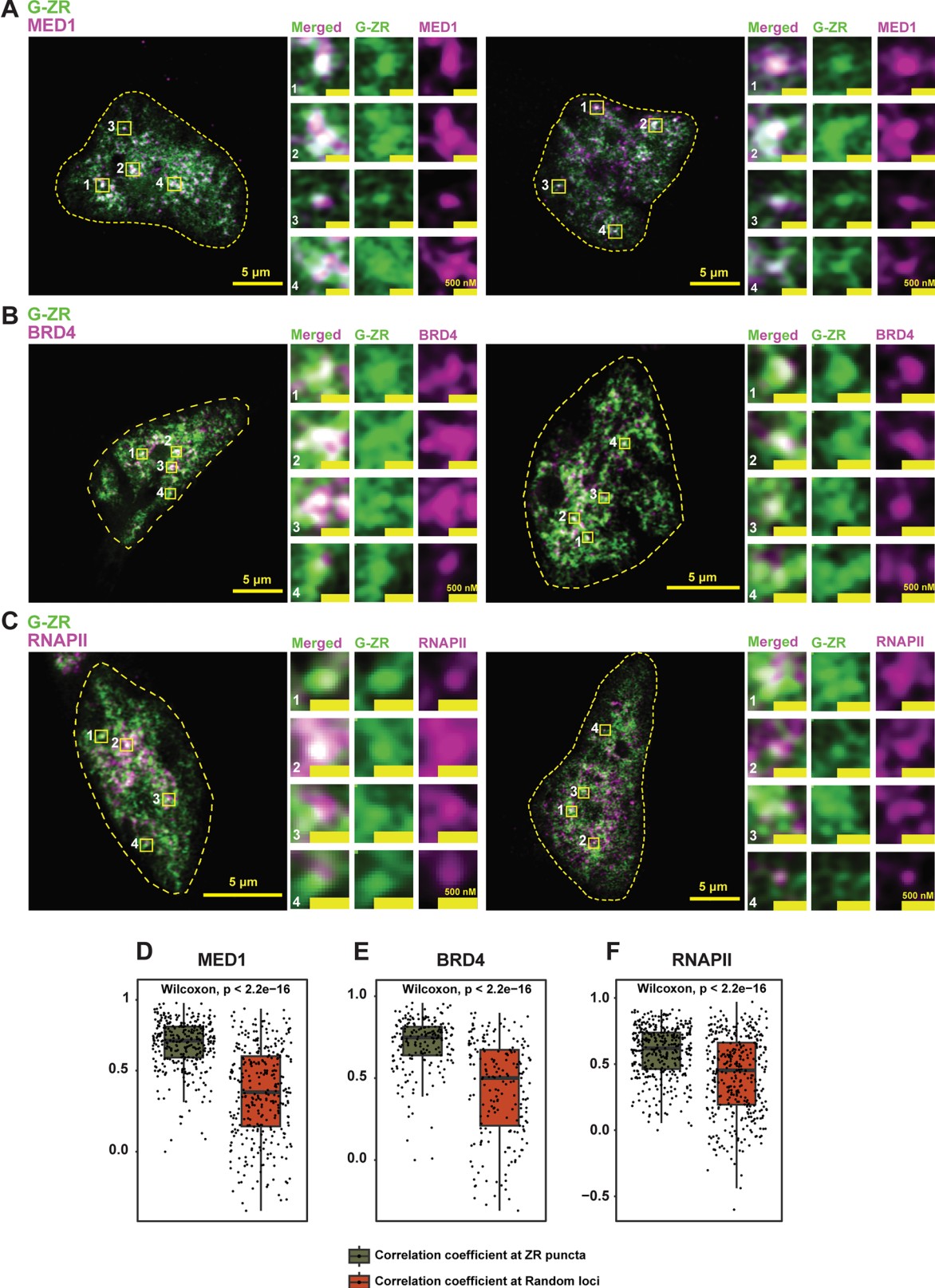

**Extended Data Fig. 2 | ZFTA-RELA condensates are enriched with members of transcription machinery.** More representative super resolution images of hNSCs expressing G-ZR and stained for (**a**) MED1, (**b**) BRD4, and (**c**) RNAPII. Zoomed-in views of 4 loci is provided on the right side of each image with each channel individually as well as merged. Pixel intensity correlation coefficient values between G-ZR and (**d**) MED1, (**e**) BRD4, and (**f**) RNAPII at condensates and randomly selected nuclear loci (n = 319, 197 and 372 condensates from 9, 16 and 17 cells for MED1, BRD4 and RNAPII, respectively. Cells were pooled from 2 independent replicates; two-side Wilcoxon. Boxplots represent median and interquartile range (IQR). The whiskers extend to the smallest/largest values within 1.5xIQR).

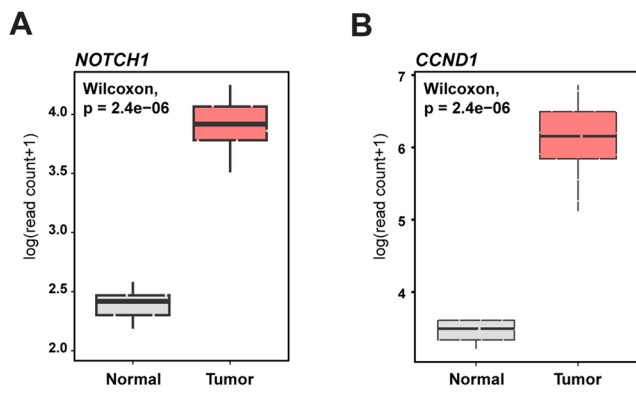

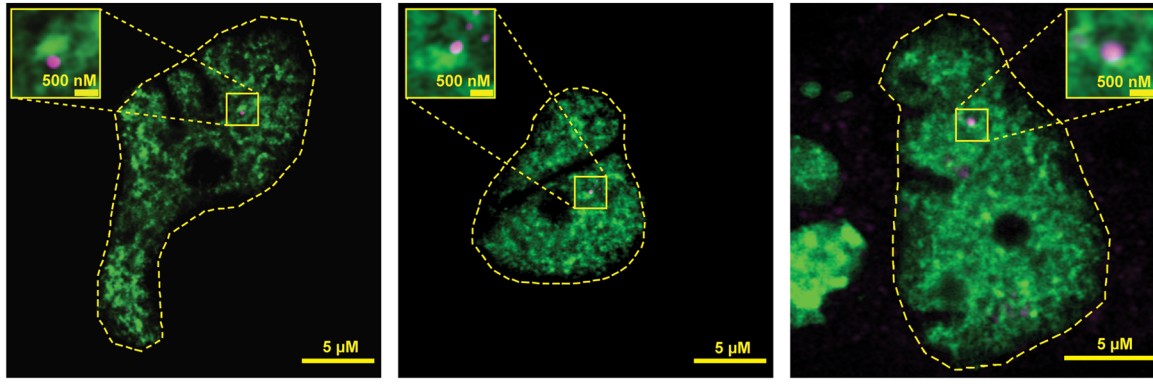

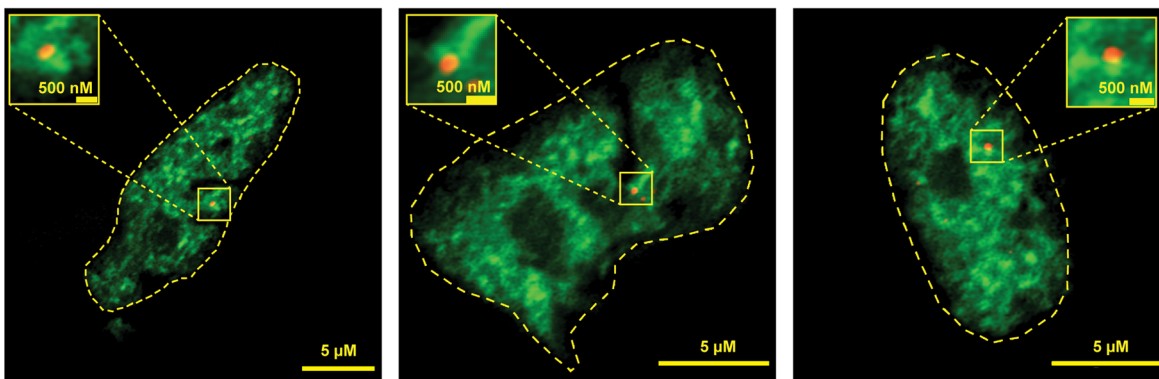

**Extended Data Fig. 3 | ZFTA-RELA condensate are associated with sites of active transcription for Ependymoma oncogenes.** Expression comparison between mouse IUE model and healthy brain tissue for (**a**) *NOTCH1* and (**b**) *CCND1* (n = 14 mice for tumor and 9 mice for normal; two-sided Wilcoxon test. Boxplots represent median IQR. The whiskers extend to the smallest/largest values within 1.5xIQR. More representative super resolution images of Nascent RNA-FISH in neural stem cells for (**c**) *NOTCH1* and (**d**) *CCND1*.

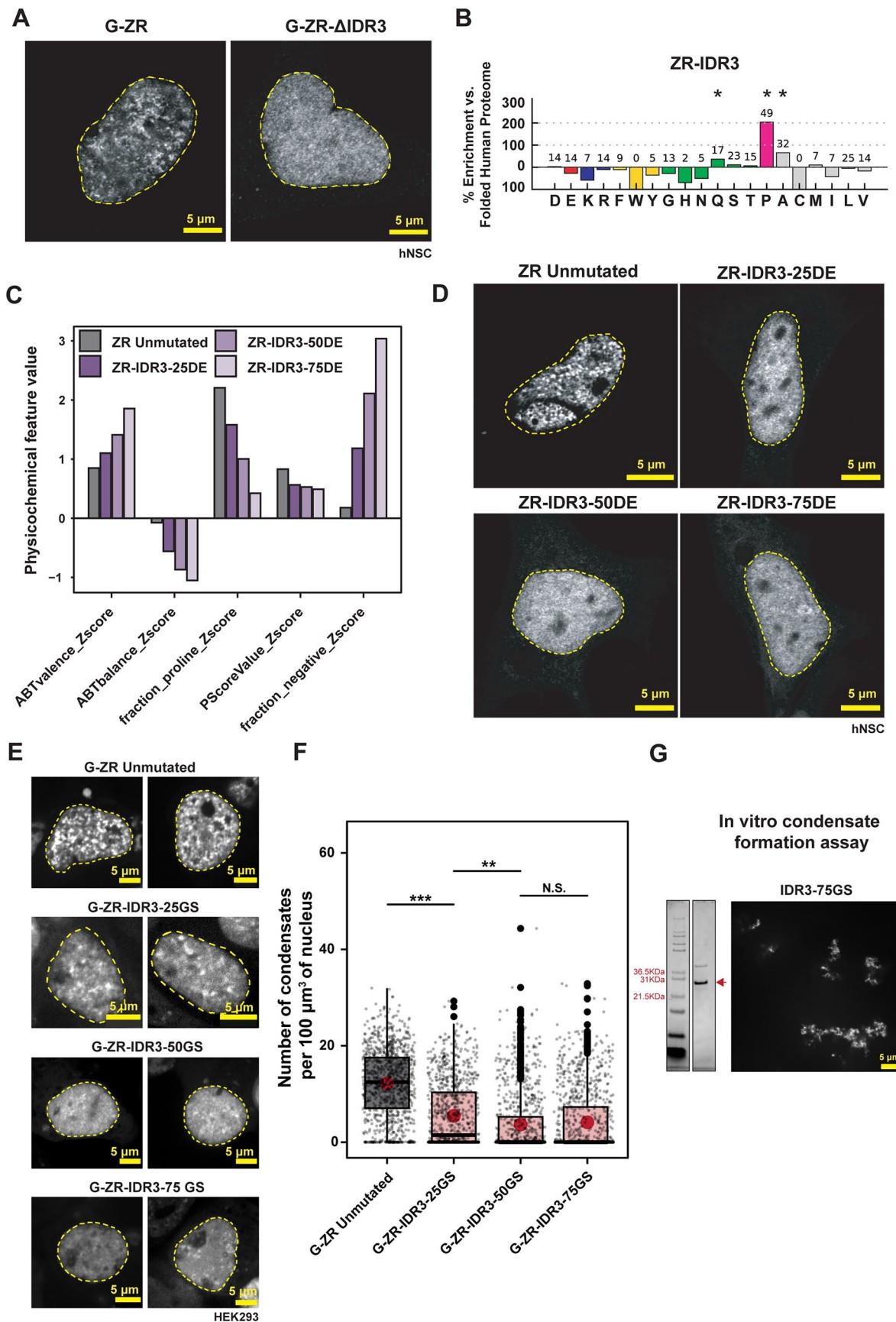

**Extended Data Fig. 4 | See next page for caption.**

**Extended Data Fig. 4 | ZFTA-RELA IDR3 gradual mutations pinpoint residues necessary for condensate formation.** (**a**) Representative images of hNSCs expressing G-ZR or G-ZR-ΔIDR3. (**b**) Amino acid enrichment analysis of the IDR3. Highly enriched amino acids are marked with *. (**c**) Significantly altered physiochemical features in generation of substitution mutants (G-ZR-IDR3-25DE, −50DE, or −75DE). For a full description refer to ref. 23. (**d**) Representative images of hNSCs expressing G-ZR or IDR3 D/E substitution mutants (G-ZR-IDR3-25DE, −50DE, or −75DE). (**e**) Representative images of HEK293T cells expressing ZR unmutated or IDR3 G/S mutants. (**f**) Number of condensates quantification for HEK293T cells expressing unmutated ZR or IDR3 G/S mutants. (n = 996, 1062, 1009 and 943 independent cells for G-ZR unmutated, G-ZR-IDR3-25GS, G-ZR-IDR3-50GS and G-ZR-IDR3-75GS, respectively. Images from one imaging session was used for analysis and each cell was considered a technical replicate; two-sided ANOVA test. Boxplots represent median and IQR. The whiskers extend to the smallest/largest values within 1.5xIQR. Outliers are shown with circles. Stars indicate the p values-see below). (**g**) Representative in vitro condensate formation assay image performed using purified IDR3-75GS. Coomassie stained protein gel image of the eluent is provided on the left side to indicate the purity of protein sample. Arrow points to the expected band for the purified protein. N.S = non-significant, * = p value < 0.05, ** = p value < 0.01 and *** = p value < 0.001.

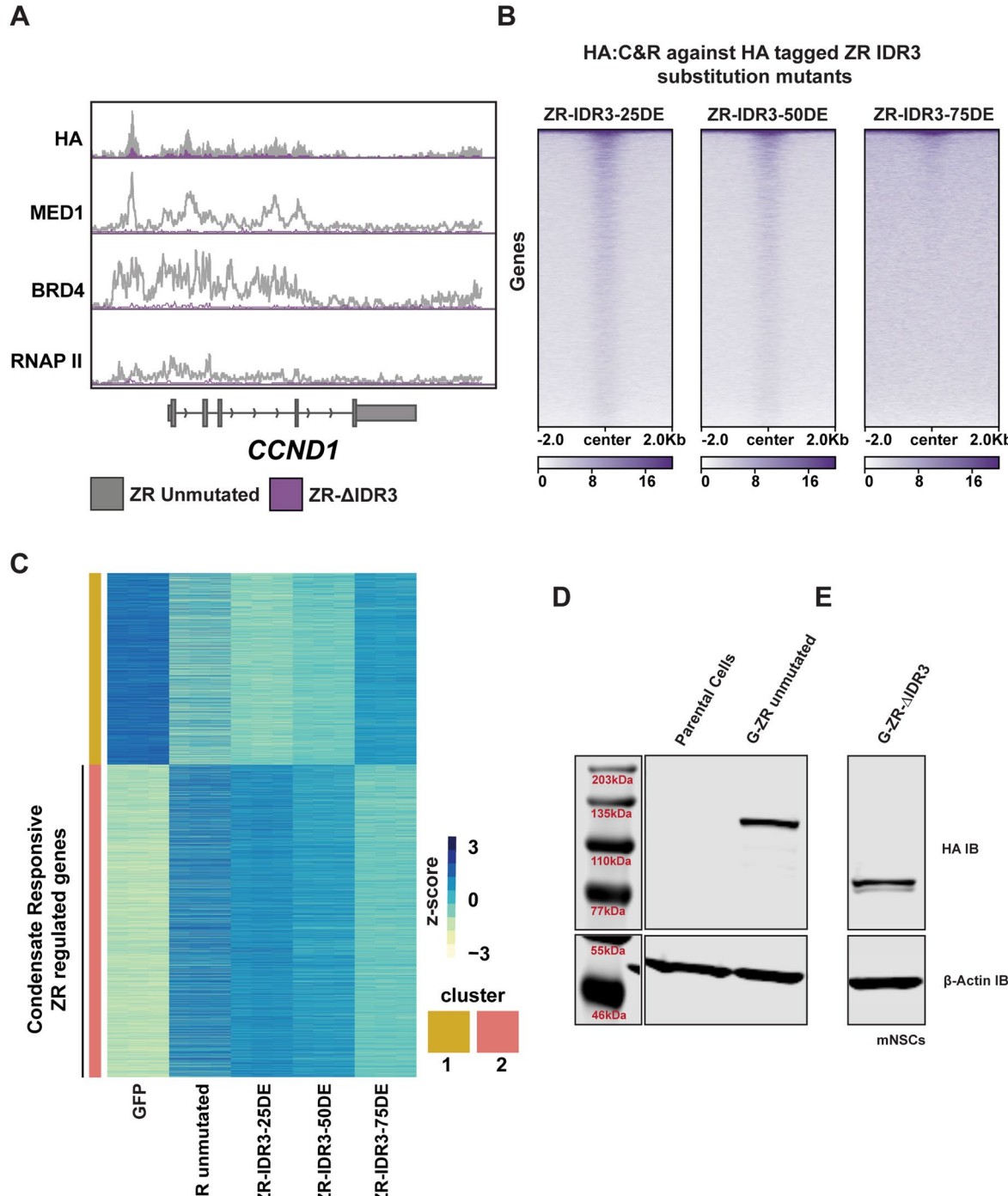

**Extended Data Fig. 5 | ZFTA-RELA IDR3 deletion or mutation alters chromatin occupancy and transcriptional activity.** (**a**) CCND1 locus depicting a loss of genomic occupancy and diminished recruitment of transcriptional machinery following IDR3 deletion. (**b**) HA C&R heatmap comparison between ZR-IDR3- 25DE, −50DE and −75DE. (**c**) Unsupervised hierarchical clustering analysis of ZR regulated genes based on their response to gradual condensate perturbation. (**d**) and (**e**) Western blot validation of expression of unmutated ZR and ZR-ΔIDR3 in NSCs prior to intracranial injection.

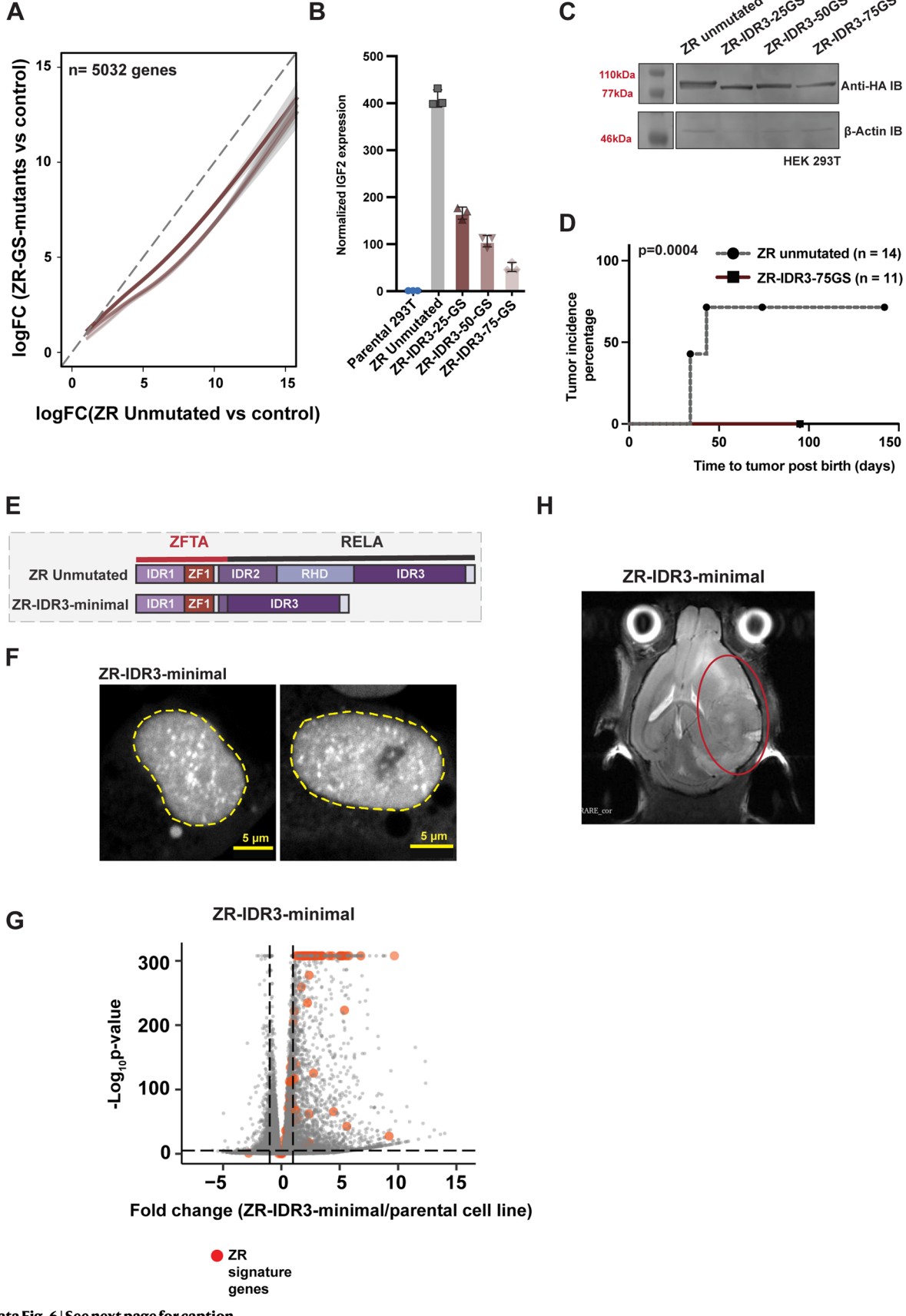

**Extended Data Fig. 6 | See next page for caption.**

**Extended Data Fig. 6 | Abolishing ZFTA-RELA condensates disrupts gene activation and tumorigenicity.** (**a**) Transcriptional activity comparison between unmutated ZR and ZR-IDR3-25GS, −50GS or −75GS. (n = 5032 genes, data is represented as fitted regression line with the shaded area denoting the 95% confidence interval). (**b**) Normalized *IGF2* expression comparison between unmutated ZR and gradual G/S mutants. (data is represented as mean ± s.d. N = 3 technical replicates) (**c**) Western blot analysis indicating expression of each protein in the cells used for qRT-PCR analysis in panel **b**. (**d**) Time to tumor formation for ZR unmutated and ZR-IDR3-75GS. Tumor incidence is defined as the first detection of tumor in MRI scan. P = 0.0004 (n = 14 mice for ZR unmutated and n = 11 mice for ZR-IDR3-75GS; Mantel-Cox test). (**e**) Schematic cartoon for unmutated ZR and ZR-IDR3-minimal mutant. (**f**) Representative HEK293T cells expressing ZR-IDR3-minimal mutant. (**g**) Global transcriptional changes following ZR-IDR3-minimal expression in HEK293T cells. ZR signature genes are highlighted with red dots (two-sided Wald test corrected for multiple testing using the BH method. N = 3 independent replicates). (**h**) Representative MRI image of brain tumor generated using ZR-ID3-minimal mutant using IUE technique.

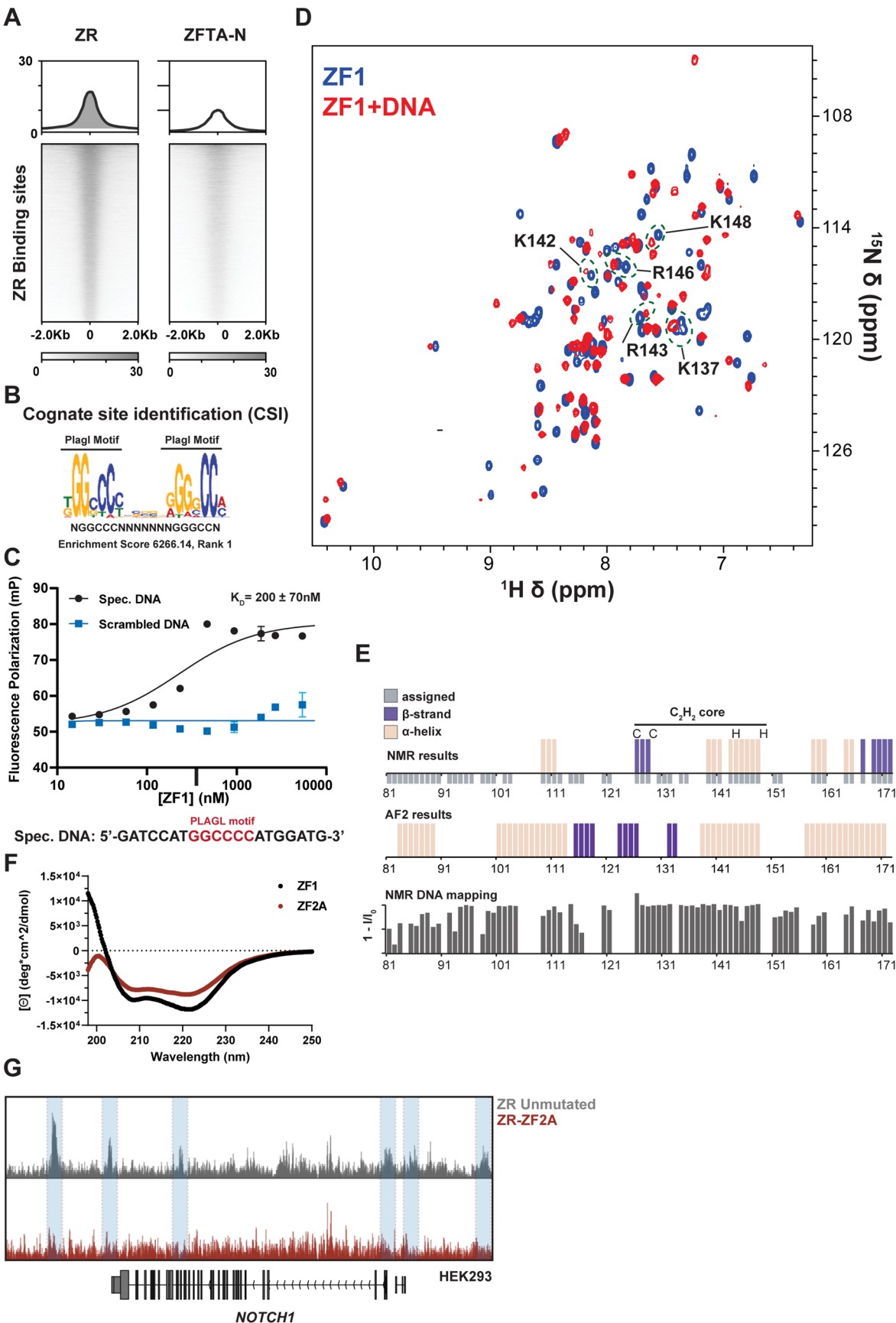

**Extended Data Fig. 7 | See next page for caption.**

**Extended Data Fig. 7 | NMR data pinpoints residues necessary for ZFTA-RELA ZnF1 folding and DNA binding.** (**a**) CUT&RUN data indicating ZF1 mediates DNA binding with similar binding sites as ZR. (**b**) Cognate site identification (CSI) experiments[26] showed that ZR binds to PLAGL motifs. The DNA sequence with highest enrichment score is shown. (**c**) Fluorescence polarization (FP) results showing that purified ZF1 protein binds specifically to the PLAGL motif identified by CSI (data is represented as mean ± s.d; data is fit to a 1:1 binding model. N = 3 technical replicates for each condition. One independent replicate is shown but experiment was repeated 3 time independently to ensure reproducibility). (**d**) $^1$H-$^{15}$N HSQC spectra of ZF1 in the presence (red peaks) and absence (blue peaks) of DNA. (**e**) Validation of AF2 ZF1 structure using NMR. (top panel)

Residues with assigned NMR resonances are denoted with negative grey bars. Secondary structure predictions from NMR data using CSI 3.0 are indicated by positive-colored bars (a-helix, orange; b-strand, purple). (middle panel) Secondary structure observed in the AF2 structure of ZF1 as assigned by STRIDE (colored as the top panel). (bottom panel) Mapping ZF1/DNA interactions based on amide backbone NMR intensity changes in the 2D $^1$H-$^{15}$N HSQC spectrum of ZF1 (from panel **d**) after addition of DNA [plotted as $(1 - I/I_0)$, where I is resonance intensity in the presence of DNA and $I_0$ in the absence of DNA]. (**f**) Circular Dichroism spectra of purified MBP-tagged ZF1 and ZF2A showing that structure of ZF2A mutant is disrupted. (**g**) Loss of DNA binding at CCND1 locus following ZF1 mutation.

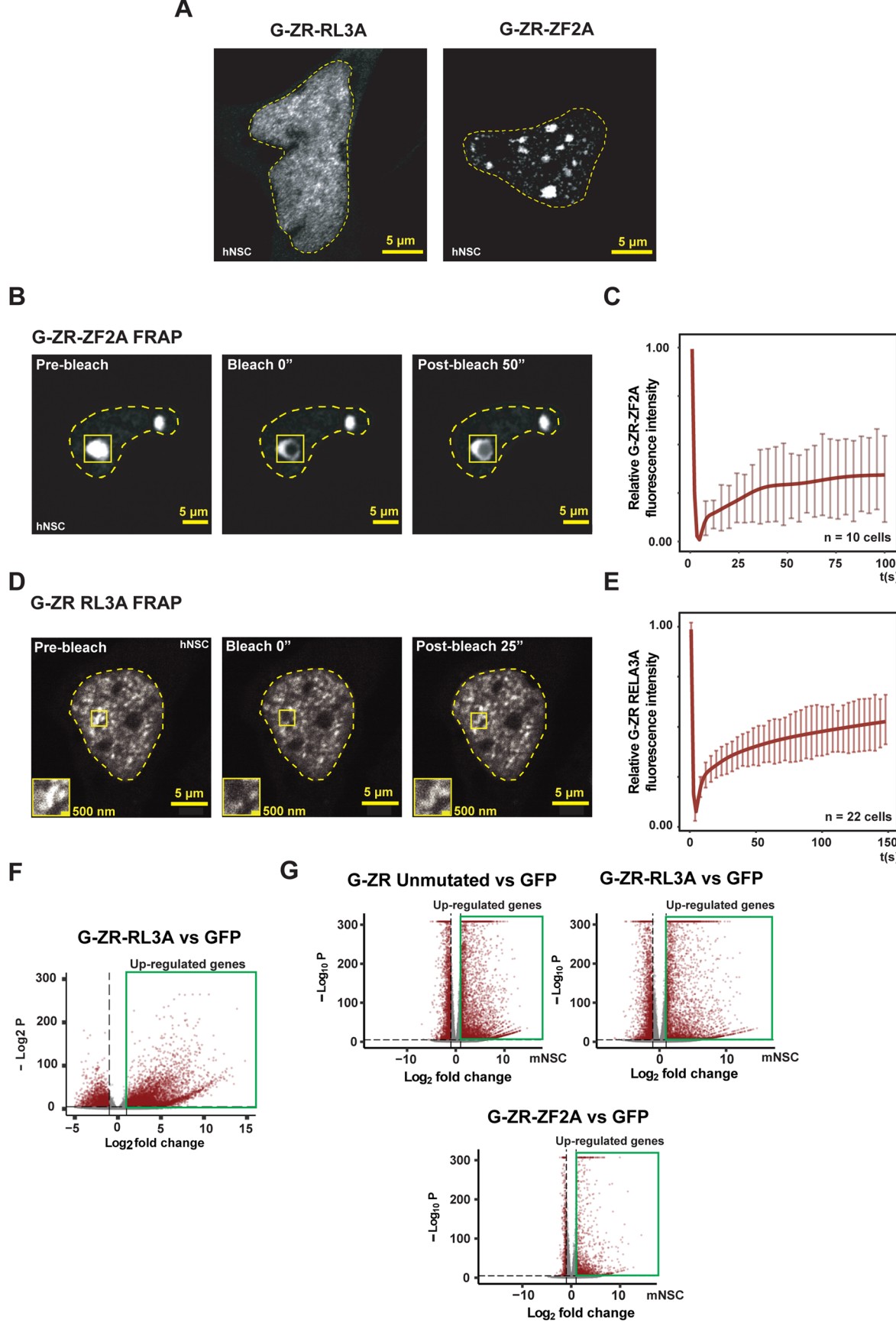

**Extended Data Fig. 8 | See next page for caption.**

**Extended Data Fig. 8 | ZFTA-RELA DNA binding mutants retain liquid-like characteristics.** (**a**) Representative images of hNSCs expressing G-ZR-RL3A or G-ZR-ZF2A. (**b**) Representative images of hNSCs expressing G-ZR-ZF2A before, at the time of bleaching and post-bleach. (**c**) Normalized G-ZR-ZF2A fluorescence signal at the bleached area (n = 10 independent cells from 2 independent replicates; data is represented as the fitted curve through mean values, using leoss method, ± s.d). (**d**) Representative images of hNSCs expressing G-ZR-RL3A before, at the time of bleaching and post-bleach. Zoom-in of the bleached area is provided at the bottom left corner of the images. (**e**) Normalized G-ZR-RL3A fluorescence signal at the bleached area (n = 22 independent cells from 2 independent replicates. Data is represented as the fitted curve through mean values, using leoss method, ± s.d). (**f**) Global expression changes following G-ZR-RL3A expression in HEK 293 T cells (two-sided Wald test corrected for multiple testing using the BH method. N = 3 independent replicates). (**g**) Global expression changes following expression of G-ZR, G-ZR-RL3A or G-ZR-ZF2A in mNSCs (two-sided Wald test corrected for multiple testing using the BH method. N = 3 independent replicates).

**A**

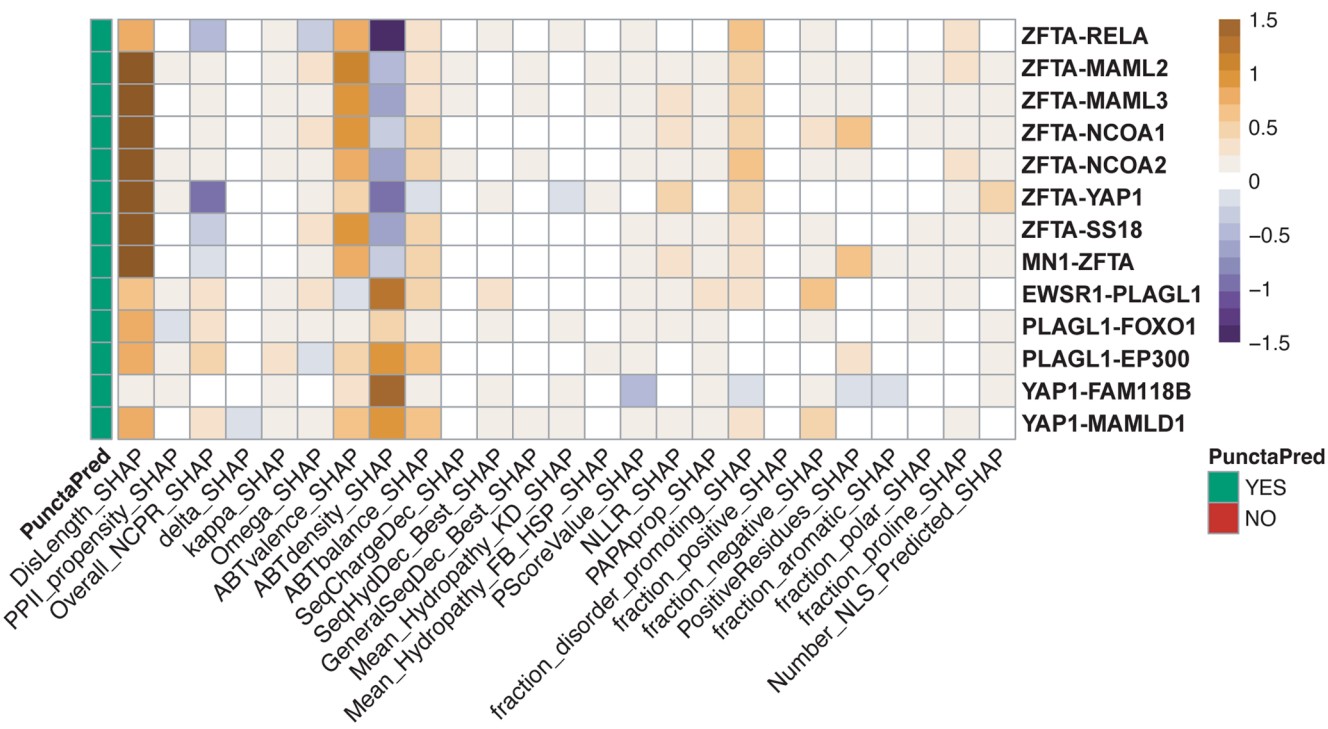

**B**

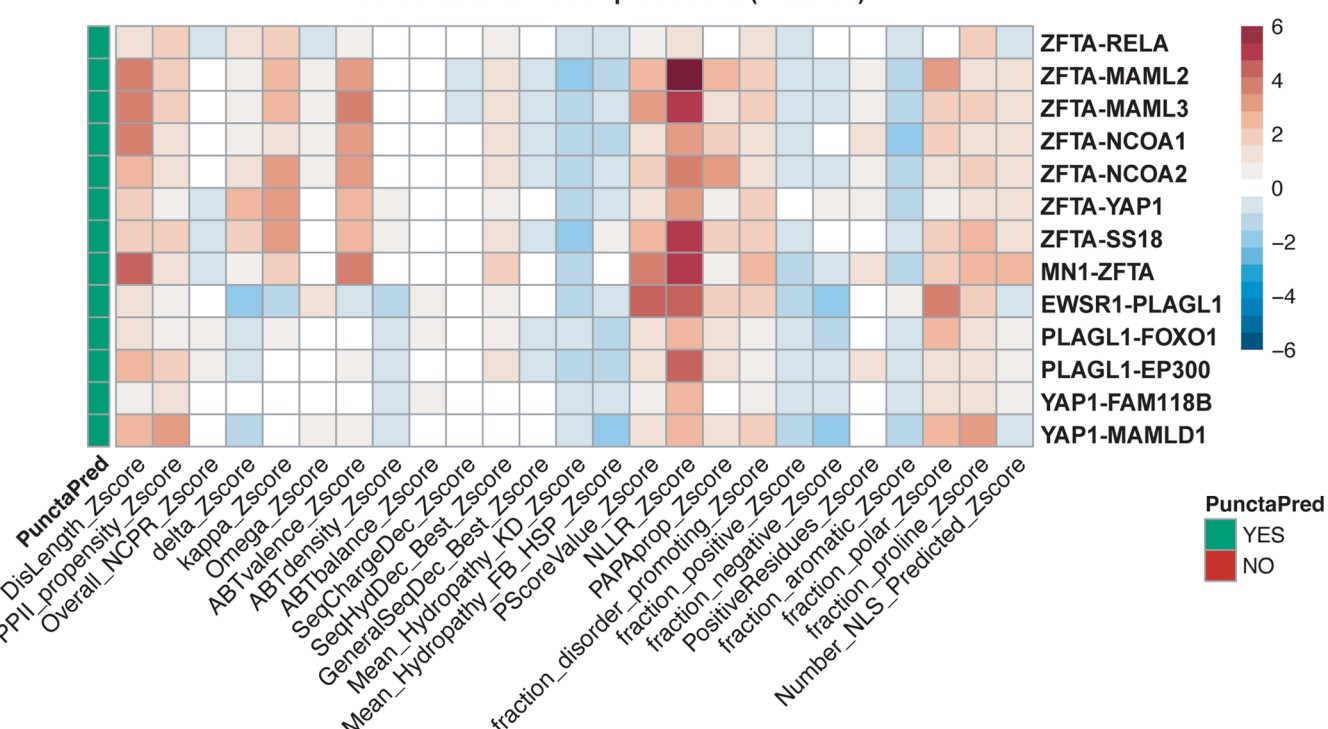

**Extended Data Fig. 9 | Contribution of physiochemical features of ZFTA FOs towards condensate formation.** (**a**) Contribution of each physiochemical feature in predicting the condensate formation probability of each fusion. A positive value indicates favorable contribution. For a full description of each physiochemical feature refer to ref. 23. (**b**) Values of 25 physiochemical features used in FO-Puncta-ML model condensate formation predictions (Z-scores).

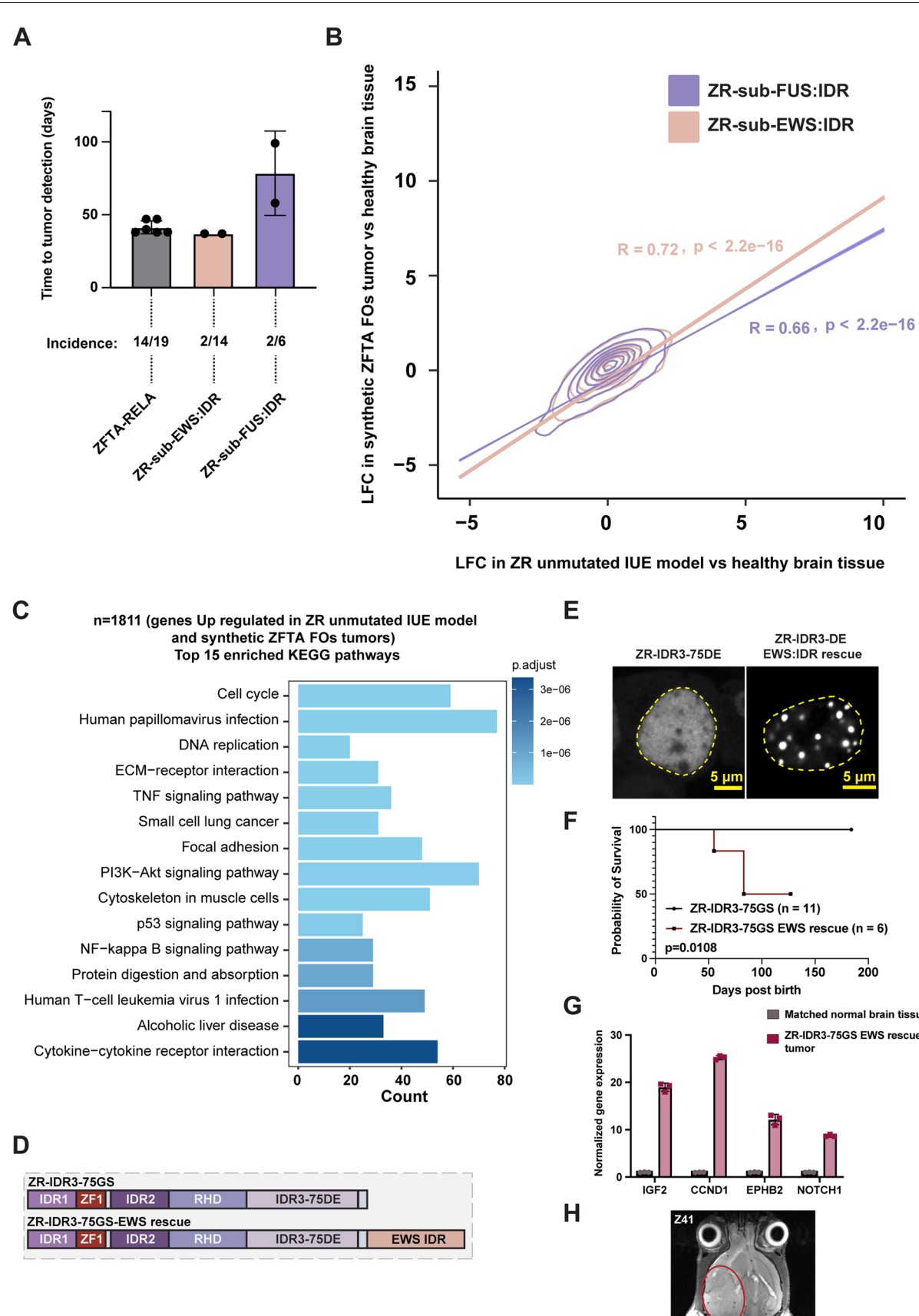

**Extended Data Fig. 10 | See next page for caption.**

**Extended Data Fig. 10 | Synthetic ZFTA fusions drive oncogenic transcription and tumorigenesis. (a)** Time to tumor detection between ZFTA-RELA, ZR-sub-EWS:IDR or ZR-sub-FUS:IDR synthetic fusions (data are represented as mean ± s.d). N = 19, 14 and 6 total IUE mice for ZFTA-RELA, ZR-sub-EWS:IDR and ZR-sub-FUS:IDR, respectively. 14 mice for ZFTA-RELA and 2 mice for each ZR-sub-EWS:IDR or ZR-sub-FUS:IDR developed tumors. **(b)** Log fold change in gene expression in ZR unmutated IUE model versus healthy brain tissue compared against log fold change in gene expression in synthetic ZFTA FOs (ZR-sub-FUS:IDR and ZR-sub-EWS:IDR) vs healthy brain tissue. LFCs for ZR-sub-FUS:IDR and ZR-sub-EWS:IDR were calculated using 3 technical replicates from 1 tumor compared to matched normal brain tissue from the same mouse. For ZR unmutated, 2 independent IUE tumors were used. **c)** KEGG pathway enrichment for genes that are up regulated (LFC > 1 and p-value < 0.05) in ZR unmutated and synthetic ZFTA FOs IUE (one-sided p-value for over-representation analysis is calculated by hypergeometric distribution with BH correction). **(d)** Schematic representation of ZR-IDR3 DE mutant and corresponding EWS rescue construct. **(e)** Representative images of HEK293T cells expressing ZR-IDR3 DE mutant or the corresponding EWS rescue construct. **(f)** Survival probability curve for mice undergoing IUE using ZR-IDR3-75GS or the EWS rescue construct (Mantel-Cox test). N = 11 independent mice for ZR-IDR3-75GS and 6 independent mice for ZR-IDR3-75GS EWS rescue. **g)** Gene expression comparison using qPCR. A few commonly overexpressed genes in ZR ependymoma were compared in matched normal brain tissue vs EWS rescue tumor tissues (data are represented as mean ± s.d). N = 3 technical replicates from 1 ZR-IDR3-75GS EWS rescue tumor and matched normal brain tissue from the same mouse. **(h)** Representative MRI image of mouse harboring brain tumor generated using ZR-IDR3-75DE EWS rescue construct.

# Reporting Summary

## Statistics

For all statistical analyses, confirm that the following items are present in the figure legend, table legend, main text, or Methods section.

| n/a | Confirmed | |
|---|---|---|
| ☐ | ☒ | The exact sample size (*n*) for each experimental group/condition, given as a discrete number and unit of measurement |
| ☐ | ☒ | A statement on whether measurements were taken from distinct samples or whether the same sample was measured repeatedly |
| ☐ | ☒ | The statistical test(s) used AND whether they are one- or two-sided<br>*Only common tests should be described solely by name; describe more complex techniques in the Methods section.* |
| ☐ | ☒ | A description of all covariates tested |
| ☐ | ☒ | A description of any assumptions or corrections, such as tests of normality and adjustment for multiple comparisons |
| ☐ | ☒ | A full description of the statistical parameters including central tendency (e.g. means) or other basic estimates (e.g. regression coefficient) AND variation (e.g. standard deviation) or associated estimates of uncertainty (e.g. confidence intervals) |
| ☐ | ☒ | For null hypothesis testing, the test statistic (e.g. *F*, *t*, *r*) with confidence intervals, effect sizes, degrees of freedom and *P* value noted<br>*Give P values as exact values whenever suitable.* |
| ☒ | ☐ | For Bayesian analysis, information on the choice of priors and Markov chain Monte Carlo settings |
| ☒ | ☐ | For hierarchical and complex designs, identification of the appropriate level for tests and full reporting of outcomes |
| ☒ | ☐ | Estimates of effect sizes (e.g. Cohen's *d*, Pearson's *r*), indicating how they were calculated |

*Our web collection on statistics for biologists contains articles on many of the points above.*

## Software and code

Policy information about availability of computer code

| Data collection | Proteome Discoverer 3.0 for proteomic data acquisition, ZEN 2.1 (blue edition) for microscopic images acquisition. |
|---|---|
| Data analysis | Cell Ranger 7.0.0 for scRNA-seq, R v.4.2 for multiple analyses including: Seurat Rpackage v.3.1.5 for scRNA-seq, RiboR for ribo-seq, and edgeR v3 for RNA-seq. FIJI forimage analysis. GraphPad Prism 8.2.0 for statistical analysis. |

For manuscripts utilizing custom algorithms or software that are central to the research but not yet described in published literature, software must be made available to editors and reviewers. We strongly encourage code deposition in a community repository (e.g. GitHub). See the Nature Portfolio guidelines for submitting code & software for further information.

## Data

Policy information about availability of data

All manuscripts must include a data availability statement. This statement should provide the following information, where applicable:

- Accession codes, unique identifiers, or web links for publicly available datasets
- A description of any restrictions on data availability
- For clinical datasets or third party data, please ensure that the statement adheres to our policy

Sequencing data that support the findings of this study have been deposited in the Gene Expression Omnibus (GEO) under accession code GSE247456. All analysis was done on human genome version hg38. Proteomics datasets have been deposited and are available at the ProteomeXchange Consortium under accession code

PXD063298. Other forms of source data are provided in this study. Reagents generated or any other information supporting the findings of this study are available from the corresponding author on reasonable request.

# Research involving human participants, their data, or biological material

Policy information about studies with human participants or human data. See also policy information about sex, gender (identity/presentation), and sexual orientation and race, ethnicity and racism.

| | |
|---|---|
| Reporting on sex and gender | We describe 11 families with allelic variants in AIRIM. Sex was assigned and reported in the appropriate figures and tables. This is not a population study, therefore sex- and gender based analyses were not performed. |
| Reporting on race, ethnicity, or other socially relevant groupings | Ancestry of the patients was reported by family members. |
| Population characteristics | No data on population characteristics was collected or used in this study. |
| Recruitment | N/A |
| Ethics oversight | Ethical oversight for all human data described in this study was provided by Phoenix Children's Hospital (PCH IRB #15-080), University College London Queen Square Institute of Neurology (22/NE/0080, project ID 310045), Heidelberg University (S-186/2012), King Faisal Hospital Specialist & Research Centre (20DG1533: RAC#2121053 and 23DG0161: RAC# 2210029), National and Kapodistrian University of Athens (16434/25-07-22), and King Abdullah International Medical Research Center (IRB/1470/24, project number NRC23R/177/02). |

Note that full information on the approval of the study protocol must also be provided in the manuscript.

# Field-specific reporting

Please select the one below that is the best fit for your research. If you are not sure, read the appropriate sections before making your selection.

☒ Life sciences ☐ Behavioural & social sciences ☐ Ecological, evolutionary & environmental sciences

For a reference copy of the document with all sections, see nature.com/documents/nr-reporting-summary-flat.pdf

# Life sciences study design

All studies must disclose on these points even when the disclosure is negative.

| | |
|---|---|
| Sample size | Proteomic analysis: For each batch, 300 EBs, 300 day 10 organoids and 100 day 15 organoids were used for control and V190G mutant. Samples from three individual batches were analyzed, as a higher degree of accuracy was desired than typical in the field for detecting quantitative changes at individual proteins across organoid development.<br>scRNAseq: 50 day 5 embryoid bodies, 3-6 day 10 neuroepithelia, 3-6 day 15 organoids and 3 day 30 organoids of each genotype were pooled for each dissociation. For each condition, approximately 10,000 cells were sequenced, similar to other studies in the field, where large effects can be easiliy identified at this sample size.<br>Single organoid ribo-seq: Two control and three mutant day 10 organoids were used, where large effects can be easiliy identified at this sample size.<br>Bulk RNA-seq: 300 organoids were pooled and sequenced per genotype, per replicate, with a total of 2 biological replicates, where large effects can be easiliy identified at this sample size.<br>Organoid size analysis: a minimum of n=2 individual batches were performed, with mesaurements on multiple individual organoids per experiments.<br>Immunofluorescence/TUNEL/OP-Puro imaging/Mitochondrial aggregation: a miniimum of n=3 individual oragnoid staining was performed for quantitative analysis.<br>No statistical methods were used to predetermine sample sizes. Sample size was determined based on previous studies in the field(Lancaster, M. A. et al. Cerebral organoids model human brain development and microcephaly. Nature 501, 10.1038/nature12517 (2013), Benito-Kwiecinski, Silvia et al. Cell, Volume 184, Issue 8, 2084 - 2102.e19 An early cell shape transition drives evolutionary expansion of the human forebrain). |
| Data exclusions | No data were excluded from imaging-based analysis, scRNA-seq, and ribo-seq. |
| Replication | Western blots, sea-horse, electron microscopy and teratoma assays were runat least twice, independently, with similar results. Organoid size quantification, pH3 counting, TUNEL quantification, OP-puro quantification and proteomic analysis were performed on at least two independent biological replicates with similar results. |
| Randomization | Samples were assigned to groups based on genotype and treatment conditions. |
| Blinding | No blinding was performed. All samples were processed and analyzed equally. |

# Reporting for specific materials, systems and methods

We require information from authors about some types of materials, experimental systems and methods used in many studies. Here, indicate whether each material, system or method listed is relevant to your study. If you are not sure if a list item applies to your research, read the appropriate section before selecting a response.

## Materials & experimental systems

| n/a | Involved in the study |
|---|---|
| ☐ | ☒ Antibodies |
| ☐ | ☒ Eukaryotic cell lines |
| ☒ | ☐ Palaeontology and archaeology |
| ☐ | ☒ Animals and other organisms |
| ☒ | ☐ Clinical data |
| ☒ | ☐ Dual use research of concern |
| ☒ | ☐ Plants |

## Methods

| n/a | Involved in the study |
|---|---|
| ☒ | ☐ ChIP-seq |
| ☒ | ☐ Flow cytometry |
| ☐ | ☒ MRI-based neuroimaging |

## Antibodies

| | |
|---|---|
| Antibodies used | PAX6 (BioLegend, 90130, 1:100)<br>BLBP (Abcam, ab32423, 1:100)<br>ZEB2(OTI1E12) (OriGene, TA802113, 1:100)<br>ZO1(1/ZO-1) (BD Biosciences, 610966, 1:100)<br>GFP (R&D Systems, AF4240, 1:100)<br>HSP60 (Proteintech, 15282-1-AP, 1:100)<br>Vimentin(V9) (Thermo Fisher, MA5-11883, 1:100)<br>KI67(SolA15) (Thermo Fisher, 14-5698-82, 1:100)<br>p53(7F5) (Cell Signaling, 2527S, 1:100)<br>OCT3/4(C-10) (Santa Cruz, sc-5279, 1:100)<br>SOX2(E4) (Santa Cruz, sc-365823, 1:100)<br>RSL24D1 (Proteintech, 25190-1-AP, 1:100 for IF)<br>RSL24D1 (Proteintech, 25190-1-AP, 1:500 for WB)<br>RPL28 (Abcam, ab138125, 1:1000 for WB)<br>TSC1(D43E2) (Cell Signaling, 6935, 1:1000 for WB)<br>GAPDH(6C5) (Millipore, MAB374, , 1:1000 for WB)<br>Secondary antibodies:<br>Donkey-anti-Mouse Alexa 488, 568 conjugated secondary (Thermofisher A-21202, A10037, 1:250);<br>Donkey-anti-Rabbit Alexa 488 conjugated secondary (Thermofisher A-21206, A10042, 1:250);<br>Donkey-anti-Rat Alexa 488, 568 conjugated secondary (Thermofisher A-21208, 1:250);<br>Donkey-anti-Goat Alexa 488, 568, 647 conjugated secondary antibodies(Thermofisher, A-11055, A-11057, A-21447, 1:250).<br>Peroxidase IgG Fraction Monoclonal Mouse Anti-Rabbit IgG, light chain specific, (Jackson ImmunoResearch, 211-032-171, 1:5000).<br>Peroxidase AffiniPure Donkey Anti-Mouse IgG (H+L) (Jackson ImmunoResearch, 715-035-150, 1:1000). |
| Validation | PAX6 (BioLegend, 90130, 1:100): Validated by BioLegend and used in 337 scientific literatures.<br>BLBP (Abcam, ab32423, 1:100): Validated by Abcam and used in 99 scientific literatures.<br>ZEB2 (OTI1E12) (OriGene, TA802113, 1:100): Validated by OriGene and used in 8 scientific literatures.<br>ZO1 (1/ZO-1) (BD Biosciences, 610966, 1:100): Validated by BD Biosciences and used in 83 scientific literatures.<br>GFP (R&D Systems, AF4240, 1:100): Validated by R&D Systems and used in 17 scientific literatures.<br>HSP60 (Proteintech, 15282-1-AP, 1:100): Validated by Proteintech and used in 149 scientific literatures.<br>Vimentin (V9) (Thermo Fisher, MA5-11883, 1:100): Validated by Thermo Fisher and used in 246 scientific literatures.<br>KI67 (SolA15) (Thermo Fisher, 14-5698-82, 1:100): Validated by Thermo Fisher and used in 414 scientific literatures.<br>p53 (7F5) (Cell Signaling, 2527S, 1:100): Validated by Cell Signaling Technology and used in 666 scientific literatures.<br>OCT3/4 (C-10) (Santa Cruz, sc-5279, 1:100): Validated by Santa Cruz Biotechnology and used in 2693 scientific literatures.<br>SOX2 (E4) (Santa Cruz, sc-365823, 1:100): Validated by Santa Cruz Biotechnology and used in 355 scientific literatures.<br>RSL24D1 (Proteintech, 25190-1-AP, 1:100): Validated by Proteintech and used in 4 scientific literatures.<br>RPL28 (Abcam, ab138125): Validated by Abcam and used in 4 scientific literatures.<br>TSC1 (D43E2) (Cell Signaling, 6935): Validated by Cell Signaling Technology and used in 99 scientific literatures.<br>GAPDH (6C5) (Millipore, MAB374): Validated by Millipore and used in 428 scientific literatures. |

## Eukaryotic cell lines

Policy information about cell lines and Sex and Gender in Research

| | |
|---|---|
| Cell line source(s) | SCVI274 was obtained from the Stanford CVI Biobank. Human ES cell line H9 (WA09) was obtained from WiCell. The HEK293T cell line was obtained from ATCC (American Type Culture Collection) |
| Authentication | SCVI274 cells and its derivatives were validated by karyotyping by WiCell. Engineered cells were validated by sequencing. H9 |

| Authentication | cells were authenticated by short tandem repeat (STR) profiling. HEK293T cells were validated by confirming their morphology and growth characteristics |
|---|---|
| Mycoplasma contamination | All lines tested negative for mycoplasma contamination |
| Commonly misidentified lines (See ICLAC register) | None |

# Animals and other research organisms

Policy information about studies involving animals; ARRIVE guidelines recommended for reporting animal research, and Sex and Gender in Research

| Laboratory animals | Female immunodeficiency NOD-SCID mice (~10 weeks old) were used for teratoma assays. Mice were housed in 12-hr light/12-hr dark cycle 22.1–22.3 °C and 33–44% humidity. |
|---|---|
| Wild animals | No wild animals were used in the study. |
| Reporting on sex | Female mice were used in teratoma assays |
| Field-collected samples | No field collected samples were used in the study. |
| Ethics oversight | UT Southwestern Institutional Animal Care and Use Committee (IACUC): APN-2018-102430 |

Note that full information on the approval of the study protocol must also be provided in the manuscript.

# Plants

| Seed stocks | *Report on the source of all seed stocks or other plant material used. If applicable, state the seed stock centre and catalogue number. If plant specimens were collected from the field, describe the collection location, date and sampling procedures.* |
|---|---|
| Novel plant genotypes | *Describe the methods by which all novel plant genotypes were produced. This includes those generated by transgenic approaches, gene editing, chemical/radiation-based mutagenesis and hybridization. For transgenic lines, describe the transformation method, the number of independent lines analyzed and the generation upon which experiments were performed. For gene-edited lines, describe the editor used, the endogenous sequence targeted for editing, the targeting guide RNA sequence (if applicable) and how the editor was applied.* |
| Authentication | *Describe any authentication procedures for each seed stock used or novel genotype generated. Describe any experiments used to assess the effect of a mutation and, where applicable, how potential secondary effects (e.g. second site T-DNA insertions, mosiacism, off-target gene editing) were examined.* |

# Magnetic resonance imaging

## Experimental design

| Design type | Retrospective study. |
|---|---|
| Design specifications | Retrospective review of neuroimaging MRI studies initially acquired for clinical purposes. |
| Behavioral performance measures | N/A |

## Acquisition

| Imaging type(s) | MRI |
|---|---|
| Field strength | 1.5 and 3.0 TESLA |
| Sequence & imaging parameters | Due to the number of participating centers in the patient cohort, there was significant heterogeneity in terms of scanner manufacturer, sequences acquired, and imaging parameters. Minimum MR imaging sequences for inclusion were axial T1WI and axial T2WI, all with ≤5-mm section thicknesses. Additional sequences including T2 FLAIR, SWI, DWI/DTI, and gradient recalled echo were reviewed in most cases (when available). |
| Area of acquisition | Brain |

Diffusion MRI  ☒ Used  ☐ Not used

Parameters *Specify # of directions, b-values, whether single shell or multi-shell, and if cardiac gating was used.*

## Preprocessing

| | |
|---|---|
| Preprocessing software | N/A |
| Normalization | *If data were normalized/standardized, describe the approach(es): specify linear or non-linear and define image types used for transformation OR indicate that data were not normalized and explain rationale for lack of normalization.* |
| Normalization template | *Describe the template used for normalization/transformation, specifying subject space or group standardized space (e.g. original Talairach, MNI305, ICBM152) OR indicate that the data were not normalized.* |
| Noise and artifact removal | *Describe your procedure(s) for artifact and structured noise removal, specifying motion parameters, tissue signals and physiological signals (heart rate, respiration).* |
| Volume censoring | *Define your software and/or method and criteria for volume censoring, and state the extent of such censoring.* |

## Statistical modeling & inference

| | |
|---|---|
| Model type and settings | N/A |
| Effect(s) tested | N/A |

Specify type of analysis: ☒ Whole brain  ☐ ROI-based  ☐ Both

| | |
|---|---|
| Statistic type for inference | N/A |

(See Eklund et al. 2016)

| | |
|---|---|
| Correction | N/A |

## Models & analysis

| n/a | Involved in the study |
|---|---|
| ☒ ☐ | Functional and/or effective connectivity |
| ☒ ☐ | Graph analysis |
| ☒ ☐ | Multivariate modeling or predictive analysis |

