## [Peer Review File · Nature Cell Biology]

Synthetic ZFTA-fusions pinpoint disordered protein domain acquisition as a mechanism of brain tumorigenesis

Corresponding Author: Dr Stephen Mack

Version 0:

Decision Letter:

*Please delete the link to your author homepage if you wish to forward this email to co-authors.

Dear Dr Mack,

I am sorry for the delay. Your manuscript, "ZFTA fusion oncoproteins drive tumorigenesis through transcriptional condensates", has now been seen by 3 referees, who are experts in fusion oncoproteins (referee 1); cancer (referee 2); and phase separation (referee 3). As you will see from their comments (attached below) they find this work of potential interest, but have raised substantial concerns, which in our view would need to be addressed with considerable revisions before we can consider publication in Nature Cell Biology.

Nature Cell Biology editors discuss the referee reports in detail within the editorial team, including the chief editor, to identify key referee points that should be addressed with priority, and requests that are overruled as being beyond the scope of the current study. To guide the scope of the revisions, I have listed these points below. We are committed to providing a fair and constructive peer-review process, so please feel free to contact me if you would like to discuss any of the referee comments further.

I should stress that the referees' concerns point to unclear mechanistic links between phase separation, transcriptional activity, and tumorigenesis which would need to be addressed with experiments and data, and reconsideration of the study for this journal and re-engagement of referees would depend on strength of these revisions.

In particular, it would be essential to:

A) Further characterize phase separation behaviour of the ZR fusion oncoproteins, including assessing whether or not this requires other transcriptional modulators, IDRs, DNA binding (all Reviewers)

B) Assess whether these ZR condensates have any novel target genes and/or alterations in chromatin accessibility (Reviewers #2 and #3)

C) Provide further experimental evidence to assess whether or not ZR condensation and transcriptional activity has a causal link with tumorigenesis (Reviewers #2 and #3)

D) All other referee concerns pertaining to strengthening existing data, providing controls, methodological details, clarifications and textual changes, should also be addressed.

E) Finally please pay close attention to our guidelines on statistical and methodological reporting (listed below) as failure to do so may delay the reconsideration of the revised manuscript. In particular please provide:

We would be happy to consider a revised manuscript that would satisfactorily address these points, unless a similar paper is published elsewhere, or is accepted for publication in Nature Cell Biology in the meantime.

- ensure that it conforms to our format instructions and publication policies (see below and www.nature.com/nature/authors/).

- provide a point-by-point rebuttal to the full referee reports verbatim, as provided at the end of this letter.

- provide the completed Editorial Policy Checklist (found here <https://www.nature.com/authors/policies/Policy.pdf>), and Reporting

Summary (found here https://www.nature.com/authors/policies/ReportingSummary.pdf). This is essential for reconsideration of the manuscript and these documents will be available to editors and referees in the event of peer review. For more information see http://www.nature.com/authors/policies/availability.html or contact me.

Nature Cell Biology is committed to improving transparency in authorship. As part of our efforts in this direction, we are now requesting that all authors identified as 'corresponding author' on published papers create and link their Open Researcher and Contributor Identifier (ORCID) with their account on the Manuscript Tracking System (MTS), prior to acceptance. ORCID helps the scientific community achieve unambiguous attribution of all scholarly contributions. You can create and link your ORCID from the home page of the MTS by clicking on 'Modify my Springer Nature account'. For more information please visit please visit www.springernature.com/orcid.

Link Redacted

We would like to receive a revised submission within six months. We would be happy to consider a revision even after this timeframe, however if the resubmission deadline is missed and the paper is eventually published, the submission date will be the date when the revised manuscript was received.

We hope that you will find our referees' comments, and editorial guidance helpful. Please do not hesitate to contact me if there is anything you would like to discuss.

Best wishes,

Daryl

Daryl Jason Verzosa David, PhD

Senior Editor, Nature Cell Biology
Advisory Editor, npj Biological Physics and Mechanics
Nature Portfolio

Heidelberger Platz 3, 14197 Berlin, Germany
Email: daryl.david@nature.com
ORCID: <https://orcid.org/0000-0002-9253-4805>

Reviewers' Comments:

Reviewer #1:

Remarks to the Author:

In this work, Arabzade et al. provide a detailed characterization of the molecular mechanisms of the ZFTA-RELA (ZR) fusion oncoprotein, which is a driver of ependymoma. While the endogenous RELA and ZFTA proteins show distinct localization patterns, the ZF fusion oncoprotein localizes to nuclear punctae, which are characteristic of transcriptional condensates, as has been shown for other cancer-causing fusion oncoproteins. These punctae recapitulate some hallmarks of phase-separated condensates, such as dynamic fusion and quick recovery in FRAP experiments. Within these structures, the ZR fusion oncoproteins co-localizes with transcriptional regulators and with nascent mRNA of ZR target genes. Mutational analysis of predicted IDRs in the ZR fusion oncoprotein reveals IDR3 (in the RELA moiety of the fusion oncoprotein) and to a lesser extent also IDR2 (spanning the breakpoint region of the fusion oncoprotein) as important for the characteristic localization pattern of ZR. Graded mutagenesis of residues within IDR3 that contribute to multivalent interactions to negatively charged amino acids gradually decreased the formation of ZR nuclear punctae. Loss as well as mutation of IDR3 caused a decrease in ZR DNA binding and target gene induction. Furthermore, the authors identify a Zinc Finger in the ZFTA moiety of the ZR fusion as necessary for condensate formation, association with target genes and induction of gene expression. Finally, they show that other ZFTA fusion oncoproteins show similar localization patterns, and the extent of gene dysregulation is correlated with their ability to form nuclear speckles.

This is an interesting study proposing that biomolecular condensation of the ZFTA-RELA fusion oncoprotein is required for its oncogenic activity. While most of the conclusions are supported by the data, the work suffers from several shortcomings that diminish the general relevance of the findings.

Major points:

- While the cellular phenotypes of ZR localization are consistent with the formation of phase-separated condensates, this conclusion is not fully supported in the absence of results that directly prove that the ZR fusion oncoprotein undergoes phase separation, either alone or with the transcriptional modulators it interacts with in cells, or with chromatin/DNA. More direct evidence for this should be provided, e.g. by performing in vitro LLPS assays with wild-type vs. IDR-/ZnF-mutant variants of ZR.

- It is known that protein localization in general, and phenomena like biomolecular condensation in particular are highly dependent on cellular protein levels. Most of the work is performed in overexpression systems, and no controls for protein levels are shown. As this questions the general relevance of their findings, the authors should provide information about the levels of transfected proteins throughout all figures of the manuscript. Furthermore, several of the results should be validated in patient-derived cells/cell lines that express endogenous ZR, such as the interaction of the fusion oncoproteins with BRD4, MED1 and PolII in condensates.
- Endogenous ZFTA localizes to the nucleolus. As this organelle is also a phase-separated structure, it would be interesting to show (i) ZFTA dynamics in this compartment via FRAP and (ii) whether loss of any of the four Zinc Fingers in the ZFTA sequence alters localization and condensation properties of wild-type ZFTA.
- In the main text (111-112) the authors mention that “unexpectedly, full length ZFTA (G-ZFTA) was largely restricted to the nucleolus”, and this is taken as the normal behavior of the WT protein. Given that ZFTA has been reported to show a different subcellular localization in other cell types, have the authors considered that the eGFP tag may be modifying the behavior of the WT protein? An IF experiment in cells expressing the untagged protein could help to clarify this.
- The authors should also include some more representative pictures of the colocalization experiments shown in Figure 2 A-F in the Supplementary Data. Can they include a detailed image of ROIs with both channels merged?
- For the RNA-FISH experiments represented in Figure 2 K-N, only one representative picture is provided. Could the authors include some more in the Supplementary Material?
- While the work characterizes several structural features of the ZR fusion oncoproteins, there is no common way of representing them in the figures. It would make it easier for the reader to annotate Zinc Fingers and other domains in Figure 3A (which only contains the annotation of IDRs) and include the annotation of IDRs in Figures 1 and 5.
- More experimental data should be shown to further characterize the function of IDR2 in the ZR sequence. Deletion of this IDR also causes changes in ZR localization. Can the authors also perform the gradual mutagenesis approach to further map which residues in this IDR are responsible for the effect instead of deleting it altogether?
- Is IDR3 of the RELA sequence sufficient to cause condensation and target gene induction when fused to the ZFTA sequence (or just the ZFTA ZnF1)?
- A more detailed analysis of global chromatin binding of the ZR-ZF2A mutant is missing. Which transcription factor binding motifs are enriched in the CUT&Run data of wild-type vs. mutant ZR?
- What is the overlap of genes that are dysregulated by all ZFTA fusions studied in Figure 6, and which of them depend on ZnF1 in the ZFTA moiety?

Minor points:

- What is the staining in Figure S1B?
- Can the authors show a negative control staining for the RNA FISH experiment in Figure 2, such as an mRNA whose expression does not depend on ZR?
- Figure 4C shows that deletion of IDR3 causes loss of colocalization of ZR with MED1, BRD4 and PolII. Is there a global re-localization of these factors upon inactivation of ZR?
- What exactly is shown in Figure 4G? What does “gene expression” refer to?
- Figure S5G: include FRAP results of the ZR-RL3A mutant as a control.
- Figure S5D: the 15N-1H HSQC shows the chemical shift perturbation of ZF1+DNA, nevertheless the plot is too small to appreciate the cross-peaks properly. It would be useful to have an inset in the graph or provide more data to show the extent of this perturbation for the residues especially involved in the binding (those that have been marked as red and orange spheres in the main Figure 5B).
- The sentence in lines 249 – 250 is not supported by data.
- The sentence in lines 250 – 253 and the corresponding figure do not contribute relevant information and can be deleted.
- The manuscript does not contain any information about the role of the endogenous ZFTA protein. This could be included to provide context.
- In the description of Figure 5 (383-384), notes (G) and (F) are mislabeled.
- Scale bars and contours in some pictures are too small or missing. The format of scale bars and contours for the imaging data must be uniform and clearly visible in every picture.

Reviewer #2:

Remarks to the Author:

Arabzade and co-workers explore the mechanistic basis for the tumorigenic properties of the supratentorial ependymoma-associated

ZFTA-RELA (ZR) fusion oncoprotein. Using high-quality microscopy experiments, the investigators show that ZR forms dynamic nuclear assemblies that have properties similar to liquid-like condensates and these nuclear condensates are required for tumorigenesis. Mutagenesis of ZR revealed that one of three intrinsically disordered regions (IDR) was responsible for condensate formation and tumorigenesis. Condensate-modulating ZR IDR mutations impaired genomic occupancy at oncogenic loci, and inhibited the recruitment of transcriptional effector proteins, such as MED1, BRD4 and RNA polymerase II.

The experiments described in this study are generally straightforward and well-conducted. The investigators might want to consider the following queries:

- The investigators show that ZR condensates are associated with active transcription and are enriched at key sites of ZR-driven oncogene expression, but have they identified any novel genes so enriched?
- Although the investigators demonstrate that mutating three enriched amino acids in the IDR3 to negatively charged aspartic acid and glutamic acid decreased condensation formation, they need to do the same experiment mutating amino outside of the IDRs to aspartic acid and glutamic to prove this is an IDR-specific effect.
- Given that the IDR3-mut 25,50,75 had an apparent “dose effect” on transcriptomic changes (Fig. 4G), it would be very interesting to know whether these changes in DEGs reflected a quantitative change with only a relative change in the expression levels of the same ZR-associated genes, or whether there were specific genes whose regulation changed substantially between the various mutants.
- Were the tumorigenesis experiments done using the IDR3-75 mutant and if so, it would be important to know the effects of the IDR3-25,50 mutants on tumorigenesis (e.g. relative to the point above, this might also help identify yet to be identified tumorigenesis - dependent genes).
- The negatively charged aspartic acid and glutamic acids that were placed in the IDR3 mutants would be predicted to be potentially significantly disruptive to a number of other protein-protein/DNA interactions. So, how do we know that the lack of transcription factor binding to sites of ZR DNA binding is not a direct disruption of DR-TF interactions rather than having to do with condensate formation? What do negatively charged mutations in the other IDRs and in non-IDR locations in then ZR protein do to TF binding and on the overall ZR-associated transcriptomic signature?
- To the above point, and more generally, the authors show that IDR3 mutagenesis disrupts (to a variable extent) condensate binding and abrogates tumorigenesis, but they have not shown that these two phenomenon are related. Can the investigators determine a way to artificially restore condensate formation with the ZR-IDR3 mutant, thereby demonstrating the direct link between tumorigenesis, genomic transcriptomic signatures, and condensate formation (e.g. possibly using a condensation-dependent IDR from another oncogenic fusion protein fused to the IDR3-mut construct – or possible even used to substitute for the IDR3 thereby obviating the possibility of direct TF association with the ZR-IDR3 was a mechanism as described above)?
- Since the ZR-IDR3mut apparently does not significant effect DNA binding it would be important to do an extensive analysis of its ability (and that of the other IDR mutations) to change genomic chromatin accessibility (e.g. ATAC-seq).
- It is not surprising that mutation of the ZFTA zinc finger domain in ZR inhibited ZR DNA binding and in doing so abrogated the transcriptomic and tumorigenic properties of ZR. It is, however, a bit surprising that the zinc-finger mutant, ZR-ZF2A, abolished the formation of condensates. How does this jive with the identification of the ZR IDR3 to be required to form condensates? Is oncogenic fusion protein binding to DNA a requisite for condensate formation – cannot just proteins with the correct IDRs form condensates, regardless of their stability to bind DNA?
- Do physical means such as osmotic stress effect the ability of the IDR3-mut to form condensates?

Finally, and most importantly, one must question the novelty of this manuscript in that there are a growing number of reports demonstrating the necessity of IDR-mediated condensate formation for the tumorigenic role of various oncogenic fusion proteins – including the Yap-fusion proteins role in supratentorial ependymomas as reported in Nature Cell Biology just a few months ago (reference #24). Thus, even should the authors choose to perform more extensive mechanistic studies as suggested above, the novelty of the major observations reported in this manuscript are modest at best and rather predictable given the current state of the literature, thereby not warranting publication in as high impact a journal as NCB in my opinion.

Reviewer #3:

Remarks to the Author:

Comments on “ZFTA fusion oncoproteins drive tumorigenesis through transcriptional condensates” by Mack, Kriwachi and colleagues

In this study, Arabzade et al. studies driver genes/proteins in ependymomas (EPNs), an aggressive brain tumor. These drivers are oncofusions due to chromosome translocation. Over 95% of EPNs are driven by a gene fusion involving the zinc finger translocation associated (ZFTA) protein, and less than 5% EPNs involves fusions with YAP1. The most frequent fusion partner with ZFTA is RELA and the fusion gene is called ZR. The molecular mechanisms of ZR in driving tumorigenesis are the focus of this study.

They firstly showed that ZR forms dynamic nuclear condensates reminiscent of transcription condensates in EPN models. Indeed the ZR condensates enrich with transcriptional machineries at ependymoma oncogenes including NOTCH1 and CCND1. Using mutagenesis analysis, they nailed an IDR, IDR3, responsible for condensation. By grossly mutating three types of enriched amino acid residuals (P, A, Q), they generated ZR variants with progressively weaker condensation capacity. Apparently condensation encoded within IDR3 of ZR is required for its chromatin binding, transcriptional activity and oncogenicity. Using NMR and AlphaFoldDB structural analysis, the authors designed mutation within ZFTA zinc finger domain (called ZF1) that abolish its DNA binding capacity. This mutation also loses condensation for oncogenic transcription, ZR’s condensation capacity appears to be general among oncofusions for EPNs as all other

known ZFTA fusions are predicted to form condensates and many are experimentally shown to form transcriptional condensates.

This study has convincingly showed that condensates by a number fusion genes are important drivers for oncogenicity in a class of solid tumor. Given that being said, a few crucial link is required for further substantiation. Please see my critiques below for details.

Major:

- 1) To establish the causality between condensation and transcriptional activation, oncogenicity, the authors used loss-of-function mutations such as deletion of IDR3 or global substitution of 25%, 50% or even 75% of enriched amino acids. It will substantially strengthen the story if the authors can carry out rescue-type of experiment. E.g. grafting orthogonal phase separation-competent IDRs into DeltaIDR3 or MUT25, MUT50, MUT75 and see whether transcriptional activation and oncogenicity are restored upon re-introducing condensation into the mutant ZR.
- 2) Please provide exact sequences of all fusion genes used in the study and do further analysis on the potential heterogeneity of the fusions of two identical genes. For example, do all ZFTA fusion genes in Figure 6 still contain ZF1? YAP1 doesn't bind DNA directly. It needs TEAD1-4 for DNA binding. Do YAP1-KDM2B and YAP-MAMLD1 still contain TEAD-binding domain? Figure 6D, the condensates of G-ZFTA-MAML2 look like homotypic condensates off chromatin. Does this fusion still contain DBD?
- 3) Figure 4H is not cited in the main text; Legends for Figure 2J-N are missing; Legends for Figure 5H are missing.

Minor:

- 1) Line 136, "Fig.2K, N" don't seem to support the text.
- 2) The fonts in figures are messy. Please use consistent fonts.
- 3) Line 168, it should be 3D instead of 3C.
- 4) Please provide exact sequences of IDR3-MUT25, IDR3-MUT50, IDR3-MUT75.
- 5) Paper citation needs more optimization. For example, Ref 13 is better re-cite at line 291. In addition, multiple nice papers on fusion oncoproteins are not cited. E.g. PMID: 33674598; PMID: 37400539; PMID: 29930090; PMID: 32929202.
- 6) Line 702, the protein concentration can't be 380 mM.

REFERENCES – are limited to a total of 70 for Articles, Resources, Technical Reports; and 40 for Letters. This includes references in the main text and Methods combined. References must be numbered sequentially as they appear in the main text, tables and figure legends and Methods and must follow the precise style of Nature Cell Biology references. References only cited in the Methods should be numbered consecutively following the last reference cited in the main text. References only associated with Supplementary Information

(e.g. in supplementary legends) do not count toward the total reference limit and do not need to be cited in numerical continuity with references in the main text. Only published papers can be cited, and each publication cited should be included in the numbered reference list, which should include the manuscript titles. Footnotes are not permitted.

Methods should be written concisely, but should contain all elements necessary to allow interpretation and replication of the results. As a guideline, Methods sections typically do not exceed 3,000 words. The Methods should be divided into subsections listing reagents and techniques. When citing previous methods, accurate references should be provided and any alterations should be noted. Information must be provided about: antibody dilutions, company names, catalogue numbers and clone numbers for monoclonal antibodies; sequences of RNAi and cDNA probes/primers or company names and catalogue numbers if reagents are commercial; cell line names, sources and information on cell line identity and authentication. Animal studies and experiments involving human subjects must be reported in detail, identifying the committees approving the protocols. For studies involving human subjects/samples, a statement must be included confirming that informed consent was obtained. Statistical analyses and information on the reproducibility of experimental results should be provided in a section titled "Statistics and Reproducibility".

All Nature Cell Biology manuscripts submitted on or after March 21 2016 must include a Data availability statement at the end of the Methods section. For Springer Nature policies on data availability see <http://www.nature.com/authors/policies/availability.html>; for more information on this particular policy see <http://www.nature.com/authors/policies/data/data-availability-statements-data-citations.pdf>. The Data availability statement should include:

- Accession codes for primary datasets (generated during the study under consideration and designated as "primary accessions") and secondary datasets (published datasets reanalysed during the study under consideration, designated as "referenced accessions"). For primary accessions data should be made public to coincide with publication of the manuscript. A list of data types for which submission to community-endorsed public repositories is mandated (including sequence, structure, microarray, deep sequencing data) can be found here <http://www.nature.com/authors/policies/availability.html#data>.
- Unique identifiers (accession codes, DOIs or other unique persistent identifier) and hyperlinks for datasets deposited in an approved repository, but for which data deposition is not mandated (see here for details <http://www.nature.com/sdata/data-policies/repositories>).
- At a minimum, please include a statement confirming that all relevant data are available from the authors, and/or are included with the manuscript (e.g. as source data or supplementary information), listing which data are included (e.g. by figure panels and data types) and mentioning any restrictions on availability.
- If a dataset has a Digital Object Identifier (DOI) as its unique identifier, we strongly encourage including this in the Reference list and citing the dataset in the Methods.

We recommend that you upload the step-by-step protocols used in this manuscript to [protocols.io](http://www.protocols.io). More details can be found at <https://www.protocols.io/help/publish-articles>.

All imaging data should be accompanied by scale bars, which should be defined in the legend.

Cropped images of gels/blots are acceptable, but need to be accompanied by size markers, and to retain visible background signal within the linear range (i.e. should not be saturated). The boundaries of panels with low background have to be demarked with black lines. Splicing of panels should only be considered if unavoidable, and must be clearly marked on the figure, and noted in the legend with a statement on whether the samples were obtained and processed simultaneously. Quantitative comparisons between samples on different gels/blots are discouraged; if this is unavoidable, it should only be performed for samples derived from the same experiment with gels/blots were processed in parallel, which needs to be stated in the legend.

- We do not recommend using Adobe Photoshop for designing figures, but we can accept Photoshop generated (.PSD or .TIFF) files only

if each element included in the figure (text, labels, pictures, graphs, arrows and scale bars) are on separate layers. All text should be editable in 'type layers' and line-art such as graphs and other simple schematics should be preserved and embedded within 'vector smart objects' - not flattened raster/bitmap graphics.

Unprocessed scans of all key data generated through electrophoretic separation techniques need to be presented in a supplementary figure that should be labelled and numbered as the final supplementary figure, and should be mentioned in every relevant figure legend. This figure does not count towards the total number of figures and is the only figure that can be displayed over multiple pages, but should be provided as a single file, in PDF or TIFF format. Data in this figure can be displayed in a relatively informal style, but size markers and the figure panels corresponding to the presented data must be indicated.

The total number of Supplementary Figures (not including the "unprocessed scans" Supplementary Figure) should not exceed the number of main display items (figures and/or tables (see our Guide to Authors and March 2012 editorial <http://www.nature.com/ncb/authors/submit/index.html#supinfo>; <http://www.nature.com/ncb/journal/v14/n3/index.html#ed>). No restrictions apply to Supplementary Tables or Videos, but we advise authors to be selective in including supplemental data.

GUIDELINES FOR EXPERIMENTAL AND STATISTICAL REPORTING

REPORTING REQUIREMENTS – To improve the quality of methods and statistics reporting in our papers we have recently revised the reporting checklist we introduced in 2013. We are now asking all life sciences authors to complete two items: an Editorial Policy Checklist (found here <https://www.nature.com/authors/policies/Policy.pdf>) that verifies compliance with all required editorial policies and a reporting summary (found here <https://www.nature.com/authors/policies/ReportingSummary.pdf>) that collects information on experimental design and reagents. These documents are available to referees to aid the evaluation of the manuscript. Please note that these forms are dynamic 'smart pdfs' and must therefore be downloaded and completed in Adobe Reader. We will then flatten them for ease of use by the reviewers. If you would like to reference the guidance text as you complete the template, please access these flattened versions at <http://www.nature.com/authors/policies/availability.html>.

STATISTICS – Wherever statistics have been derived the legend needs to provide the n number (i.e. the sample size used to derive statistics) as a precise value (not a range), and define what this value represents. Error bars need to be defined in the legends (e.g. SD, SEM) together with a measure of centre (e.g. mean, median). Box plots need to be defined in terms of minima, maxima, centre, and percentiles. Ranges are more appropriate than standard errors for small data sets. Wherever statistical significance has been derived, precise p values need to be provided and the statistical test used needs to be stated in the legend. Statistics such as error bars must not be derived from n<3. For sample sizes of n<5 please plot the individual data points rather than providing bar graphs. Deriving statistics from technical replicate samples, rather than biological replicates is strongly discouraged. Wherever statistical significance has been derived, precise p values need to be provided and the statistical test stated in the legend.

Version 1:

Decision Letter:

Our ref: NCB-A53952A

30th April 2025

Dear Dr. Mack,

I do once again apologize for the very long delay. As previously mentioned, unfortunately Reviewer #2 was unable to review your revisions. However, we had since approached Reviewer #3 to comment on your responses to Reviewer #2's previous concerns; Reviewer #3 has told us that they are satisfied with your responses to Reviewer #2's concerns.

Thank you for submitting your revised manuscript "Synthetic ZFTA-fusions pinpoint IDR acquisition as a mechanism of brain tumor development" (NCB-A53952A). It has now been seen by the original referees and their comments are below. The reviewers find that the paper has improved in revision, and therefore we'll be happy in principle to publish it in Nature Cell Biology, pending minor revisions to satisfy the referees' final requests and to comply with our editorial and formatting guidelines.

Thank you again for your interest in Nature Cell Biology Please do not hesitate to contact me if you have any questions.

Sincerely,
Daryl

Daryl Jason Verzosa David, PhD

Senior Editor, Nature Cell Biology
Advisory Editor, npj Biological Physics and Mechanics
Nature Portfolio

Heidelberger Platz 3, 14197 Berlin, Germany
Email: daryl.david@nature.com
ORCID: <https://orcid.org/0000-0002-9253-4805>

Reviewer #1 (Remarks to the Author):

The authors have done a fantastic job in addressing all points I have raised. The new results further strengthen the conclusions and significantly improve the quality of the work. My only suggestion would be to keep the original title of the manuscript. While I appreciate the relevance of the dataset on synthetic ZFTA fusions that is now included in Figure 7, most of the work deals with "natural" versions of ZFTA fusion oncoproteins. Therefore, I feel that the original title ("ZFTA fusion oncoproteins drive tumorigenesis through transcriptional condensates") appears better suited to describe the content of the work. I do not have any other requests and support publication of this manuscript in Nature Cell Biology.

Reviewer #3 (Remarks to the Author):

The authors have addressed my major concerns.

Version 2:

Decision Letter:

Dear Dr Mack,

I am pleased to inform you that your manuscript, "Synthetic ZFTA-fusions pinpoint disordered protein domain acquisition as a mechanism of brain tumorigenesis", has now been accepted for publication in *Nature Cell Biology*.

Over the next few weeks, your paper will be copyedited to ensure that it conforms to *Nature Cell Biology* style. Once your paper is typeset, you will receive an email with a link to choose the appropriate publishing options for your paper and our Author Services team will be in touch regarding any additional information that may be required.

Publication is conditional on the manuscript not being published elsewhere and on there being no announcement of this work to any media outlet until the online publication date in *Nature Cell Biology*.

Please note that *Nature Cell Biology* is a Transformative Journal (TJ). Authors may publish their research with us through the traditional subscription access route or make their paper immediately open access through payment of an article-processing charge (APC). Authors will not be required to make a final decision about access to their article until it has been accepted. [Find out more about Transformative Journals](https://www.springernature.com/gp/open-research/transformative-journals)

Authors may need to take specific actions to achieve compliance with funder and institutional open access mandates. If your research is supported by a funder that requires immediate open access (e.g. according to [Plan S principles](https://www.springernature.com/gp/open-science/plan-s-compliance) or the [NIH public access policy](https://www.springernature.com/gp/open-science/us-federal-agency-compliance)) then you should select the gold OA route, and we will direct you to the compliant route where possible. Because authors warrant under our subscription licensing terms that they haven't committed to licensing any version of their article under a licence inconsistent with the terms of our agreement – including the applicable embargo period – publication under the subscription model isn't suitable for authors whose funders require no embargo.

If you have not already done so, we strongly recommend that you upload the step-by-step protocols used in this manuscript to protocols.io (<https://protocols.io>), an open online resource that allows researchers to share their detailed experimental know-how. All uploaded protocols are made freely available and are assigned DOIs for ease of citation. Protocols and Nature Portfolio journal papers in which they are used can be linked to one another, and this link is clearly and prominently visible in the online versions of both. Authors who performed the specific experiments can act as primary authors for the Protocol as they will be best placed to share the methodology details, but the Corresponding Author of the present research paper should be included as one of the authors. By uploading your Protocols onto protocols.io, you are enabling researchers to more readily reproduce or adapt the methodology you use, as well as increasing the visibility of your protocols and papers. You can also establish a dedicated workspace to collect your lab Protocols. Further information can be found at <https://www.protocols.io/help/publish-articles>.

Nature Cell Biology encourages authors presenting evidence for cell, biological, molecular, and genetic interactions to consider communicating these findings using Biofactoid (<https://biofactoid.org/>). This tool helps users share a searchable representation of interactions (e.g. binding, gene expression, post-translational modification) between genes, gene products, or chemicals. Information added to Biofactoid, with author attribution, is shared on social media and public databases, such as Pathway Commons, where it can be discovered and analyzed in the context of a large and growing corpus of knowledge.

With kind regards,

Daryl

Daryl Jason Verzosa David, PhD

Senior Editor, Nature Cell Biology
Advisory Editor, npj Biological Physics and Mechanics
Nature Portfolio

Heidelberger Platz 3, 14197 Berlin, Germany
Email: daryl.david@nature.com
ORCID: <https://orcid.org/0000-0002-9253-4805>

** Visit the Springer Nature Editorial and Publishing website at http://editorial-jobs.springernature.com?utm_source=ejp_NCB_email&utm_medium=ejp_NCB_email&utm_campaign=ejp_NCB for more information about our career opportunities. If you have any questions please click [here](mailto:editorial.publishing.jobs@springernature.com).

Dear Dr. David,

Thank you for the opportunity to address reviewer and editorial comments in a revised manuscript to *Nature Cell Biology*. In the response letter, you will find our point-by-point response to the major editorial comments, as well as concerns raised by Reviewers 1 to 3.

A) EDITORIAL MAJOR COMMENTS

A) Further characterize phase separation behaviour of the ZR fusion oncoproteins, including assessing whether or not this requires other transcriptional modulators, IDRs, DNA binding (all Reviewers)

Our study primarily focuses on the key role of IDR3 in mediating ZR oncogenic function. In response we have successfully expressed and purified IDR3 in E.coli cells, as well as two IDR loss-of-function mutants, ZR-IDR3-75DE and ZR-IDR3-75GS (**Figure 3, Extended Data 4**). Purified IDR3 proteins were subject to liquid-liquid phase separation (LLPS) assays. In these studies, we found that ZR-IDR3 is capable of self-associating through homotypic interactions and forming *in vitro* condensates independent of the presence of DNA or other transcriptional modulators. LLPS characterization of ZR-IDR3 mutants led to phenotypically different results, with a loss of condensate forming capability in the ZR-IDR3-75DE mutant, and formation of protein aggregates in the ZR-IDR3-75GS mutant. These findings support our conclusion that the ZR gene fusion leads to aberrant nuclear condensate formation important for oncogene expression and tumor development.

B) Assess whether these ZR condensates have any novel target genes and/or alterations in chromatin accessibility (Reviewers #2 and #3)

In response, we have performed gene expression analysis of unmutated ZR protein as compared to mutant variants in which IDR3 underwent graded mutagenesis (**Figure 4I, Extended Data Figure 5F**). Unsupervised hierarchical clustering revealed a distinct gene set ('cluster 2') that was shown to be highly associated with nuclear condensate formation. Many genes in 'cluster 2' are over-represented by ZR target genes as expected, however we also identified novel pathways enriched in glial cell differentiation and synaptic transmission. Many of these targets, particularly those in chemical transmission, have been reported as important regulators of ependymoma tumor progression, however the mechanisms regulating their expression are largely unclear (*Chen et al. 2024, Nature*) [5]. We provide these new data and analysis that demonstrates novel target genes and pathways that were uncovered by investigating gene expression programs impacted by nuclear condensate formation.

Reviewers also raised an important point about the ability of ZR to alter chromatin accessibility. In response, we performed ATAC-seq on neural stem cells (NSCs) expressing i) ZR unmutated, ii) ZR- Δ IDR3, iii) ZR-ZFD2A, iv) ZR-IDR3-25DE, v) ZR-IDR3-50DE, and vi) ZR-IDR3-75DE. We did not identify significant changes in chromatin accessibility programs between parental NSCs and those expressing ZR unmutated or mutated variants (**Response Figure 20**). This is in line with our hypothesis that ZR engages a very specific developmental chromatin accessibility programs and likely maintains expression of cell proliferative genes (i.e. *CCND1* and *NOTCH1*) during the process of tumor initiation. Our findings provide a potential mechanistic link between condensates formed at key brain developmental loci, and connect findings from this manuscript to a related pre-print manuscript currently under review: Please see pre-print:

<https://www.biorxiv.org/content/10.1101/2024.08.12.607603v1.full> [1].

C) Provide further experimental evidence to assess whether or not ZR condensation and transcriptional activity has a causal link with tumorigenesis (Reviewers #2 and #3)

In response to this major comment, we have performed a series of new *in vivo* experiments to test the ability of ZR-IDR3-DE and ZR-IDR3-GS mutants to form brain tumors in mice as compared to unmutated ZR controls (**Figure 4J-L, Extended Data Figure 6A-D**). *In vivo* experiments were conducted using an *In Utero Electroporation* (IUE) model of ZR murine ependymoma, in which radial glial cells were electroporated with piggyBAC constructs to stably express fusion oncoproteins during embryonic brain development at day E16.5 [2]. Critically, ZR-IDR3-75DE and ZR-IDR3-75GS mutants that disrupt condensate formation and oncogene transcription completely abrogated tumor initiation. This was in contrast with IUE of unmutated ZR protein that led to brain tumors formed with high penetrance.

While these *in vivo* experiments were loss-of-function studies, in our revised manuscript we have since made synthetic ZFTA fusions, replacing the IDR3 region of ZR with IDRs from other proteins known to drive nuclear condensate formation (e.g., *EWSR1* and *FUS*) (**Figure 7**). These synthetic mutants have never been described before in biomedical research. Critically, synthetic fusions that formed nuclear condensates, similar to ones formed by ZFTA-RELA, were able to drive expression of known ZFTA-RELA target genes and could initiate brain tumors *in vivo* with a similar transcriptional signature compared to ZR ependymoma. (**Figure 7**). To our knowledge, we are amongst the first to investigate the impact of IDR 'replacement' as a gain-of-function approach to understand the impact of nuclear condensates on transcription and brain tumor initiation.

B) RESPONSE TO REVIEWER #1:

Remarks to the Author: In this work, Arabzade et al. provide a detailed characterization of the molecular mechanisms of the ZFTA-RELA (ZR) fusion oncoprotein, which is a driver of ependymoma. While the endogenous RELA and ZFTA proteins show distinct localization patterns, the ZF fusion oncoprotein localizes to nuclear punctae, which are characteristic of transcriptional condensates, as has been shown for other cancer-causing fusion oncoproteins. These punctae recapitulate some hallmarks of phase-separated condensates, such as dynamic fusion and quick recovery in FRAP experiments. Within these structures, the ZR fusion oncoproteins co-localizes with transcriptional regulators and with nascent mRNA of ZR target genes. Mutational analysis of predicted IDRs in the ZR fusion oncoprotein reveals IDR3 (in the RELA moiety of the fusion oncoprotein) and to a lesser extent also IDR2 (spanning the breakpoint region of the fusion oncoprotein) as important for the characteristic localization pattern of ZR. Graded mutagenesis of residues within IDR3 that contribute to multivalent interactions to negatively charged amino acids gradually decreased the formation of ZR nuclear puncta. Loss as well as mutation of IDR3 caused a decrease in ZR DNA binding and target gene induction. Furthermore, the authors identify a Zinc Finger in the ZFTA moiety of the ZR fusion as necessary for condensate formation, association with target genes and induction of gene expression. Finally, they show that other ZFTA fusion oncoproteins show similar localization patterns, and the extent of gene dysregulation is correlated with their ability to form nuclear speckles.

This is an interesting study proposing that biomolecular condensation of the ZFTA-RELA fusion oncoprotein is required for its oncogenic activity. While most of the conclusions are supported by the data, the work suffers from several shortcomings that diminish the general relevance of the findings.

MAJOR POINTS

Reviewer 1, Point 1:

While the cellular phenotypes of ZR localization are consistent with the formation of phase-separated condensates, this conclusion is not fully supported in the absence of results that directly prove that the ZR fusion oncoprotein undergoes phase separation, either alone or with the transcriptional

modulators it interacts with in cells, or with chromatin/DNA. More direct evidence for this should be provided, e.g. by performing in vitro LLPS assays with wild-type vs. IDR-/ZnF-mutant variants of ZR.

Response:

We thank the reviewer for this comment. We agree that *in vitro* liquid-liquid phase separation (LLPS) assays would support the conclusions of our manuscript. These assays require expression and purification of ZR protein at a sufficient concentration, which can be challenging for large and highly disordered proteins. To this end, we performed the following experiments to attempt to express and purify full length GFP tagged ZR in prokaryotic and eukaryotic cells.

(1) Bacterial Studies: Recombinant proteins are commonly expressed in *E. coli* to generate sufficient quantities of protein for experimental studies. A major obstacle we encountered was that full-length ZR protein was proteolyzed during expression.

(2) Mammalian Studies: In parallel, we engineered N- and C- terminal Halo tagged ZR and expressed the construct at large scale in HEK293T cell liquid culture. Following successful expression, Halo-tagged ZR was covalently bound to HaloLink Resin (Promega) for purification, that was validated by western blot and mass spectrometry. We then performed a series of high stringency washes consisting of up to 6 high salt (500 mM NaCl) washes.

Unfortunately, due to the native ability of ZR to bind tightly to other interacting proteins we were unable to purify ZR protein alone with high stringency (a purity of less than 15%) (**Response Figure 1A**). Mass-spec analysis of the purified protein sample revealed binding of heat shock and tubulin protein

contaminants (**Response Figure 1B**). While the eluted ZR protein (and contaminants) can be subjected to LLPS assays, we are not confident in the interpretation of these findings that probe ZR phase separation capacity, specifically. As a further attempt, we could improve purification with the addition of a high concentration of urea, however this involved denaturing the contaminating proteins and ZR itself (data not shown). We are not confident that the ZR protein refolds properly following this harsh treatment, and hesitant to draw conclusions about ZR phase separation from these experiments.

Since our study prioritized IDR3 of ZR as having a significant role in condensate formation and gene regulation, we focused on expression and purification of IDR3 in *E. coli*. This was successful resulting in purified ZR-IDR3 unmutated and mutated variants denoted ZR-IDR3-75DE and ZR-IDR3-75-GS

Response Figure 1: (A) Gel electrophoresis analysis of N terminally Halo tagged ZFTA-RELA purified from HEK293T cells. Proteins were detected using comassie staining. (B) Mass-spec analysis of protein eluent after 6 high salt washes. GFP tagged ZFTA-RELA is marked by red. (C) Representative images of in vitro droplet assays performed using ZR-IDR3 unmutated, ZR-IDR3-75DE or ZR-IDR3-75GS. Coomassie stained protein gel image of each eluent is provided on the left side to indicate the purity of protein samples.

(Response Figure 1C, Figure 3I, Extended Data Figure 4G). LLPS assays were performed demonstrating that unmutated ZR-IDR3 exhibits homotypic (self-self) phase separation in the presence of the molecular crowder, PEG 8k, while ZR-IDR3-75DE loses this capability at similar protein concentrations. ZR-IDR3-75GS also lost phase separation propensity and instead formed protein aggregates. These experiments provide additional support for the key role of IDR3 in mediating liquid-liquid phase separation of ZFTA-RELA.

Reviewer 1, Point 2:

It is known that protein localization in general, and phenomena like biomolecular condensation in particular are highly dependent on cellular protein levels. Most of the work is performed in overexpression systems, and no controls for protein levels are shown. As this questions the general relevance of their findings, the authors should provide information about the levels of transfected proteins throughout all figures of the manuscript. Furthermore, several of the results should be validated in patient-derived cells/cell lines that express endogenous ZR, such as the interaction of the fusion oncoproteins with BRD4, MED1 and PolII in condensates.

Response:

This is a valid point raised by *Reviewer #1*. In response, we provide western blot analysis demonstrating that expression of HA tagged ZR in condensate imaging experiments is performed at ZR expression levels very similar to those seen in either patient derived xenograft (PDX) cell lines or ZR-driven patient tumors (**Response Figure 2**). A major hurdle to detection of ZR condensates in native models has been the lack of ZR fusion specific antibodies. While endogenous tagging is theoretically possible, ZR gene fusions often arise by chromothripsis that results in significant genomic instability of the ZR locus[3]. In our hands, this has made it very challenging to knock-in GFP tags into the endogenous locus of ZR specifically.

Response Figure 2: Expression comparison between mNSC stably expressing ZFTA-RELA, Patient-derived xenograft model (1425 cell line) and ZR ependymoma patient tumor sample. ZFTA-RELA protein is identified using an antibody against RELA(p65).

Reviewer 1, Point 3:

Endogenous ZFTA localizes to the nucleolus. As this organelle is also a phase-separated structure, it would be interesting to show (i) ZFTA dynamics in this compartment via FRAP and (ii) whether loss of any of the four Zinc Fingers in the ZFTA sequence alters localization and condensation properties of wild-type ZFTA.

Response:

We thank the reviewer for this comment. We have included in our manuscript fluorescence recovery after photobleaching (FRAP) studies of the ZFTA wildtype (WT) protein. In this experiment, we expressed a GFP tagged ZFTA protein, at its full length containing 4 zinc finger (ZF) domains, in human neural stem cells (hNSCs) and show that ZFTA is in active equilibrium with the surrounding nuclear environment as evidenced by FRAP assays (**Response Figure 3A-B, Extended Data**

Figure 1B-C). These findings support the dynamic and liquid-like nature of ZFTA protein within the nucleolus. Given that the first zinc finger is always included in ZFTA fusion oncoproteins (and is necessary for ZFTA-RELA chromatin binding and tumorigenicity), we tested its role in nucleolar localization of ZFTA protein. The first zinc finger was deleted from the ZFTA gene and hNSCs were transfected with the ZF1 deletion construct (denoted ZFTA full length- Δ ZF1). ZFTA full length- Δ ZF1 protein was then detected by immunofluorescence assays using an HA antibody and the nucleolus was identified with endogenous NPM1 immunostaining. We observed that ZFTA- Δ ZF1 retained its nucleolar localization and in some cases was found in smaller subnuclear structures that did not overlap with NPM1 (**Response Figure 3C**). These findings provide evidence that wildtype ZFTA is localized to the nucleolus and has dynamic liquid-like behavior, a characteristic that is not dependent on the ZF1 domain.

Response Figure 3: (A) FRAP analysis of G-ZFTA full length in hNSC. Representative images of pre-bleach, bleach and post-bleach are provided. (B) Normalized relative fluorescence intensity of G-ZFTA full length is provided over time following bleach. Values are averaged over 23 cells. (C) Representative images of hNSC transiently expressing ZFTA full length Δ ZF1. ZFTA full length Δ ZF1 is detected via anti HA tag staining and NPM1 staining mark nucleolus.

Reviewer 1, Point 4:

In the main text (111-112) the authors mention that “unexpectedly, full length ZFTA (G-ZFTA) was largely restricted to the nucleolus”, and this is taken as the normal behavior of the WT protein. Given that ZFTA has been reported to show a different subcellular localization in other cell types, have the authors considered that the eGFP tag may be modifying the behavior of the WT protein? An IF experiment in cells expressing the untagged protein could help to clarify this.

Response:

We apologize for the poor choice of words, regarding the “unexpected” behavior of full length ZFTA which has not yet been thoroughly investigated in biomedicine. The reviewer raises an important point that the addition of an enhanced green fluorescence protein (eGFP) tag could modify the behavior of the ZFTA-full length protein. Given the lack of a robust and validated commercial antibody against ZFTA protein, specific staining for the endogenous ZFTA protein has been challenging to this date. To address this issue, we removed the eGFP tag and included an HA tag. Notably, HA tagging ZR protein does not measurably disrupt its oncogenic function (*Arabzade et al., Cancer Discovery, 2021; Zheng et al., Cancer Discovery, 2021*) [2, 4]. We expressed HA-tagged-ZFTA-full length in human NSCs and performed co-immunofluorescence staining against HA tag and NPM1 as a marker for nucleolus. In these new immunofluorescence assays, we found similar localization patterns for HA-tagged ZFTA and NPM1 demonstrating nucleolar co-localization of ZFTA full length protein, irrespective of the tag (**Response Figure 4, Extended Data Figure 1A**). We thank the reviewer for their important suggestions on improving the rigor of our experimental studies.

Reviewer 1, Point 5:

The authors should also include some more representative pictures of the colocalization experiments shown in Figure 2 A-F in the Supplementary Data. Can they include a detailed image of ROIs with both channels merged?

Response:

We have included more representative images in (**Response Figure 5A-C, Extended Data Fig.2A-C**). These new images include magnified images of regions of interest, showing each channel separately as well as channels merged.

Response Figure 5: More representative images of hNSC transiently expressing G-ZR and stained for (A) MED1, (B) BRD4 and (C) RNAPII. For each image, zoomed-in view at 4 loci is provided on the right side, including the G-ZR channel and the staining channel individually and merged.

Reviewer 1, Point 6:

For the RNA-FISH experiments represented in Figure 2 K-N, only one representative picture is provided. Could the authors include some more in the Supplementary Material?

Response:

Response Figure 6: More representative images Nascent RNA_FISH experiment in hNSC for **(A) NOTCH1** and **(B) CCND1**.

We thank the reviewer for this comment and have included more representative images in the supplementary data (**Response Figure 6A-B, Extended Data Figure 3C-D**).

Reviewer 1, Point 7:

While the work characterizes several structural features of the ZR fusion oncoproteins, there is no common way of representing them in the figures. It would make it easier for the reader to annotate Zinc Fingers and other domains in Figure 3A (which only contains the annotation of IDRs) and include the annotation of IDRs in Figures 1 and 5.

Response:

We apologize to the reviewer for the lack of clarity in our figures and appreciate the important suggestion to include schematics that more clearly present the structure of ZR with more detailed and

consistent annotation of ZFs and IDRs across figures. These elements have now been revised in the main manuscript across all main and extended figures.

Reviewer 1, Point 8:

More experimental data should be shown to further characterize the function of IDR2 in the ZR sequence. Deletion of this IDR also causes changes in ZR localization. Can the authors also perform the gradual mutagenesis approach to further map which residues in this IDR are responsible for the effect instead of deleting it altogether?

Response:

We thank *Reviewer #1* for this important comment. We observed that deletion of ZR-IDR1 leads to a complete loss of nuclear localization, possibly due to its proximity to the ZF1 domain. In line with this hypothesis, we have previously shown that deletion of the ZF1 domain impairs nuclear localization of the ZFTA-RELA protein (*Kupp et al, Cancer Discovery, 2021*) [5]. As suggested by *Reviewer #1*, we performed gradual mutagenesis of ZR-IDR1. An analysis of the IDR1 composition revealed that arginine, glycine and serine amino acids are overrepresented in IDR1, hence, these amino acids were mutated to negatively charged aspartic acid (D) and glutamic acid (E) (**Response Figure 7A**). We observed similar changes in subcellular localization for all three of the gradual mutants that resulted in cytosolic expression (**Response Figure 7B**). As a result, these new ZR-IDR1 mutants failed to activate oncogenic target gene expression (**Response Figure 7C**).

Deletion of ZR-IDR2 does not alter cellular localization nor does it significantly impact condensate formation, as compared to the deletion of IDR3. We performed similar gradual mutagenesis studies in ZR-IDR2 where the overrepresented amino acids, proline and glycine, were mutated to aspartic acid and glutamic acid (**Response Figure 8A-B**). 25% and 50% mutations had less pronounced impact on condensate formation, but 75% mutation led to changes in subcellular localization likely due to mutations proximal to the ZF1 region (data not shown). RNA-seq of cells expressing IDR2 gradual mutants also revealed a less pronounced effect on ZR target transcription as compared to IDR3 gradual and deletion mutants (**Response Figure 8C**).

Finally, to support these findings, we generated and characterized a new ZR mutant construct that expressed only *ZFTA* and *IDR3* from *RELA* (denoted ZR-IDR3-minimal). ZR-IDR3-minimal is capable of activating ZR oncogenic programs and initiating brain tumors in mice, albeit at a lower penetrance (**Response Figure 9, Extended Data Figure 6E-G**). These functional findings demonstrate that while ZR-IDR2 may still contribute to tumor development, its effects are not essential for oncogenic transcription and tumor initiation.

Response Figure 7: (A) Amino acid enrichment analysis of the IDR1. Highly enriched amino acids are marked with *. (B) Representative images of HEK293T cells expressing ZR unmutated or the indicated mutant. Proteins were visualized using immunofluorescence against HA tag. (C) IGF2 expression comparison between ZR unmutated and IDR1 gradual mutants using qRT-PCR.

Response Figure 8: (A) Amino acid enrichment analysis of the IDR2. Highly enriched amino acids are marked with *. (B) Representative images of HEK293T cells expressing the indicated mutant. Proteins were visualized using immunofluorescence against HA tag. (C) Expression comparison for ZR downstream target genes between ZR unmutated and IDR2 or IDR3 gradual mutants.

Reviewer 1, Point 9:

Is IDR3 of the RELA sequence sufficient to cause condensation and target gene induction when fused to the ZFTA sequence (or just the ZFTA ZnF1)?

Response:

This is an important point raised by *Reviewer #1*. In response, we have made a ZR-IDR3-minimal mutant (see *Point 8 above*) that encodes the ZFTA sequence from ZFTA-RELA (e.g., including IDR1, ZF1 and a small portion of IDR2) fused to IDR3 alone. This construct forms nuclear condensates, activates ZR target genes, and is sufficient to drive tumor formation *in vivo* (**Response Figure 9, Extended Data Figure 6E-G**). These experiments further demonstrate the minimal components needed to form biomolecular condensates, activate oncogenic transcription, and initiate tumor formation. It is important to point out that, the ZFTA-IDR3 minimal mutant has a reduced incidence of tumor formation, suggesting a contributing role of ZR-IDR2 and/or other components of endogenous RELA. Alternatively, it is unclear whether joining ZFTA to IDR3 in a minimalistic approach may also impact function of these domains through improper folding or proper recruitment of necessary interacting proteins. These latter two factors - technical in nature - may confound such biological interpretations.

Reviewer 1, Point 10:

A more detailed analysis of global chromatin binding of the ZR-ZF2A mutant is missing. Which transcription factor binding motifs are enriched in the CUT&RUN data of wildtype vs. mutant ZR?

Response:

We have included a detailed global chromatin binding and transcription factor motif analysis of the ZR-ZF2A mutant as compared to unmutated ZR (**Response Figure 10**). Notably, ZR-ZF2A has very few binding sites across the genome, and DNA motifs detected are not confidently supported. This contrasts with ZR unmutated that consistently enriches for the Plag family TF motif ('GGGCC'), which we have validated is bound by ZF1 of ZFTA in cognate site identification and fluorescence polarization assays (**Figure 5A-D, Extended Data Figure 7A-E**). Our findings suggest a model in which ZR does not effectively engage RELA sites; instead, endogenous RELA (that binds with ZR) is recruited to the nucleus to activate canonical RELA inflammatory target genes.

Reviewer 1, Point 11:

What is the overlap of genes that are dysregulated by all ZFTA fusions studied in Figure 6, and which of them depend on ZnF1 in the ZFTA moiety?

Response:

This is a great suggestion by *Reviewer #1* and in response we have identified genes consistently up-regulated by ZFTA fusion oncoprotein variants ($\text{LogFC} > 1$ $p\text{-value} < 0.05$). Of these, 369 genes overlapped with genes that were significantly under-expressed ($\text{LogFC} < 1$ & $p\text{-value} < 0.05$) upon ZF1 mutation (**Response Figure 11**). Notably, these genes were enriched for GO terms such as monoamine transport, catecholamine regulation, and GPCR signaling. These findings are consistent with our previous study highlighting the role of neurotransmitter signaling in ZFTA EPN providing a

functional basis for expression of these key genes involved in tumor progression (*Chen et al. 2024, Nature*) [6].

Response Figure 11: Global analysis of all genes that are affected by ZFTA fusion variants and are dependent on ZF1. Key enriched pathways are shown on the left side.

MINOR POINTS

Reviewer 1, Point 12:

What is the staining in Figure S1B?

Response:

We apologize for the lack of clarity and thank the reviewer for this comment. There is no staining in Figure S1B because cells were electroporated with GFP-tagged ZFTA-RELA (denoted as G-ZR). This has been clarified in the figure legends and text of the manuscript.

Reviewer 1, Point 13:

Can the authors show a negative control staining for the RNA FISH experiment in Figure 2, such as an mRNA whose expression does not depend on ZR?

Response:

We thank *Reviewer #1* for this comment. To address this, we selected two genes not regulated by the ZR fusion but were reported to be expressed in neural stem cells. (A major hurdle, was that ZR binds to thousands of active sites/genes in the genome, making this selection difficult and potentially biased towards repressed genes). We designed RNA-FISH probes targeting the intronic regions of these selected genes (sequences provided in **Supplementary Table 4**). Human neural stem cells were transfected with GFP tagged ZR, and RNA-FISH experiments were performed as described in the methods section. Unfortunately, we could not rigorously identify loci within nuclei representing sites of transcription for these genes. It is possible that the expression of these genes was not high enough to detect reliable RNA FISH loci as compared to background/non-specific binding. We apologize for not being able to include these data in the revised manuscript.

Reviewer 1, Point 14:

Figure 4C shows that deletion of IDR3 causes loss of colocalization of ZR with MED1, BRD4 and PolII. Is there a global re-localization of these factors upon inactivation of ZR?

Response:

In response we compared the MED1 and BRD4 CUT&RUN enrichment peaks between unmutated ZR and ZR- Δ IDR3 (**Response Figure 12**). We found that expression of unmutated ZR led to the acquisition of many new MED1 and BRD4 genomic binding sites as compared to expression of ZR- Δ IDR3. ZR- Δ IDR3 also led to acquisition of a small proportion of BRD4 and MED1 binding sites not observed in unmutated ZR expressing tumors however we did not observe a global re-localization. We thank *Reviewer #1* for this insightful comment.

Response Figure 12: Venn diagrams showing the number of (A) MED1 or (B) BRD4 peaks that are shared or unique between ZR unmutated or ZR- Δ IDR3.

Reviewer 1, Point 15:

What exactly is shown in Figure 4G? What does “gene expression” refer to?

Response:

We apologize for lack of clarity and thank the reviewer for bringing this to our attention. “Gene expression” refers to Log2 fold change (LFC) in expression compared to GFP control.

Reviewer 1, Point 16:

Figure S5G: include FRAP results of the ZR-RL3A mutant as a control.

Response:

We thank the reviewer for this comment and have now included FRAP results of the ZR-RL3A mutant in the manuscript (**Response Figure 13, Extended Data Figure 8E-F**).

Response Figure 13: (A) FRAP analysis of G-ZR RELA3A in hNSC. Representative images of pre-bleach, bleach and post-bleach are provided. (B) Normalized relative fluorescence intensity of G-ZR RELA3A is provided over time following bleach. Values are averaged over 22 cells.

Reviewer 1, Point 17:

Figure S5D: the 15N-1H HSQC shows the chemical shift perturbation of ZF1+DNA, nevertheless the plot is too small to appreciate the cross-peaks properly. It would be useful to have an inset in the graph or provide more data to show the extent of this perturbation for the residues especially involved in the binding (those that have been marked as red and orange spheres in the main Figure 5B).

Response:

We thank the reviewer for this comment and have now included a much bigger version of this plot in the manuscript (**Extended Data Figure 7D**). The residues involved in binding are clearly marked in the plot.

Reviewer 1, Point 18:

The sentence in lines 249 – 250 is not supported by data.

Response:

We apologize for this error and have since included the appropriate figure callout.

Reviewer 1, Point 19:

The sentence in lines 250 – 253 and the corresponding figure do not contribute relevant information and can be deleted.

Response:

We have removed the corresponding figure and deleted the associated information from the manuscript text.

Reviewer 1, Point 20:

The manuscript does not contain any information about the role of the endogenous ZFTA protein. This could be included to provide context.

Response:

We thank *Reviewer #1* for this important comment. While the molecular function of ZFTA protein is poorly understood, particularly with respect to transcriptional control, its expression patterns have been described in important developmental ependymal cell types linked to ependymoma cellular origins [7]. We have now included in the introduction a discussion about the knowledge gaps in ZFTA-full length biology and provided background on our limited understanding of this protein.

Reviewer 1, Point 21:

In the description of Figure 5 (383-384), notes (G) and (F) are mislabeled.

Response:

We have corrected this mislabeling error and thank the reviewer for this correction.

Reviewer 1, Point 22:

Scale bars and contours in some pictures are too small or missing. The format of scale bars and contours for the imaging data must be uniform and clearly visible in every picture.

Response:

In response, we have provided images in the entire manuscript with visible scale bars.

C) RESPONSE TO REVIEWER #2:

Remarks to the Author: Arabzade and co-workers explore the mechanistic basis for the tumorigenic properties of the supratentorial ependymoma-associated ZFTA-RELA (ZR) fusion oncoprotein. Using high-quality microscopy experiments, the investigators show that ZR forms dynamic nuclear assemblies that have properties similar to liquid-like condensates and these nuclear condensates are required for tumorigenesis. Mutagenesis of ZR revealed that one of three intrinsically disordered regions (IDR) was responsible for condensate formation and tumorigenesis. Condensate-modulating ZR IDR mutations impaired genomic occupancy at oncogenic loci, and inhibited the recruitment of transcriptional effector proteins, such as MED1, BRD4 and RNA polymerase II.

The experiments described in this study are generally straightforward and well-conducted. The investigators might want to consider the following queries:

Reviewer 2, Point 1:

The investigators show that ZR condensates are associated with active transcription and are enriched at key sites of ZR-driven oncogene expression, but have they identified any novel genes so enriched?

Response:

We thank *Reviewer #2* for this comment. To address the point of novel gene regulation, we analyzed differentially expressed genes (DEGs) and performed unsupervised hierarchical clustering to identify subsets of genes with the strongest response to graded perturbation of ZR condensate formation (**Response Figure 14A-E, Figure 4H-I, Extended Data Figure 5F**). 'Cluster 2' represents the genes responsive to condensate perturbation. Many of these genes were enriched in oncogenic fusion targets such as *IGF2*, *CCND1*, and *NOTCH1*. Critically, novel pathways that were identified included glial cell differentiation and neurotransmitter transporter pathways that we have recently shown to be important in ZR ependymoma; however, gene regulatory mechanisms that govern such pathways are poorly defined [5]. These findings provide novel insights into new mechanisms that govern genes that regulate tumor-neuronal interactions in ependymoma and the processes of tumor progression[6]. Additionally, as suggested by *Reviewer #2*, we also found unique DEGs for each of the ZR-IDR3-25, 50, and 75DE mutants, suggesting distinct sensitivities to condensate modulation (**Response Figure 14D-E**). We thank *Reviewer #2* for suggesting more in-depth exploration of ZR condensate responsive genes that have led to new insights on gene regulatory mechanisms in ependymoma pathogenesis.

Reviewer 2, Point 2:

Although the investigators demonstrate that mutating three enriched amino acids in the IDR3 to negatively charged aspartic acid and glutamic acid decreased condensation formation, they need to do the same experiment mutating amino outside of the IDRs to aspartic acid and glutamic to prove this is an IDR-specific effect.

Response:

We appreciate this comment that introducing charged amino acid residues such as aspartic acid and glutamic acid could have general, charge-related effects that could influence interpretation of their effects on ZR IDR3 function and prioritization. Based on the suggested line of experiments, we were concerned that mutating amino acids outside of IDRs, would have confounding effects by impacting the proper folding of structured domains. In response, we generated a new series of ZR IDR3 mutants by substituting enriched IDR3 residues with glycine and serine instead of D/Es, which we

termed ZR-IDR-G/S mutants. Our rationale is that introduction of GS mutations preserved charge balance within IDR3 but still alters enriched amino acids that we hypothesize contribute to multivalent interactions underlying condensate formation. In a similar fashion as the ZR-IDR-D/E mutants, ZR-IDR3-G/S mutants disrupted condensate formation, decreased oncogenic transcription, and inhibited tumor initiation (**Response Figure 15, Extended Data Figure 4E-G,6A-D**). These orthogonal mutagenesis experiments highlight the significance of ZR-IDR3 in the formation of transcriptional condensates as a key function of the oncogenic protein.

To further evaluate the specificity of IDR3 in modulating condensate formation, we performed similar gradual mutagenesis in IDR1 and IDR2 as well. Our analysis revealed that in IDR1, arginine, glycine and serine were over-represented compared to the human proteome (**Response Figure 16A**). Similar to deletion of IDR1, all three gradual IDR1 mutants led to protein mis-localization and as a result failed to activate downstream target genes (**Response Figure 16B-C**). As for ZR-IDR2, we identified proline and glycine as the enriched amino acids and performed gradual mutagenesis on these residues (**Response Figure 17A**). 25% and 50% substitutions had minimal impact on condensate formation, however, 75% mutagenesis lead to protein mis-localization (**Response Figure 17B**). RNA-seq analysis supported these findings with reduced gene expression albeit not reaching the level of down-regulation seen in ZR-IDR3-75DE or ZR- Δ IDR3 mutants (**Response Figure 17C**). We thank *Reviewer #2* for this comment that has led to a series of revised experiments that we feel has strengthened the manuscript.

Response Figure 14: (A) Unsupervised hierarchical clustering analysis ZR regulated genes based on their response to gradual condensate perturbation. (B) Expression comparison of the condensate response genes (cluster 2 in panel A) between GFP control, ZR unmutated and gradual mutants. (C) pathway enrichment analysis of the condensate responsive genes. (D) Overall number of either unchanged, upregulated or downregulated upon expression of ZR IDR3 D/E mutants. (E) Venn diagrams indicating the number of genes that are unique to each of IDR3 D/E mutants or shared amongst them. A few example genes are noted from the unique pool of genes.

Response Figure 15: (A) Representative images of indicated protein expressed in HEK293T cells. (B) Number of condensates comparison between ZR unmutated and gradual G/S mutants. (C) Transcriptional activity between ZR unmutated and gradual G/S mutants. (D) Normalized *IGF2* expression comparison between ZR unmutated and gradual G/S mutants. (E) Western blot analysis indicating expression of each protein in the cells used for qRT-PCR analysis in panel D. (F) Time to tumor formation for ZR unmutated and ZR-IDR3-75G/S.

Response Figure 16: (A) Amino acid enrichment analysis of the IDR1. Highly enriched amino acids are marked with *. **(B)** Representative images of HEK293T cells expressing ZR unmutated or the indicated mutant. Proteins were visualized using immunofluorescence against HA tag. **(C)** IGF2 expression comparison between ZR unmutated and IDR1 gradual mutants using qRT-PCR.

Response Figure 17: (A) Amino acid enrichment analysis of the IDR2. Highly enriched amino acids are marked with *. **(B)** Representative images of HEK293T cells expressing indicated mutant. Proteins were visualized using immunofluorescence against HA tag **(C)** Expression comparison for ZR downstream target genes between ZR unmutated and IDR2 or IDR3 gradual mutants.

Reviewer 2, Point 3:

Given that the IDR3-mut 25,50,75 had an apparent “dose effect” on transcriptomic changes (Fig. 4G), it would be very interesting to know whether these changes in DEGs reflected a quantitative change with only a relative change in the expression levels of the same ZR-associated genes, or whether there were specific genes whose regulation changed substantially between the various mutants.

Response:

We thank the reviewer for this excellent suggestion. We analyzed differentially expressed genes (DEGs) between ZR IDR condensate mutants and performed unsupervised hierarchical clustering of the gene expression programs (**Response Figure 14, Figure 4H-I, Extended Data Figure 5F**). ZR target genes are sensitive to condensate modulation and decrease in expression as IDR mutagenesis is increased. As detailed in *Reviewer #2 Point 1*, we also observed novel sets of glial cell differentiation and neurotransmission associated genes that were predicted to be condensate regulated based on pathway enrichment analysis of RNA-seq data. In addition, we observed significant differentially expressed genes across the different, graded IDR mutants, suggesting potentially disparate thresholds for gene activation when IDR3 was mutated to different degrees (**Response Figure 14**).

Reviewer 2, Point 4:

Were the tumorigenesis experiments done using the IDR3-75 mutant and if so, it would be important to know the effects of the IDR3-25,50 mutants on tumorigenesis (e.g. relative to the point above, this might also help identify yet to be identified tumorigenesis -dependent genes)

Response:

We thank the reviewer for this suggestion that connects the cell biology and transcriptional impact of the IDR mutants to tumor formation. In response, we designed and synthesized the ZR-IDR3-DE-25%, 50%, and 75% mutants in piggyBac expression constructs, and performed *in vivo in utero* electroporation tumor initiation studies as described in Arabzade *et al.*, *Cancer Discovery* 2021. A total of 42 *in vivo* mouse experiments were performed. Expression of ZR unmutated initiates tumors with a median time to tumor formation of ~40 days. By comparison, ZR-IDR3-DE-25% forms tumors at a much later time point of ~110 days, and ZR-IDR3-50% and 75% completely fail to initiate tumorigenesis (**Response Figure 18, Figure 4J-L**). Furthermore, we performed longitudinal tracing of detected tumors by magnetic resonance imaging, which revealed that tumors formed by ZR unmutated grew at a significantly faster rate than those formed by ZR-IDR3-DE25% (**Response Figure 18C, Figure 4L**). Together, these data support that ZR condensate formation is essential for tumor formation, and a threshold for condensate formation and oncogene activation is needed to initiate ependymoma development *in vivo*.

Response Figure 18: Time to tumor formation for ZR unmutated and each gradual IDR3 D/E mutants. **(B)** Representative MRI images of age-matched mice harboring brain tumors generated with ZR unmutated or ZR-IDR3-25DE. **(C)** Tumor volume tracing using MRI images for tumors generated with ZR unmutated or ZR-IDR3-25DE.

Reviewer 2, Point 5:

The negatively charged aspartic acid and glutamic acids that were placed in the IDR3 mutants would be predicted to be potentially significantly disruptive to a number of other protein-protein/DNA interactions. So, how do we know that the lack of transcription factor binding to sites of ZR DNA binding is not a direct disruption of ZR-TF interactions rather than having to do with condensate formation? What do negatively charged mutations in the other IDRs and in non-IDR locations in the ZR protein do to TF binding and on the overall ZR-associated transcriptomic signature?

Response:

In response, we have established and characterized a new set of ZR IDR mutants by substitution with glycine and serine amino acids (denoted IDR G/S mutants) instead of aspartic or glutamic acids. Our rationale is that introduction of GS mutations preserved charge balance within IDR3 but still alters enriched amino acids that we hypothesize contribute to multivalent interactions underlying

condensate formation. We demonstrate that ZR-IDR3-G/S mutants: i) Disrupt ZR condensate formation, ii) Significantly impair ZR oncogenic activation, and iii) Impair/Delay tumor initiation (**Response Figure 15, Extended Data Figure 4E-F,6A-D**). These findings are aligned with the D/E mutants and support the key functional role of IDR3 in mediating ZR oncogenic capacity.

Introduction of D/E mutations in IDR1 significantly alters ZR nuclear localization, resulting in accumulation of ZR in the cytosol (**Response Figure 16B-C**). This is most likely due to the proximity of mutations to the ZF1, that is critical for nuclear localization and contains a potential cryptic nuclear localization signal (*Kupp et al., Cancer Discovery, 2021*). Based on this information, we are hesitant to introduce D/E mutations into non-IDR regions as this is likely to significantly impact the structure of folded domains and have unanticipated consequences upon ZR molecular function.

As for ZR-IDR2, we identified proline and glycine as the enriched amino acids and performed gradual mutagenesis on these residues (**Response Figure 17A**). 25% and 50% substitutions had minimal impact on condensate formation, however, 75% mutagenesis lead to protein mis-localization (**Response Figure 17B**). RNA-seq analysis supported these findings with reduced gene expression albeit not reaching the level of down-regulation seen in ZR-IDR3-75DE or ZR- Δ IDR3 mutants (**Response Figure 17C**).

These new data to support our prioritization of IDR3 as a key regulator of ZR condensate formation through the characterization of ZR-IDR3-G/S mutants. Our proposed model is that IDR-rich proteins such as BRD4, Mediator complex proteins, and transcription factors (TFs) are recruited to sites of ZR binding through IDR-driven interactions. As Reviewer #2 points out, there may be critical TFs that are recruited to ZR binding sites that have yet to be identified and functionally validated. Characterizing the proteomic landscape of TFs found in ZR condensates is currently a part of ongoing experiments and will be a future priority.

We thank *Reviewer #2* for their comments that have led to a more rigorous approach to studying the role of IDRs in ZR driven ependymoma.

Reviewer 2, Point 6:

To the above point, and more generally, the authors show that IDR3 mutagenesis disrupts (to a variable extent) condensate binding and abrogates tumorigenesis, but they have not shown that these two phenomena are related. Can the investigators determine a way to artificially restore condensate formation with the ZR-IDR3 mutant, thereby demonstrating the direct link between tumorigenesis, genomic transcriptomic signatures, and condensate formation (e.g. possibly using a condensation-dependent IDR from another oncogenic fusion protein fused to the IDR3-mut construct – or possible even used to substitute for the IDR3 thereby obviating the possibility of direct TF association with the ZR-IDR3 was a mechanism as described above)?

Response:

We thank *Reviewer #2* for this comment and their suggestions. Establishing causality between transcriptional condensate formation and tumorigenicity has been a challenge in the field, and to our knowledge, has not been rigorously addressed in gain-of-function type experiments. To provide further evidence for a link between condensate formation and tumorigenicity of ZR, we generated a series of synthetic fusions in which we substituted IDR3 of ZR with IDRs found within other condensate forming proteins (**Response Figure 19A-B, Figure 7A-B**). With the exception of *NCOA2*, these IDRs were derived from proteins that have never been reported to form gene fusions with *ZFTA* in human biology. New synthetic ZR mutants were denoted *ZFTA-sub-IDR* (i.e. *ZFTA-sub-EWS:IDR*).

Due to the synthetic nature of these experiments, many of the mutants we established resulted in aberrant nuclear localization or protein aggregation (**Response Figure 19C, Figure 7C**).

Response Figure 19: (A) Schematic cartoon of “IDR swapping” experiment from designing the synthetic fusions to downstream functional assessments (B) Schematic representation of the synthetic fusions. (C) Representative images of HEK293T cells expressing indicated synthetic fusion. (D) ZR downstream gene expression comparison between ZFTA-RELA and synthetic fusions (E) Representative MRI images of tumors generated using ZR-sub-FUS:IDR and ZR-sub-EWS:IDR synthetic fusions. (F) time to tumor detection between ZFTA-RELA, ZR-sub-EWS:IDR or ZR-sub-FUS:IDR synthetic fusions. (G) Global gene expression comparison for tumors generated using (G) ZR-sub-FUS:IDR or (H) ZR-sub-EWS:IDR compared to healthy brain tissue. ZR ependymoma signature genes are marked with red dots.

Only ZFTA-RELA IDR substitutions with *EWS* and *FUS* IDRs resulted in morphologically similar condensates appropriately localized to the nucleus (**Response Figure 19C, Figure 7C**). These mutants denoted *ZFTA-sub-EWS:IDR* and *ZFTA-sub-FUS:IDR* were advanced for further evaluation. *ZFTA-sub-EWS:IDR* and *ZFTA-sub-FUS:IDR* were able to establish similar transcriptional programs as the *ZFTA-RELA* (**Response Figure 19D-G, Figure 7D-G**). Furthermore, we were able to generate mouse ependymoma brain tumors with these synthetic fusions with similar transcriptional profiles as tumors generated with *ZFTA-RELA* (**Response Figure 19E-G, Figure 7E-G, Extended Data Figure 10A**).

To our knowledge, this is the first time gain-of-function rescue experiments have been performed by grafting IDRs into fusion oncoproteins, which both restore condensate formation, molecular function, and tumorigenic capacity. We are grateful to *Reviewer #2* for suggesting this important line of experiments that provide further support of the potential causal link between IDRs, condensate formation, and fusion oncoprotein biology

Reviewer 2, Point 7:

Since the ZR-IDR3mut apparently does not significant effect DNA binding it would be important to do an extensive analysis of its ability (and that of the other IDR mutations) to change genomic chromatin accessibility (e.g. ATAC-seq).

Response:

To address this comment, we generated neural stem cells stably expressing unmutated ZR, ZFD2A and three ZR-IDR3-25/50/75DE mutants and performed ATAC-seq (**Response Figure 20**). ATAC-seq profiling revealed similar chromatin accessibility states between mNSCs and mNSCs expressing ZR unmutated or any of the mutants, indicating that this fusion oncoprotein does not significantly or globally alter chromatin accessibility. Instead, we have proposed a model in which ZR binding sites are accessible in distinct cells, namely radial glial progenitor cells, present during embryonic brain development (<https://www.biorxiv.org/content/10.1101/2024.08.12.607603v1.full>) [1]. ZR hijacks PLAG family transcription factor motifs and maintains accessibility at proliferation genes that are normally repressed during neuronal differentiation. We therefore propose a model in which ZR condensate formation sustains specific epigenomic programs in developing RGC progenitor cells as a mechanism of neoplastic transformation.

Response Figure 20: (A) Western blot validation of constitutive expression of indicated construct in stable mNSC cell lines used for ATAC-seq. **(B)** ATAC-seq comparison between mNSC and stable mSNC cells expression each indicated protein.

Reviewer 2, Point 8:

It is not surprising that mutation of the ZFTA zinc finger domain in ZR inhibited ZR DNA binding and in doing so abrogated the transcriptomic and tumorigenic properties of ZR. It is, however, a bit surprising that the zinc-finger mutant, ZR-ZF2A, abolished the formation of condensates. How does this jive with the identification of the ZR IDR3 to be required to form condensates? Is oncogenic fusion protein binding to DNA a requisite for condensate formation – cannot just proteins with the correct IDRs form condensates, regardless of their stability to bind DNA?

Response:

We thank the reviewer for this comment and apologize for the lack of clarity. We've seen that although mutations in the ZFTA zinc finger domain abrogates DNA binding, the ZR-ZF2A protein is still capable of forming nuclear condensates. These remaining condensates are far fewer in numbers, but larger than the ones seen for the unmutated wildtype ZR protein. These condensates appear to be in active equilibrium with their surroundings as evidenced by the FRAP experiment shown in **(Response Figure 21, Extended Data Figure 8C-D)**. Moreover, our *in vitro* analysis reveals that purified ZR-IDR3 is capable of forming condensates in the presence of a crowding agent, in contrast with ZR-IDR3-75DE and -75GS mutants that lose this capability **(Response Figure 21C, Figure 3I, Extended Data Figure 4G)**.

Response Figure 21: (A) FRAP analysis of G-ZR-ZF2A. Representative images of pre-bleach, bleach and post-bleach are provided. (B) Normalized relative fluorescence intensity of G-ZR-ZF2A is provided over time following bleach. Values are averaged over 10 cells. (C) Representative images of in vitro droplet assays performed using IDR3 unmutated, IDR-75DE or IDR3-75GS. Coomassie stained protein gel image of each eluent is provided on the left side to indicate the purity of protein samples.

Reviewer 2, Point 9:

Do physical means such as osmotic stress effect the ability of the IDR3-mut to form condensates?

Response:

Osmotic regulation may contribute to the proliferation, survival, and migration of cancer cells. We agree with Reviewer #2 that this may be an important factor that regulates ZR function, but we have not yet measured these effects in a natively forming tumor microenvironment. We feel that these studies need to be pursued *in vivo*, first, before extension of osmotic stress related studies in ZR condensates and their associated IDR mutants. We thank Reviewer #2 for this comment that will lead to a new line of future studies, likely to provide important insights into ZR oncogenic function.

Reviewer 2, Point 10:

Finally, and most importantly, one must question the novelty of this manuscript in that there are a growing number of reports demonstrating the necessity of IDR-mediated condensate formation for the tumorigenic role of various oncogenic fusion proteins – including the Yap-fusion proteins role in supratentorial ependymomas as reported in Nature Cell Biology just a few months ago (reference #24). Thus, even should the authors choose to perform more extensive mechanistic studies as

suggested above, the novelty of the major observations reported in this manuscript are modest at best and rather predictable given the current state of the literature, thereby not warranting publication in as high impact a journal as NCB in my opinion.

We are grateful for the comments that *Reviewer #2* has provided that has (in our opinion) led to an improved manuscript and supportive of the initial findings. While descriptions of *YAP1* gene fusions and condensates have previously been reported in supratentorial (ST) ependymoma, these *YAP1*-driven subtypes of ependymoma are exceedingly rare, and constitute less than 2% of all ependymomas observed in a recent North American clinical trial (*Children's Oncology Group-ACNS0831, correspondence: Dr. David Ellison*). By comparison, *ZFTA*-gene fusion driven supratentorial ependymomas constitute the vast majority of supratentorial ependymoma cases (>90%). Importantly, many in the neuro-oncology community consider *YAP1* fusion ependymoma as molecularly and clinically disparate disease entities [8-10]. In our view, extensions of *YAP1*-fusion biology to *ZFTA*-fusion biology cannot be made without experimentally studying the mechanisms of *ZFTA*-fusion proteins. We feel that our manuscript reveals important insights into the molecular function of *ZFTA-RELA* and *ZFTA*-associated fusion oncoproteins in terms of biophysical behavior, oncogene regulation, and *in vivo* tumorigenesis. We also link these findings to structural components of *ZFTA*, including the characterization of a key zinc finger domain by NMR spectroscopy. Guided by the comments of *Reviewer #2*, we also demonstrate the generation of synthetic fusion oncoproteins that form nuclear biomolecular condensates, drive transcription, and initiate brain tumors in mice. These gain-of-function studies have potentially broader ramifications for IDRs enriched across oncoproteins in cancer and the general relevance to condensate formation, oncogene expression, and tumor initiation. We thank *Reviewer #2* for their insightful comments that we feel have led to an improved manuscript that is suitable for reaching the wide readership of *Nature Cell Biology*.

D) RESPONSE TO REVIEWER #3:

Remarks to the Author: Comments on “ZFTA fusion oncoproteins drive tumorigenesis through transcriptional condensates” by Mack, Kriwacki and colleagues

In this study, Arabzade et al. studies driver genes/proteins in ependymomas (EPNs), an aggressive brain tumor. These drivers are oncofusions due to chromosome translocation. Over 95% of EPNs are driven by a gene fusion involving the zinc finger translocation associated (ZFTA) protein, and less than 5% EPNs involves fusions with YAP1. The most frequent fusion partner with ZFTA is RELA and the fusion gene is called ZR. The molecular mechanisms of ZR in driving tumorigenesis are the focus of this study.

They firstly showed that ZR forms dynamic nuclear condensates reminiscent of transcription condensates in EPN models. Indeed the ZR condensates enrich with transcriptional machineries at ependymoma oncogenes including NOTCH1 and CCND1. Using mutagenesis analysis, they nailed an IDR, IDR3, responsible for condensation. By grossly mutating three types of enriched amino acid residuals (P, A, Q), they generated ZR variants with progressively weaker condensation capacity. Apparently condensation encoded within IDR3 of ZR is required for its chromatin binding, transcriptional activity and oncogenicity. Using NMR and AlphaFoldDB structural analysis, the authors designed mutation within ZFTA zinc finger domain (called ZF1) that abolish its DNA binding capacity. This mutation also loses condensation for oncogenic transcription. ZR's condensation capacity appears to be general among oncofusions for EPNs as all other known ZFTA fusions are predicted to form condensates and many are experimentally shown to form transcriptional condensates.

This study has convincingly showed that condensates by a number fusion genes are important drivers for oncogenicity in a class of solid tumor. Given that being said, a few crucial link is required for further substantiation. Please see my critiques below for details.

We thank *Reviewer #3* for their positive comments and critiques that have led to new experimental datasets included in the revised manuscript.

MAJOR COMMENTS:

Reviewer 3, Point 1:

To establish the causality between condensation and transcriptional activation, oncogenicity, the authors used loss-of-function mutations such as deletion of IDR3 or global substitution of 25%, 50% or even 75% of enriched amino acids. It will substantially strengthen the story if the authors can carry out rescue-type of experiment. E.g. grafting orthogonal phase separation-competent IDRs into DeltaIDR3 or MUT25, MUT50, MUT75 and see whether transcriptional activation and oncogenicity are restored upon re-introducing condensation into the mutant ZR.

Response

To address *Reviewer #3* comments that only loss-of-function ZR IDR substitution mutants were described, we have since made synthetic ZR mutants, replacing the IDR3 region of ZR with IDRs from other proteins known to drive nuclear condensate formation (e.g., *EWSR1* and *FUS*) (**Response Figure 22, Figure 7**). These synthetic mutants (see schematic) have never been described before in biomedical research. Critically, synthetic fusions that formed nuclear condensates, similar to ones formed by ZFTA-RELA, were able to drive expression of known ZFTA-RELA target genes and could initiate brain tumors *in vivo* with a similar transcriptional signature compared to ZR ependymoma. (**Response Figure 22, Figure 7**). To our knowledge, we are amongst the first to investigate the

impact of IDR 'grafting' as a gain-of-function approach to understand the impact of nuclear condensates on transcription and brain tumor initiation. We thank *Reviewer #3* for their critical feedback that has helped further support the conclusions of our manuscript.

Response Figure 22: (A) Schematic cartoon of “IDR swapping” experiment from designing the synthetic fusions to downstream functional assessments (B) Schematic representation of the synthetic fusions. (C) Representative images of HEK293T cells expressing indicated synthetic fusion. (D) ZR downstream gene expression comparison between ZFTA-RELA and synthetic fusions (E) Representative MRI images of tumors generated using ZR-sub-FUS:IDR and ZR-sub-EWS:IDR synthetic fusions. (F) time to tumor detection between ZFTA-RELA, ZR-sub-EWS:IDR or ZR-sub-FUS:IDR synthetic fusions. (G) Global gene expression comparison for tumors generated using (G) ZR-sub-FUS:IDR or (H) ZR-sub-EWS:IDR compared to healthy brain tissue. ZR endependymoma signature genes are marked with red dots.

Reviewer 3, Point 2:

Please provide exact sequences of all fusion genes used in the study and do further analysis on the potential heterogeneity of the fusions of two identical genes. For example, do all ZFTA fusion genes in Figure 6 still contain ZF1? YAP1 doesn't bind DNA directly. It needs TEAD1-4 for DNA binding. Do YAP1-KDM2B and YAP-MAMLD1 still contain TEAD-binding domain? Figure 6D, the condensates of G-ZFTA-MAML2 look like homotypic condensates off chromatin. Does this fusion still contain DBD?

Response:

We apologize to *Reviewer #3* for the lack of clarity, and in response have included the sequences of all ZFTA fusion oncoproteins and associated mutant variants in **Supplementary Table 5**. To address their specific question regarding ZFTA fusion variants, all ZFTA fusion oncoproteins contain the ZFTA-ZF1 binding domain. YAP1-KDM2B, YAP1-MAMLD1, and YAP1-FAM11B8 all contain the TEAD binding domains. Finally, ZFTA-MAML2 contains all four ZFTA ZF DNA binding domains.

Reviewer 3, Point 3:

Figure 4H is not cited in the main text; Legends for Figure 2J-N are missing; Legends for Figure 5H are missing.

Response:

We apologize to *Reviewer #3* for this error and now provide the corrections in our revised manuscript.

MINOR COMMENTS:

Reviewer 3, Point 4:

Line 136, “Fig.2K,N” don't seem to support the text.

Response:

In our revised manuscript we have adjusted the text and figure call outs. We thank the *Reviewer #3*.

Reviewer 3, Point 5:

The fonts in figures are messy. Please use consistent fonts.

Response:

In the revised manuscript we have now provided figures and extended figures with uniform fonts and more consistent font sizes and styles. We apologize for the lack of clarity in the presentation of figures in our original submission.

Reviewer 3, Point 6:

Line 168, it should be 3D instead of 3C.

Response:

We apologize to *Reviewer #3* for this error and now provide the correction in our revised manuscript.

Reviewer 3, Point 7:

Please provide exact sequences of IDR3-MUT25, IDR3-MUT50, IDR3-MUT75.

Response:

We have included in **Supplementary Table 5** a listing of all of the mutants we have described along with coding sequences.

Reviewer 3, Point 8:

Paper citation needs more optimization. For example, Ref 13 is better re-cite at line 291. In addition, multiple nice papers on fusion oncoproteins are not cited. E.g. PMID: 33674598; PMID: 37400539; PMID: 29930090; PMID: 32929202.

Response:

We thank *Reviewer #3* for this comment and have included appropriate citations as recommended.

Reviewer 3, Point 9:

Line 702, the protein concentration can't be 380 mM.

Response:

We apologize to *Reviewer #3* for this error and now provide the correction (380 μ M) in our revised manuscript.

E) REFERENCE LIST

1. Kardian, A., et al., *Dominant Malignant Clones Leverage Lineage Restricted Epigenomic Programs to Drive Ependymoma Development*. bioRxiv, 2024: p. 2024.08.12.607603.
2. Arabzade, A., et al., *ZFTA-RELA Dictates Oncogenic Transcriptional Programs to Drive Aggressive Supratentorial Ependymoma*. *Cancer Discov*, 2021. **11**(9): p. 2200-2215.
3. Parker, M., et al., *C11orf95-RELA fusions drive oncogenic NF- κ B signalling in ependymoma*. *Nature*, 2014. **506**(7489): p. 451-5.
4. Zheng, T., et al., *Cross-Species Genomics Reveals Oncogenic Dependencies in ZFTA/C11orf95 Fusion-Positive Supratentorial Ependymomas*. *Cancer Discov*, 2021. **11**(9): p. 2230-2247.
5. Kupp, R., et al., *ZFTA Translocations Constitute Ependymoma Chromatin Remodeling and Transcription Factors*. *Cancer Discov*, 2021. **11**(9): p. 2216-2229.
6. Chen, H.C., et al., *Histone serotonylation regulates ependymoma tumorigenesis*. *Nature*, 2024. **632**(8026): p. 903-910.
7. Herranz-Pérez, V., et al., *Ependymoma associated protein Zfta is expressed in immature ependymal cells but is not essential for ependymal development in mice*. *Sci Rep*, 2022. **12**(1): p. 1493.
8. Pajtler, K.W., et al., *YAP1 subgroup supratentorial ependymoma requires TEAD and nuclear factor I-mediated transcriptional programmes for tumorigenesis*. *Nat Commun*, 2019. **10**(1): p. 3914.
9. Pajtler, K.W., et al., *Molecular Classification of Ependymal Tumors across All CNS Compartments, Histopathological Grades, and Age Groups*. *Cancer Cell*, 2015. **27**(5): p. 728-43.
10. Capper, D., et al., *DNA methylation-based classification of central nervous system tumours*. *Nature*, 2018. **555**(7697): p. 469-474.